# Provable Domain Adaptation for Offline Reinforcement Learning with Limited Samples

**Weiqin Chen**                                        *chenw18@rpi.edu*
*Rensselaer Polytechnic Institute*

**Xinjie Zhang**                                       *xz3236@columbia.edu*
*Columbia University*

**Sandipan Mishra**                                    *MISHRS2@rpi.edu*
*Rensselaer Polytechnic Institute*

**Santiago Paternain**                                 *paters@rpi.edu*
*Rensselaer Polytechnic Institute*

**Reviewed on OpenReview:** *https://openreview.net/forum?id=xog8ThcXwy*

## Abstract

Offline reinforcement learning (RL) learns effective policies from a static target dataset. The performance of state-of-the-art offline RL algorithms notwithstanding, it relies on the size of the target dataset, and it degrades if limited samples in the target dataset are available, which is often the case in real-world applications. To address this issue, domain adaptation that leverages auxiliary samples from related source datasets (such as simulators) can be beneficial. However, establishing the optimal way to trade off the limited target dataset and the large-but-biased source dataset while ensuring provably theoretical guarantees remains an open challenge. To the best of our knowledge, this paper proposes the first framework that theoretically explores the impact of the weights assigned to each dataset on the performance of offline RL. In particular, we establish performance bounds and the existence of the optimal weight, which can be computed in closed form under simplifying assumptions. We also provide algorithmic guarantees in terms of convergence to a neighborhood of the optimum. Notably, these results depend on the quality of the source dataset and the number of samples in the target dataset. Our empirical results on the well-known *Procgen* and *MuJoCo* benchmarks substantiate the theoretical contributions in this work.

## 1 Introduction

Deep reinforcement learning (RL) has demonstrated impressive performance in a wide variety of applications, such as strategy games (Mnih et al., 2013; 2015), robotics (Levine et al., 2016; Duan et al., 2016), and recommender systems (Afsar et al., 2022; Lin et al., 2023). RL aims to learn an optimal policy that maximizes the expected discounted cumulative reward. To achieve this goal, the online RL agent learns and improves the policy by actively interacting with the environment. However, this poses a critical challenge for the real-world applications of RL, as interactions with the real-world can be significantly dangerous and expensive (Kumar et al., 2020; Levine et al., 2020; Chen et al., 2024). In this context, offline RL has emerged as a promising alternative framework for the real-world applications of RL, where the agent learns effective policies from a static and previously collected dataset.

Recent advances in offline RL algorithms have shown remarkable success across a diverse array of problems and datasets (Fujimoto et al., 2019a; Kumar et al., 2020; Kostrikov et al., 2021; Chen et al., 2021). Nevertheless, their effectiveness depends on the quality and size of the dataset. More concretely, it is worth noting

that even state-of-the-art (SOTA) offline RL algorithms like BCQ (Fujimoto et al., 2019a), CQL (Kumar et al., 2020), IQL (Kostrikov et al., 2021), DT (Chen et al., 2021) demonstrate poor performance given a limited offline RL dataset, as training on a small number of samples may lead to overfitting (Fu et al., 2019; Kumar et al., 2019). In this work, we are interested in offline RL that learns from a static dataset with limited samples. We now proceed by introducing a series of related works.

## 1.1 Related Work

**Offline reinforcement learning with dataset distillation.** Dataset distillation (Wang et al., 2018) proposes a framework for synthesizing a smaller and more efficient dataset by minimizing the gradient discrepancy of the samples from the original dataset and the distilled dataset. Synthetic (Light et al., 2024) is the first work that applies dataset distillation to offline RL and achieves comparable performance with the original large offline RL dataset. Specifically, it synthesizes a small distilled dataset by minimizing the gradient matching loss between the original offline RL dataset and the synthetic dataset. However, generating the synthetic dataset necessitates access to the original large offline RL dataset, which is often impractical in real-world scenarios. In particular, this work focuses on scenarios where only a limited number of samples are accessible at all times.

**Offline reinforcement learning with domain adaptation.** To avoid overfitting by learning from the *limited target dataset*, domain adaptation techniques (Redko et al., 2020; Farahani et al., 2021) propose to leverage an auxiliary large *source* dataset (such as simulators) with unlimited samples. H2O (Niu et al., 2022) assumes access to an unrestricted simulator, and the process of training on simulators still requires interacting with the environment online. On the other hand, ORIS (Hou et al., 2024) proposes to generate a new (source) dataset from the simulators, where a generative adversarial network (GAN) model is employed to approximate the state distribution of the original target dataset. Starting from the initial state provided by GAN, the new (source) dataset is generated by interacting with the simulator and reweighted by an additional discriminator model. These approaches emphasize the algorithmic design by employing sample-dependent weights to combine the source and target datasets during the learning process, e.g., through weighting based on the transition ratio between the source and target domains.

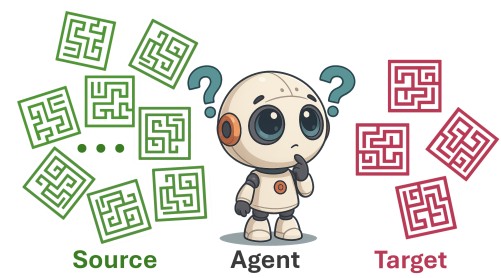

Figure 1: Schematic: the target dataset has limited samples (red), whereas the source dataset has unlimited samples (green) with a dynamics gap from the target dataset. How to strike a proper balance between the two datasets?

Given the unlimited source dataset and the limited target dataset, however, striking a proper balance between the two datasets prior to the actual learning process remains a challenging problem in offline RL (see Figure 1). Straightforward strategies involve either combining both datasets equally or using only one of them, nevertheless, neither approach offers optimality guarantees. Accordingly, this work focuses on exploring the impact of the weights assigned to each dataset on the performance of offline RL, particularly from a provably theoretical perspective.

**Offline reinforcement learning with dynamics gap.** Dynamics gap in the domain adaptation and transfer for offline RL is widely acknowledged as a significant challenge. From a practical standpoint, DARC (Eysenbach et al., 2020) and DARA (Liu et al., 2022) train a dynamics gap-related penalty by minimizing the divergence between the real and simulator trajectory distributions, and combine it with the simulator reward during the online/offline training of simulators. On the other hand, HTRL (Qu et al., 2024) theoretically investigates the sample efficiency of a hybrid transfer RL framework with a dynamics gap, which, however, requires access to the entire target domain. To our knowledge, no prior work has theoretically investigated the impact of the dynamics gap on selecting the weights assigned to the limited target dataset and the unlimited source dataset.

### 1.2 Main Contributions

The main contributions of this work are summarized as follows.

- To the best of our knowledge, this is the first work that proposes an algorithm-agnostic framework of domain adaptation for offline RL prior to the actual learning process, which exhibits a provably theoretical trade-off between the number of samples in a limited target dataset and the dynamics gap (or discrepancy) between the target and source domains.

- We establish the (expected and worst-case) performance bounds and the convergence to a neighborhood of the optimum within our framework. We further identify, under simplifying assumptions, the existence of an optimal weight for balancing the two datasets, which is typically not one of the trivial choices: treating both datasets equally or using either dataset only. All theoretical guarantees and the optimal weight will depend on the quality of the source dataset (refer to the dynamics gap) and the size of the target dataset (i.e., number of samples).

- A series of numerical experiments conducted on the well-known *Procgen* and *MuJoCo* benchmarks substantiate our theoretical contributions.

Notice that the primary focus of this paper is on the theoretical analysis of our proposed framework, while algorithm development is reserved for future research.

## 2 Balancing Target and Source Datasets

In this section, we consider the mathematical formalism of offline RL (Levine et al., 2020), namely *Markov Decision Process* (MDP) (Sutton & Barto, 2018). This work centers on the tabular MDP defined by a tuple $\mathcal{M} = (\mathcal{S}, \mathcal{A}, P, r, \rho, \gamma, T)$, where $\mathcal{S}$ and $\mathcal{A}$ are finite state and action spaces (a standard assumption in RL for theoretical guarantees), $P : \mathcal{S} \times \mathcal{A} \times \mathcal{S} \to [0, 1]$ denotes the transition probability that describes the dynamics of the system, $r : \mathcal{S} \times \mathcal{A} \to \mathbb{R}$ denotes the reward function that evaluates the quality of the action, $\rho : \mathcal{S} \to [0, 1]$ denotes the initial state distribution, $\gamma$ represents the discount factor, and $T \in \{0, 1, 2, \cdots\}$ defines the horizon length.

A *policy* is represented by the probability distribution over actions conditioned on states. In each state $s$, the agent selects an action $a$ based on the policy $\pi(a|s)$, and generates the next state $s'$. The tuple $(s, a, s')$ is referred to as the *transition* data. Offline RL considers employing a behavior policy $\pi_\beta$ to collect an offline and static dataset with $N$ transitions ($N \in \mathbb{N}_+$), i.e., $\hat{\mathcal{D}}_{\mathrm{tr}} = \{(s_i, a_i, s_i')\}_{i=1}^N$. The transitions in the dataset $\hat{\mathcal{D}}_{\mathrm{tr}}$ are collected from a domain $\mathcal{D}$. The goal of RL is to learn an optimal policy $\pi^*$ that maximizes the expected discounted cumulative reward, i.e.,

$$\pi^* = \arg\max_\pi \mathbb{E}_{\substack{s_0 \sim \rho,\, a_t \sim \pi(a_t|s_t) \\ s_{t+1} \sim P(s_{t+1}|s_t, a_t)}} \left[ \sum_{t=0}^T \gamma^t r(s_t, a_t) \right]. \tag{1}$$

In the context of *offline* RL, the policy needs to be learned from the static dataset $\hat{\mathcal{D}}_{\mathrm{tr}}$ exclusively. Offline RL algorithms (Fujimoto et al., 2019a; Levine et al., 2020; Kostrikov et al., 2021) typically train an action-value function (or Q-function) by minimizing the temporal difference (TD) error iteratively. To be formal, let $\mathcal{B}$ be a Bellman operator. This operator takes different forms depending on the specific algorithm considered. For instance, in Q-learning type methods the operator takes the form

$$\mathcal{B}Q^k(s, a) = r(s, a) + \gamma \mathbb{E}_{s' \sim P(s'|s,a)} \left[ \max_{a'} Q^k(s', a') \right], \tag{2}$$

whereas for actor-critic methods it takes the form

$$\mathcal{B}^\pi Q^k(s, a) = r(s, a) + \gamma \mathbb{E}_{s' \sim P(s'|s,a),\, a' \sim \pi(a'|s')} \left[ Q^k(s', a') \right]. \tag{3}$$

Given any transition $(s, a, s')$, let us define the TD error for any $Q$-function at the step $k$ as

$$\mathcal{E}(Q, (s, a, s')) := \left(Q(s, a) - \hat{\mathcal{B}}_{s'} Q^k(s, a)\right)^2, \tag{4}$$

where $\hat{\mathcal{B}}_{s'}$ denotes the stochastic approximation of the Bellman operator, namely a version of (2) or (3) where the expectation is replaced by evaluating the random variable at a single of realization $s'$. Note that we have dropped the dependence on $k$ on the left hand side of the above expression for simplicity. We further define the expected TD error in the domain $\mathcal{D}$ as

$$\mathcal{E}_{\mathcal{D}}(Q) = \mathop{\mathbb{E}}_{(s, a, s') \sim \mathcal{D}} \left[\mathcal{E}(Q, (s, a, s'))\right], \tag{5}$$

where $(s, a, s') \sim \mathcal{D}$ denotes that the transition $(s, a, s')$ is drawn from the probability distribution $P_{\mathcal{D}}(s, a, s')$ in the domain $\mathcal{D}$. With a slight abuse of notation, we also denote $(s, a) \sim \mathcal{D}$ to represent that the state-action pair $(s, a)$ is drawn from the probability distribution $P_{\mathcal{D}}(s, a)$ in the domain $\mathcal{D}$. With these definitions, one can define the iterations of a large class of offline RL algorithms through the following optimization problem

$$Q^{k+1} = \arg\min_Q \mathcal{E}_{\mathcal{D}}(Q), \ \forall k \in \mathbb{N}. \tag{6}$$

It is worth pointing out that the specific forms of (2) and (3) can result in poor performance in offline RL attributed to the issues with bootstrapping from out-of-distribution (OOD) actions (Fujimoto et al., 2019b; Kumar et al., 2019; 2020; Levine et al., 2020). This typically leads to an overestimation of the Q-value. To avoid this overestimation, prior works consider solely using in-distribution state-action pairs to maintain the Q-function (Fujimoto et al., 2019a), or constraining the learned policy to remain closely aligned with the behavior policy (Levine et al., 2020).

Note that the expectation in (5) poses a challenge in solving problem (6): it requires visiting every transition infinite times. In practice, one defines the empirical version of the TD error $\mathcal{E}_{\mathcal{D}}(Q)$ in (5),

$$\mathcal{E}_{\hat{\mathcal{D}}}(Q) = \frac{1}{N} \sum_{i=1}^{N} \mathcal{E}(Q, (s_i, a_i, s'_i)), \tag{7}$$

where the samples are from the dataset $\hat{\mathcal{D}}_{\text{tr}}$. Then, the offline RL algorithm is defined as the minimization of the stochastic approximation of problem (6)

$$\hat{Q}^{k+1} = \arg\min_Q \mathcal{E}_{\hat{\mathcal{D}}}(Q), \ \forall k \in \mathbb{N}. \tag{8}$$

It has been widely demonstrated that SOTA offline RL algorithms, such as BCQ (Fujimoto et al., 2019a), CQL (Kumar et al., 2020), IQL (Kostrikov et al., 2021), and DT (Chen et al., 2021), are capable of solving problem (6) given sufficient transition samples from the domain $\mathcal{D}$. Nevertheless, this assumption may not be realizable in practice, e.g., healthcare (Tang et al., 2022) and autonomous driving (Pan et al., 2017), where data collection is challenging. In this regime, $\hat{Q}^{k+1}$ generally demonstrates poor performance in approximating $Q^{k+1}$, as training on a small number of samples can lead to overfitting (Fu et al., 2019; Kumar et al., 2019).

On the other hand, in certain applications, one can rely on simulators (or related datasets) that provide a larger number of samples $\mathcal{D}' = \{(s_j, a_j, s'_j)\}_{j=1}^{N'}$ with $N' \gg N$. It is worth noting that, in general, $\mathcal{D}'$ will differ from $\mathcal{D}$ in terms of the state distribution and transition probabilities. Similar to (5), we define

$$\mathcal{E}_{\mathcal{D}'}(Q) = \mathop{\mathbb{E}}_{(s_j, a_j, s'_j) \sim \mathcal{D}'} \left[\mathcal{E}(Q, (s_j, a_j, s'_j))\right], \tag{9}$$

where $(s_j, a_j, s'_j) \sim \mathcal{D}'$ denotes that the transition $(s_j, a_j, s'_j)$ is drawn from the probability distribution $P_{\mathcal{D}'}(s_j, a_j, s'_j)$ in the source domain $\mathcal{D}'$.

We then explore in this paper a general scheme to combine the limited target dataset and the large-but-biased source dataset

$$Q_\lambda^{k+1} = \arg\min_Q (1 - \lambda)\mathcal{E}_{\hat{\mathcal{D}}}(Q) + \lambda\mathcal{E}_{\mathcal{D}'}(Q), \ \forall k \in \mathbb{N}, \tag{10}$$

where $\lambda \in [0, 1]$ denotes the weight that trades off $\mathcal{E}_{\hat{\mathcal{D}}}(Q)$ from the limited target dataset and $\mathcal{E}_{\mathcal{D}'}(Q)$ from the large-but-biased source dataset. Particularly, $\lambda \approx 1$ prioritizes the minimization of the TD error corresponding to the *source* dataset. This approach is suitable in cases where the *source* dataset is similar to the *target* (or coming from the same domain in an extreme case). On the other hand, $\lambda \approx 0$ focuses on minimizing the TD error in the *target* dataset. This method is appropriate for scenarios where data is abundant (sufficiently large $N$) or where dynamics gaps between the *target* and *source* datasets are too large.

Given these observations, it is expected that different values of $\lambda$ can attain the optimal performance, depending on the interplay between the available number of target samples and the dynamics gap between the two datasets. The following section formalizes this expectation.

## 3 Performance and Convergence Guarantees

We start this section by discussing the necessary assumptions to develop our theoretical results concerning the generalization of the solution to problem (10).

**Assumption 1.** *There exists $B > 0$ such that, for any $(s, a) \in \mathcal{S} \times \mathcal{A}$, $|r(s, a)| < B$.*

Assumption 1 is common in the literature (Azar et al., 2017; Wei et al., 2020; Zhang et al., 2021). In particular, in the case of finite state-action spaces, it is always possible to design the reward to avoid the possibility of being unbounded. Further notice that this assumption, along with the fact that a geometric series with ratio $\gamma \in [0, 1)$ converges to $1/(1 - \gamma)$ implies that

$$\max_{(s,a)\in\mathcal{S}\times\mathcal{A}} Q(s, a) \leq \frac{B}{1 - \gamma}. \tag{11}$$

**Assumption 2.** *(9) can be computed given a sufficiently large amount of samples from source domain $\mathcal{D}'$.*

To be precise, in practice one must work with the empirical counterpart of (9). Note, however, that since $N' \gg N$, the generalization error arising from the finite number of samples in the source dataset is negligible. For this reason and to simplify the exposition, we assume that (9) can be computed directly and, with a slight abuse of notation, denote $\mathcal{D}'$ to refer to both the source dataset and the source domain.

To proceed, we denote by $P_{\mathcal{D}'}(s, a)$ and $P_{\mathcal{D}}(s, a)$ the probability of a state-action pair $(s, a)$ from the source and target domains. Let us define $\hat{\mathcal{D}} = \{(s, a) \in \mathcal{S} \times \mathcal{A} \mid (s, a, \cdot) \in \hat{\mathcal{D}}_{\text{tr}}\}$ as the *transition-excluded dataset* of $\hat{\mathcal{D}}_{\text{tr}}$, $N(s, a) \in \{0, 1, \ldots, N\}$ as the number of the specific $(s, a, \cdot)$ transitions in the target dataset $\hat{\mathcal{D}}_{\text{tr}}$, and $P_{\hat{\mathcal{D}}}(s, a) = N(s, a)/N$.

**Assumption 3.** *For any $(s, a) \in \mathcal{S} \times \mathcal{A}$, $P_{\mathcal{D}}(s, a)$ and $P_{\mathcal{D}'}(s, a)$ are positive. Given any dataset $\hat{\mathcal{D}}_{tr}$ and its corresponding transition-excluded dataset $\hat{\mathcal{D}}$, there exist constants $\beta_u \geq \beta_l > 0$ such that $P_{\hat{\mathcal{D}}}(s, a)/P_{\mathcal{D}'}(s, a) \in [\beta_l, \beta_u], \ \forall (s, a) \in \hat{\mathcal{D}}$.*

**Remark 1.** *i) Assumption 3 posits that every state-action pair $(s, a)$ in the space $\mathcal{S} \times \mathcal{A}$ has non-zero probability of occurring in the source domain $\mathcal{D}'$ and the target domain $\mathcal{D}$, thus yielding a bounded ratio $P_{\mathcal{D}}(s, a)/P_{\mathcal{D}'}(s, a), \ \forall (s, a) \in \mathcal{S} \times \mathcal{A}$. This is a mild assumption in the sense that if $P_{\mathcal{D}}(s, a) = 0$ or $P_{\mathcal{D}'}(s, a) = 0$ essentially implies that the state-action pair $(s, a)$ can be ignored in the corresponding domain of interest. ii) Nevertheless, since the limited dataset $\hat{\mathcal{D}}$ is unlikely to cover the entire space of $\mathcal{S} \times \mathcal{A}$, we define $\beta_l$ and $\beta_u$ as the lower and upper bounds, respectively, of the ratio $P_{\hat{\mathcal{D}}}(s, a)/P_{\mathcal{D}'}(s, a)$ for all $(s, a) \in \hat{\mathcal{D}}$. In this sense, $P_{\hat{\mathcal{D}}}(s, a) = N(s, a)/N = 0$ for any $(s, a) \in \mathcal{S} \times \mathcal{A} \setminus \hat{\mathcal{D}}$.*

It is worth noting that the transition sample $\tau$ is independent and identically distributed (i.i.d.), as the dataset $\hat{\mathcal{D}}_{\text{tr}}$ is shuffled and the transition data are sampled i.i.d. Alternatively, one can think of the dataset as random samples from the occupancy measure, followed by a random transition.

**Remark 2.** *It is crucial to note that (10) relies solely on the limited target dataset when $\lambda = 0$, i.e., $Q_\lambda^{k+1} = \arg\min_Q (1 - \lambda)\mathcal{E}_{\hat{\mathcal{D}}}(Q) + \lambda\mathcal{E}_{\mathcal{D}'}(Q) = \arg\min_Q \mathcal{E}_{\hat{\mathcal{D}}}(Q)$. In this context, it becomes infeasible to compute $Q_\lambda^{k+1}$ for any state–action pair outside the transition-excluded dataset $\hat{\mathcal{D}}$. Hence, throughout this work, we assume $\lambda \in (0, 1]$ for any $(s, a) \in \mathcal{S} \times \mathcal{A} \setminus \hat{\mathcal{D}}$, which guarantees that $(1-\lambda)P_{\hat{\mathcal{D}}}(s, a) + \lambda P_{\mathcal{D}'}(s, a) > 0$.*

Having formally stated all assumptions in this work, we are now in conditions to proceed with the analysis, where we start by deriving the analytical expressions of $Q^{k+1}$ and $Q_\lambda^{k+1}$.

**Proposition 1.** *Let Assumptions 2 and 3 hold. Recall the empirical Bellman operator $\hat{\mathcal{B}}$ in (4). Denote by $\mathcal{B}_{\mathcal{D}}$ ($\mathcal{B}_{\mathcal{D}'}$) the Bellman operator in (2) or (3) in which $s'$ follows the transition probability of the domain $\mathcal{D}$ ($\mathcal{D}'$). Note that $Q^{k+1}$ and $Q_\lambda^{k+1}$ represent the solutions to (6) and (10). Given any dataset $\hat{\mathcal{D}}_{tr}$ and its corresponding transition-excluded dataset $\hat{\mathcal{D}}$, denote by $N$ and $N(s, a)$ the total number of samples and the amount of $(s, a, \cdot)$ transition in $\hat{\mathcal{D}}_{tr}$. At each iteration ($k = 0, 1, 2, \cdots$), it holds that*

$$Q^{k+1}(s, a) = \mathcal{B}_{\mathcal{D}}Q^k(s, a), \forall (s, a) \in \mathcal{S} \times \mathcal{A}, \tag{12}$$

$$Q_\lambda^{k+1}(s, a) = \frac{\frac{1-\lambda}{N} \sum_{j=1}^{N(s,a)} \hat{\mathcal{B}}_{\hat{s}'_j}Q^k(s, a) + \lambda P_{\mathcal{D}'}(s, a)\mathcal{B}_{\mathcal{D}'}Q^k(s, a)}{(1 - \lambda)P_{\hat{\mathcal{D}}}(s, a) + \lambda P_{\mathcal{D}'}(s, a)}, \forall (s, a) \in \mathcal{S} \times \mathcal{A}. \tag{13}$$

*Proof.* Refer to Appendix A.1. □

In addition to the analytical expressions of $Q^{k+1}(s, a)$ and $Q_\lambda^{k+1}(s, a)$, Proposition 1 implies that $Q_\lambda^{k+1}(s, a)$ (the right hand side of (13)) reduces to $\mathcal{B}_{\mathcal{D}'}Q^k(s, a)$, an analog of $Q^{k+1}(s, a)$ solely replacing $\mathcal{D}$ by $\mathcal{D}'$.

We proceed by exploring the performance of $Q_\lambda^{k+1}$, as in our problem of interest (10). To do so, we next define two quantities that will play important roles in the theoretical guarantees of $Q_\lambda^{k+1}$. Start by defining the dynamics gap (or discrepancy) between the target and source domains

$$\xi = \max_{(s,a)\in\mathcal{S}\times\mathcal{A}} \left[ \left(\mathcal{B}_{\mathcal{D}}Q^k(s, a) - \mathcal{B}_{\mathcal{D}'}Q^k(s, a)\right)^2 \right], \tag{14}$$

where, for notation simplicity, we omit the dependence of $\xi$ on the iteration $k$. Indeed, if the transition probabilities in the source and target domains match, the quantity $\xi$ in (14) becomes zero. Then, given any dataset $\hat{\mathcal{D}}_{tr}$ and its corresponding transition-excluded dataset $\hat{\mathcal{D}}$, we define a measure of the variability

$$\varsigma = \max_{(s,a)\in\hat{\mathcal{D}}} \left[ \frac{\sigma_{\hat{s}'}^2 \left(\hat{\mathcal{B}}_{\hat{s}'}Q^k(s, a)\right)}{N(s, a)} \right], \tag{15}$$

where $\sigma_{\hat{s}'}^2 \left(\hat{\mathcal{B}}_{\hat{s}'}Q^k(s, a)\right)$ denotes the variance of the empirical Bellman operator. The term $\varsigma$ above is the maximum normalized variance in the given target dataset. It is significant to note that the number of samples to keep the value $\varsigma$ constant is proportional to the variance.

With these definitions and the analytical expression of $Q_\lambda^{k+1}$, we are now in conditions of establishing the bound on the expected TD error of $Q_\lambda^{k+1}$ over the target domain.

**Theorem 1** (Expected Performance Bound). *Recall $\xi$ in (14), $\varsigma$ in (15) and define $C := \min_{(s,a)\in\mathcal{S}\times\mathcal{A}} P_{\mathcal{D}}(s, a)$. Let the conditions of Proposition 1 hold. Given any dataset $\hat{\mathcal{D}}_{tr}$, it holds at each iteration ($k = 0, 1, 2, \cdots$) that*

$$\mathbb{E}_{\hat{s}'\sim P_{\mathcal{D}}(\hat{s}'|s,a)} \left[\mathcal{E}_{\mathcal{D}}(Q_\lambda^{k+1})\right] - \mathcal{E}_{\mathcal{D}}(Q^{k+1}) \le \left(\frac{1-\lambda}{1 - \lambda + \lambda/\beta_u}\right)^2 \varsigma + \left(\left(\frac{\lambda}{(1-\lambda)\beta_l + \lambda}\right)^2 + e^{-NC}\right)\xi. \tag{16}$$

*Proof.* Refer to Appendix A.2. □

Recall that $\mathcal{E}_{\mathcal{D}}(Q^{k+1})$ represents the optimal TD error of the offline RL at iteration $k$, which cannot be computed in practice due to the finite number of samples in the target dataset. The significance of Theorem 1

is to establish a bound on the difference between the optimal TD error $\mathcal{E}_{\mathcal{D}}(Q^{k+1})$ and the expected TD error of $Q_{\lambda}^{k+1}$ (the solution to (10)) with respect to the next-state $\hat{s}'$ (from the limited target dataset) over the target domain transition probability $P_{\mathcal{D}}(\hat{s}' \mid s, a)$.

Notice that the bound in (16) arises from the general Assumption 3, which implies that the limited dataset $\hat{\mathcal{D}}$ does not cover the entire state–action space $\mathcal{S} \times \mathcal{A}$. As a result, this bound is loose in some scenarios. Before proceeding with a more detailed explanation, we formally present a tighter bound in the following theorem, under a stronger version of Assumption 3. This new result will make the discussion on the looseness of the bound in (16) more explicit.

**Theorem 2** (Tighter Expected Performance Bound)**.** *Let the conditions of Theorem 1 hold. Suppose that for any $(s,a) \in \mathcal{S} \times \mathcal{A}$, $P_{\mathcal{D}}(s,a)$, $P_{\mathcal{D}'}(s,a)$ and $P_{\hat{\mathcal{D}}}(s,a)$ are positive and there exist constants $\beta_u \geq \beta_l > 0$ such that $P_{\hat{\mathcal{D}}}(s,a)/P_{\mathcal{D}'}(s,a) \in [\beta_l, \beta_u]$, $\forall(s,a) \in \mathcal{S} \times \mathcal{A}$. Given any dataset $\hat{\mathcal{D}}_{tr}$, it holds at each iteration $(k = 0, 1, 2, \cdots)$ that*

$$\mathbb{E}_{\hat{s}' \sim P_{\mathcal{D}}(\hat{s}'|s,a)} \left[ \mathcal{E}_{\mathcal{D}}(Q_{\lambda}^{k+1}) \right] - \mathcal{E}_{\mathcal{D}}(Q^{k+1}) \leq \left( \frac{1-\lambda}{1-\lambda+\lambda/\beta_u} \right)^2 \varsigma + \left( \frac{\lambda}{(1-\lambda)\beta_l + \lambda} \right)^2 \xi. \tag{17}$$

*Proof.* Refer to Appendix A.3. $\qquad\square$

Theorems 1 and 2 imply that the expected performance bounds of $Q_{\lambda}^{k+1}$ in both (16) and (17) exhibit an intuitive form for any $\lambda \in (0, 1]$, as it jointly depends on the variance ($\varsigma$) of the limited target dataset and the bias ($\xi$) introduced by the large source dataset when $\lambda \in (0, 1)$, and reduces to the dependence on $\xi$ alone when $\lambda = 1$ (i.e., solely considers the source dataset $\mathcal{D}'$ in (10)).

Notwithstanding, we also note that the bound in (16) may be loose when $\lambda = 0$, since it depends on both $\varsigma$ and $\xi$, even though only the limited target dataset is employed in (10). Notably, such looseness does not manifest in the bound presented in Theorem 2 under $\lambda = 0$, which provides a tighter bound than that in (16). Yet, Theorem 2 relies on the stronger assumption that the limited dataset $\hat{\mathcal{D}}$ covers the entire state-action space $\mathcal{S} \times \mathcal{A}$, which is rarely the case in practice.

In addition to establishing the expected performance bounds of $Q_{\lambda}^{k+1}$, Theorems 1 and 2 imply the bias-variance trade-off sought by combining the two datasets with different weight $\lambda$ in (10). Indeed, the optimal weight $\lambda^*$ that minimizes the right hand side of (16) and (17) is discussed formally by the following corollaries. Although Theorem 2 provides a tighter bound, it is worth highlighting that Theorems 1 and 2 yield the same $\lambda^*$, as the extra term $e^{-NC}\xi$ in (16) is independent of $\lambda$.

**Corollary 1.** *Under the assumptions of Theorem 1 or Theorem 2, the optimal weight $\lambda^*$ that minimizes the bounds in (16) or (17) respectively is $\lambda^* = 0$ when $\varsigma = 0$ and $\lambda^* = 1$ when $\xi = 0$.*

*Proof.* Refer to Appendix A.4. $\qquad\square$

**Corollary 2.** *Under the assumptions of Theorem 1 or Theorem 2, if $\beta_l = \beta_u = \beta > 0$, the optimal weight $\lambda^*$ that minimizes the bound in (16) or (17) respectively takes the form*

$$\lambda^* = \frac{\beta\varsigma}{\beta\varsigma + \xi}. \tag{18}$$

*Proof.* Refer to Appendix A.5. $\qquad\square$

Recall that $\beta_l$ and $\beta_u$ denote the lower and upper bounds of the ratio $P_{\hat{\mathcal{D}}}(s,a)/P_{\mathcal{D}'}(s,a)$, $\forall(s,a) \in \hat{\mathcal{D}}$. Thus, the assumption $\beta_l = \beta_u = \beta$ in Corollary 2 implies that $P_{\hat{\mathcal{D}}}(s,a)/P_{\mathcal{D}'}(s,a)$ is a fixed ratio for any $(s,a) \in \hat{\mathcal{D}}$. This may occur when the sampling of the target dataset proportionally follows the source distribution for all $(s,a) \in \hat{\mathcal{D}}$. Since $\hat{\mathcal{D}}$ (relates to the target dataset) comprises fewer state–action pairs than $\mathcal{S} \times \mathcal{A}$ (relates to the source domain), the fixed ratio $\beta$ will be greater than 1.

It is significant to highlight that both corollaries above recover the intuition that the target dataset with no variation (or the number of samples in the target dataset is sufficiently large), i.e., $\varsigma \approx 0$, encourages

to consider the target dataset only in (10), i.e., $\lambda^* = 0$. On the other hand, when the two domains are close ($\xi \approx 0$), the optimal value of $\lambda$ is one, suggesting that one should use the source dataset solely. Although intuitive, the expected performance bounds in Theorems 1 and 2 are insufficient to claim any generalization guarantees as the tails of the distribution could be heavy. We address this concern in the next theorem by providing the generalization bound (worst-case performance bound). Moreover, since the stronger assumption in Theorem 2 is unlikely to hold in practice, we center on Theorem 1 from now on, upon which the remainder of this work is built, to maintain the generality of our results.

**Theorem 3** (Worst-Case Performance Bound). *Denote by $\beta'_u$ the upper bound of $P_{\mathcal{D}}(s,a)/P_{\mathcal{D}'}(s,a)$ for any $(s,a) \in \mathcal{S} \times \mathcal{A}$. Let the conditions of Theorem 1 and Assumption 1 hold. Given any dataset $\hat{\mathcal{D}}_{tr}$, the following bound holds at each iteration $(k = 0, 1, 2, \cdots)$ with probability at least $1 - \delta$*

$$\mathcal{E}_{\mathcal{D}}(Q_\lambda^{k+1}) - \mathcal{E}_{\mathcal{D}}(Q^{k+1})$$
$$\leq \left(\frac{1-\lambda}{1-\lambda+\lambda/\beta_u}\right)^2 \varsigma + \left(\frac{\lambda}{(1-\lambda)\beta_l+\lambda}\right)^2 \xi + e^{-NC}\xi$$
$$+ \sqrt{\frac{1}{2}\log\left(\frac{1}{\delta}\right)} \frac{|\mathcal{S}||\mathcal{A}|}{\sqrt{N}} \left(\frac{\beta'_u}{(1-\lambda)\beta_l+\lambda} \frac{2(1-\lambda)\gamma B}{1-\gamma}\right) \cdot \left(\frac{(1-\lambda)\beta_u(4B/(1-\gamma))+2\lambda\sqrt{\xi}}{(1-\lambda)\beta_l+\lambda}\right). \quad (19)$$

*Proof.* Refer to Appendix A.6. $\qquad\square$

The above theorem provides the worst-case bound of solving (10), which demonstrates the bias-variance trade-off by the two datasets with different weight $\lambda$ as well. Most importantly, both the expected and worst-case performance bounds, as shown in (16) and (19), imply that the optimal trade-off between the source and target datasets is not always trivial, indicating that $\lambda^*$ may not belong to $\{0, 0.5, 1\}$. The optimal trade-off for the expected performance depends on the number of samples in the target dataset (corresponding to $\varsigma$ and $N$), the dynamics gap (or discrepancy) between the two domains (corresponding to $\xi$), and the bounds of $P_{\hat{\mathcal{D}}}(s,a)/P_{\mathcal{D}'}(s,a)$ (corresponding to $\beta_l$ and $\beta_u$). In addition, the optimal weight for the worst-case performance bound will depend on more factors such as the reward bound $B$, the discount factor $\gamma$, the size of the state and action spaces $|\mathcal{S}|$ and $|\mathcal{A}|$, and the bound of $P_{\mathcal{D}}(s,a)/P_{\mathcal{D}'}(s,a)$ (see Remark 1) $\beta'_u$, some of which might be highly challenging to estimate in practice. Therefore, the worst-case performance bound in our work is primarily of conceptual interest. In addition, we focus mainly on the theoretical analysis of our proposed framework that balances the limited target dataset and the large-but-biased source dataset. Developing a practical and efficient algorithm to learn an approximate optimal weight remains a promising direction for future research.

Having established various performance bounds of solving (10), we are in the stage of providing the convergence guarantee. We formalize it in the next theorem, which relies on the following two quantities: the maximum of the dynamics gap over all iterations and the maximum of the variance over all iterations, given any dataset $\hat{\mathcal{D}}_{tr}$ and its corresponding transition-excluded dataset $\hat{\mathcal{D}}$

$$\xi_{\max} = \sup_{k \in \mathbb{N}} \xi(Q^k), \quad \varsigma_{\max} = \sup_{k \in \mathbb{N}} \varsigma(Q^k). \quad (20)$$

**Theorem 4** (Convergence). *Let the conditions of Theorem 1 hold. Given any dataset $\hat{\mathcal{D}}_{tr}$, it holds at each iteration $(k = 0, 1, 2, \cdots)$ that*

$$\mathbb{E}_{\hat{s}' \sim P_{\mathcal{D}}(\hat{s}'|s,a)} \left[ \mathbb{E}_{(s,a) \sim \mathcal{D}}[||Q_\lambda^{k+1}(s,a) - Q^*(s,a)||_\infty] \right] \leq \quad (21)$$
$$\gamma^{k+1} \mathbb{E}_{(s,a) \sim \mathcal{D}}[||Q^0(s,a) - Q^*(s,a)||_\infty] + \frac{1-\gamma^{k+1}}{1-\gamma}\left(\frac{1-\lambda}{1-\lambda+\lambda/\beta_u}\sqrt{\varsigma_{max}} + \left(\frac{\lambda}{(1-\lambda)\beta_l+\lambda} + e^{-NC}\right)\sqrt{\xi_{max}}\right).$$

*Proof.* Refer to Appendix A.7. $\qquad\square$

The previous theorem implies that the solution $Q_\lambda^{k+1}$ of solving (10) is guaranteed in expectation to converge to a neighborhood of the optimal $Q$-function, i.e., $Q^*$ as $k \to \infty$. This neighborhood is presented as follows

$$\mathcal{C} = \frac{1}{1-\gamma} \left( \frac{1-\lambda}{1-\lambda+\lambda/\beta_u} \sqrt{\varsigma_{\max}} + \left( \frac{\lambda}{(1-\lambda)\beta_l + \lambda} + e^{-NC} \right) \sqrt{\xi_{\max}} \right). \tag{22}$$

Apart from the discount factor $\gamma$, the neighborhood $\mathcal{C}$ depends on the weight $\lambda$, the maximal dynamics gap $\xi_{\max}$, the maximal variance $\varsigma_{\max}$, and the bounds $\beta_l$ and $\beta_u$.

## 4 Numerical Experiments

Although this work primarily focuses on theoretical analyses, we present in this section a series of numerical experiments that demonstrate the performance of solving (10) under different weight $\lambda$ and validate the corresponding theoretical contributions in the previous section.

### 4.1 *Procgen* Experiments

#### 4.1.1 Environments

We consider the well-known offline *Procgen* benchmark (Mediratta et al., 2023), which is often used to assess the domain adaptation/generalization capabilities of offline RL. We select five games/environments from *Procgen*: *Caveflyer*, *Climber*, *Dodgeball*, *Maze*, *Miner*, whose descriptions are provided in Appendix A.8.

#### 4.1.2 Experimental Setup

Our implementations as well as the datasets that have been used in this work are based on (Mediratta et al., 2023). Instead of training on a single dataset, this work trains an offline RL agent on two different datasets from the source and target domains.

**Backbone algorithms.** Recall that our framework is algorithm-agnostic, implying that various SOTA RL algorithms can apply. In this work, we select CQL (Kumar et al., 2020) and IQL (Kostrikov et al., 2021) as representative algorithms due to their promising and robust performance across a variety of offline RL tasks.

**Datasets.** Note that *Procgen* employs procedural content generation to create adaptive levels upon episode reset. Each level corresponds to a specific seed (non-negative integer) and has distinct layouts (such as the amount and position of various entities, see e.g., Figure 13) (Mediratta et al., 2023). As a result, the same action taken in the same state can lead to different successor states depending on the level (e.g., being blocked by an entity in one level but not in another), yielding level-dependent transition dynamics. In each environment, we select the target domain to span levels $[100, 199]$, and consider three distinct source domains defined over the level ranges $[0, 99]$, $[25, 124]$ and $[50, 149]$, respectively. Recall that the target dataset is expected to contain significantly fewer samples than the source dataset, i.e., $N \ll N'$. Typically, $N'$ is considered to be ten times larger than $N$. Therefore, we consider three different sizes of target datasets from levels $[100, 199]$ with $N \in \{1000, 2500, 4000\}$, and set $N' = 40000$.

**Hyperparameters.** To ensure a fair comparison, we retain all hyperparameters consistent, e.g., batch size, learning rate and network size, and solely change the weight assigned to each dataset. Key hyperparameters for the datasets and algorithms are summarized in Table 5 (refer to Appendix A.9).

#### 4.1.3 Results

Recall that the worst-case performance bound in (19) can be overly conservative, particularly when the state and action spaces are large, as the bound scales with the dimensionality of the spaces. Therefore, this subsection focuses on the expected performance bounds (16) or (17) as well as its corresponding corollaries. It is crucial to note that the variance $\varsigma$ and the dynamics gap $\xi$ are challenging to measure or estimate precisely, as it requires access to the entire source and target domains. This is not feasible within the scope of our problem of interest. Nevertheless, one can still investigate how these factors influence the expected bounds, which provide insights into the expected performance of offline RL. Specifically, we examine the

impact of each of $\varsigma$, $\xi$ and $\lambda$ on the expected performance. To achieve this, we vary one of these factors at a time while keeping the other two parameters constant. We present our findings as follows.

**Impact of the trade-off between the source and target datasets ($\lambda$).** We consider seven discrete values of $\lambda$ from $\{0, 0.2, 0.4, 0.5, 0.6, 0.8, 1\}$, where $\lambda \in \{0, 1, 0.5\}$ represents the three trivial choices: considering the limited target dataset only, employing the large-but-biased source dataset solely, treating both datasets equally. Notably, the expected performance bounds and Corollary 2 reveal that the optimal weight may not be the three trivial choices. To validate this, in each of the *Procgen* environments, we consider a target dataset comprising 1000 samples from levels $[100, 199]$ and a source dataset with 40000 samples from levels $[0, 99]$. Figure 2 depicts the results of two offline RL algorithms under various $\lambda$: CQL (upper row) and IQL (lower row). Indeed, observe that only two out of ten environments have the optimal weight to be the trivial choice, i.e., $\lambda^* = 0.5$ in *Dodgeball* for both CQL and IQL. This further underscores the importance of striking a proper trade-off between the two datasets, and reveals that trivial balancing strategies, e.g., $\lambda \in \{0, 0.5, 1\}$ are not consistently effective and can, sometimes, be catastrophic (see e.g., $\lambda = 1.0$ in *Miner*).

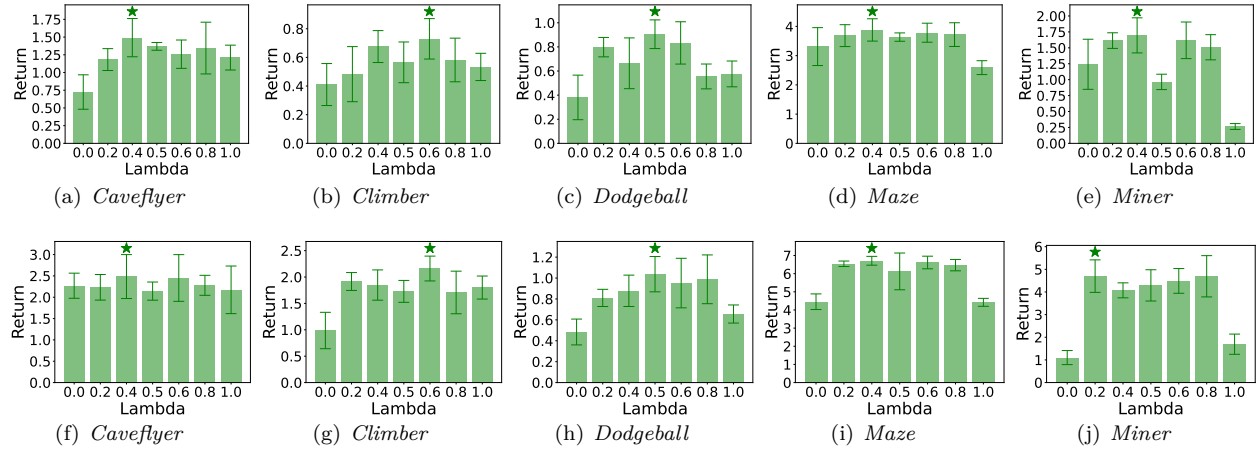

Figure 2: The performance of offline RL across five *Procgen* games. The source dataset contains 40000 samples from levels $[0, 99]$, while the target dataset comprises 1000 samples from levels $[100, 199]$. We consider seven weights, $\lambda \in \{0, 0.2, 0.4, 0.5, 0.6, 0.8, 1.0\}$, to trade off the source and target datasets with the star marking the optimal weight. Upper row: CQL as the backbone; Lower row: IQL as the backbone.

Table 1: The performance across all $\lambda \in \{0, 0.2, 0.4, 0.5, 0.6, 0.8, 1.0\}$ (mean and std) with fixed $N = 1000$ corresponding to different $\xi$. Left: CQL as the backbone algorithm; Right: IQL as the backbone algorithm.

| Game | $\xi_1$ | $\xi_2$ | $\xi_3$ |
|---|---|---|---|
| *Caveflyer* | $1.23 \pm 0.23$ | $1.28 \pm 0.26$ | $\mathbf{1.37 \pm 0.34}$ |
| *Climber* | $0.57 \pm 0.10$ | $0.73 \pm 0.16$ | $\mathbf{0.85 \pm 0.21}$ |
| *Dodgeball* | $0.67 \pm 0.17$ | $0.87 \pm 0.21$ | $\mathbf{1.29 \pm 0.42}$ |
| *Maze* | $3.52 \pm 0.41$ | $3.97 \pm 0.42$ | $\mathbf{4.35 \pm 0.52}$ |
| *Miner* | $1.27 \pm 0.48$ | $2.05 \pm 0.55$ | $\mathbf{3.12 \pm 0.80}$ |

| Game | $\xi_1$ | $\xi_2$ | $\xi_3$ |
|---|---|---|---|
| *Caveflyer* | $2.29 \pm 0.13$ | $2.30 \pm 0.13$ | $\mathbf{2.56 \pm 0.13}$ |
| *Climber* | $1.74 \pm 0.36$ | $2.14 \pm 0.63$ | $\mathbf{2.20 \pm 0.60}$ |
| *Dodgeball* | $0.83 \pm 0.20$ | $0.97 \pm 0.15$ | $\mathbf{1.39 \pm 0.47}$ |
| *Maze* | $5.91 \pm 1.02$ | $6.47 \pm 1.29$ | $\mathbf{7.33 \pm 1.37}$ |
| *Miner* | $3.58 \pm 1.51$ | $5.31 \pm 2.01$ | $\mathbf{7.24 \pm 2.81}$ |

**Impact of the dynamics gap between the source and target domains ($\xi$).** We consider three source datasets of the same size but from different domains: levels $[0, 99]$, levels $[25, 124]$, $[50, 149]$, and let $\xi_1, \xi_2$ and $\xi_3$ represent the dynamics gap between levels $[0, 99]$ and $[100, 199]$, between levels $[25, 124]$ and $[100, 199]$, and between levels $[50, 149]$ and $[100, 199]$, respectively. Thus, we obtain $\xi_1 \geq \xi_2 \geq \xi_3$, as more overlap between the levels of the two datasets demonstrates smaller discrepancies between them. Note that the bound in (16) or (17) decreases with smaller values of $\xi$, implying an improved expected performance of offline RL. Our

Table 2: The performance across all $\lambda \in \{0, 0.2, 0.4, 0.5, 0.6, 0.8, 1.0\}$ (mean and std) with fixed $\xi_3$ corresponding to different $N$. Left: CQL as the backbone algorithm; Right: IQL as the backbone algorithm.

| Game | $N = 1000$ | $N = 2500$ | $N = 4000$ | Game | $N = 1000$ | $N = 2500$ | $N = 4000$ |
|------|-----------|-----------|-----------|------|-----------|-----------|-----------|
| *Caveflyer* | $1.37 \pm 0.34$ | $1.52 \pm 0.26$ | $\mathbf{1.57 \pm 0.19}$ | *Caveflyer* | $2.56 \pm 0.13$ | $2.82 \pm 0.30$ | $\mathbf{2.90 \pm 0.32}$ |
| *Climber* | $0.85 \pm 0.21$ | $0.89 \pm 0.29$ | $\mathbf{0.95 \pm 0.26}$ | *Climber* | $2.20 \pm 0.60$ | $1.98 \pm 0.65$ | $\mathbf{2.22 \pm 0.63}$ |
| *Dodgeball* | $1.29 \pm 0.42$ | $1.34 \pm 0.30$ | $\mathbf{1.42 \pm 0.18}$ | *Dodgeball* | $1.39 \pm 0.47$ | $1.40 \pm 0.29$ | $\mathbf{1.57 \pm 0.29}$ |
| *Maze* | $4.35 \pm 0.52$ | $4.75 \pm 0.59$ | $\mathbf{5.21 \pm 0.69}$ | *Maze* | $7.33 \pm 1.37$ | $7.73 \pm 1.30$ | $\mathbf{8.23 \pm 1.22}$ |
| *Miner* | $3.12 \pm 0.80$ | $3.28 \pm 0.59$ | $\mathbf{3.33 \pm 0.75}$ | *Miner* | $7.24 \pm 2.81$ | $8.10 \pm 2.53$ | $\mathbf{8.18 \pm 2.20}$ |

numerical results of five games across three different $\xi$ values are summarized in Table 1, which supports the implication from the bound in (16) or (17).

**Impact of the size of the target dataset ($\varsigma$).** It is worth highlighting that the normalized variance $\varsigma$ in (15) decreases as $N(s, a)$ increases. Given the positive proportional relationship between $N$ and $N(s, a)$, a larger $N$ practically leads to a smaller $\varsigma$ (in expectation). Analogous to $\xi$, the bound in (16) or (17) decreases with smaller values of $\varsigma$ and/or $e^{-NC}$, thus indicating an enhanced expected performance of offline RL with larger $N$ (smaller $\varsigma$ and/or smaller $e^{-NC}$). Our numerical results of five games across three different values of $N$ are summarized in Table 2 and Figure 4, which validate the implication from the bound in (16) or (17). Nevertheless, offline RL may fail due to the coverage gaps rather than an insufficient sample size when the random sampling is non-uniform. For instance, having a large amount of data concentrated in a small portion of the target domain can lead to catastrophic performance, due to the limited generalization capability of RL (Packer et al., 2018; Kirk et al., 2023).

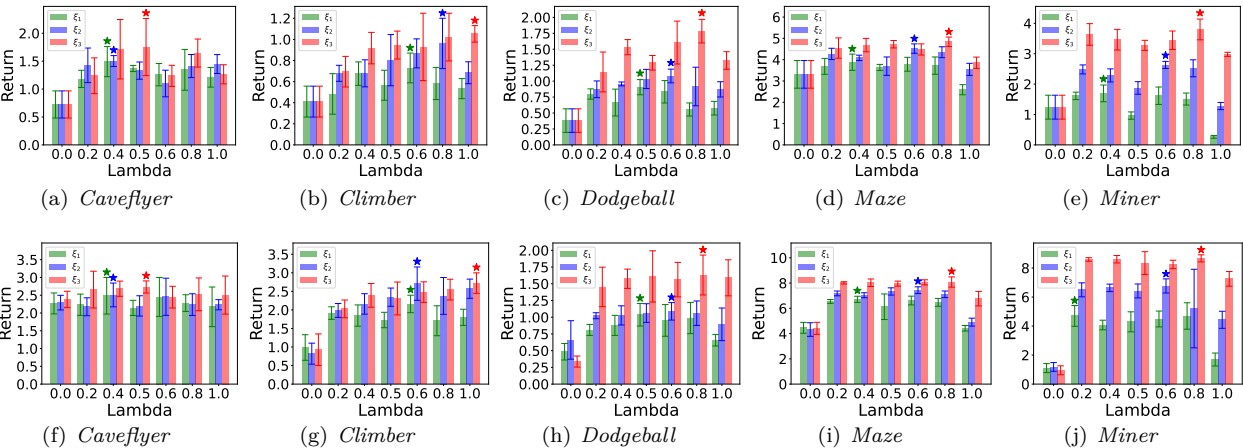

Figure 3: The performance of offline RL across five *Procgen* games. The target dataset comprises 1000 samples from levels $[100, 199]$, and three source datasets are considered, each containing 40000 samples from levels $[0, 99]$ (green, $\xi_1$), $[25, 124]$ (blue, $\xi_2$), and $[50, 149]$ (red, $\xi_3$), respectively. Seven weights, $\lambda \in \{0, 0.2, 0.4, 0.5, 0.6, 0.8, 1.0\}$, are evaluated to trade off the source and target datasets with the star marking the optimal weight for each $\xi$. Upper row: CQL as the backbone; Lower row: IQL as the backbone.

**Practical optimal trade-off between the source and target datasets ($\lambda^*$).** Notice that Theorems 1 and 2 as well as Corollaries 1 and 2 explicitly demonstrate that a smaller $\xi$ drives $\lambda^*$ closer to 1, while a smaller $\varsigma$ shifts $\lambda^*$ closer to 0. In what follows, we substantiate this implication through empirical evidence. *i*) To explore how $\lambda^*$ varies with $\xi$, we fix $N = 1000$ in each of the five *Procgen* games and select three different source datasets with $\xi_1 \geq \xi_2 \geq \xi_3$. We then implement this using seven values of $\lambda \in \{0, 0.2, 0.4, 0.5, 0.6, 0.8, 1\}$. Our numerical results in Figure 3 and Table 3 demonstrate that the optimal weight $\lambda^*$ within $\{0, 0.2, 0.4, 0.5, 0.6, 0.8, 1\}$ increases (closer to 1) as $\xi$ decreases. This substantiates

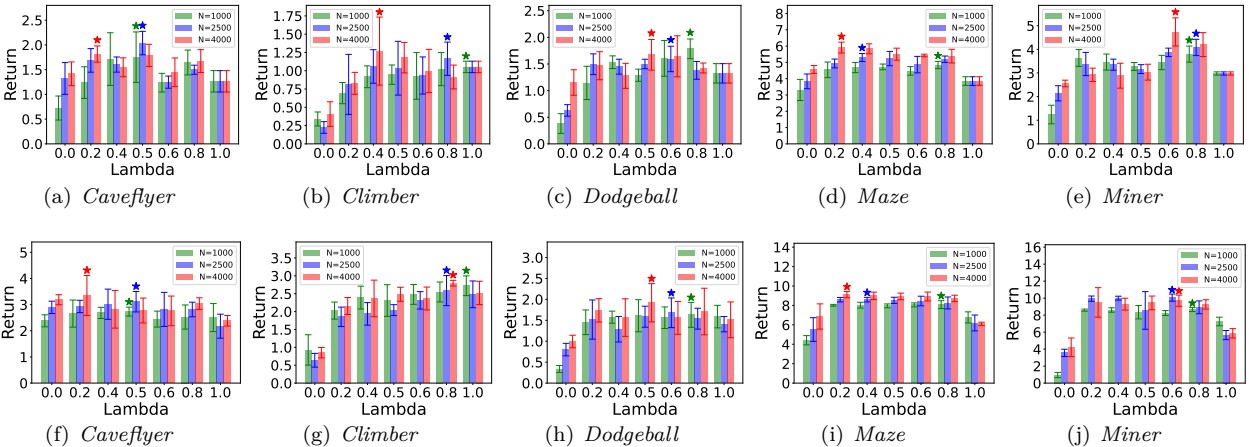

Figure 4: The performance of offline RL across five *Procgen* games. The source dataset comprises 40000 samples from levels $[50, 149]$, and target datasets from levels $[100, 199]$ are considered with three different sample sizes: $N = 1000$ (green), $N = 2500$ (blue), and $N = 4000$ (red). Seven weights, $\lambda \in \{0, 0.2, 0.4, 0.5, 0.6, 0.8, 1.0\}$, are evaluated to trade off the source and target datasets with the star marking the optimal weight for each $N$. Upper row: CQL as the backbone algorithm; Lower row: IQL as the backbone algorithm.

Corollary 2 and the intuition that greater emphasis should be placed on the source dataset when its discrepancy from the target domain is smaller. In the extreme case, where the source and target domains become identical, one should consider the large source dataset only. *ii*) To explore how $\lambda^*$ varies with $\varsigma$, we fix the dynamics gap to be $\xi_3$ in each of the five *Procgen* games. Since a larger $N$ corresponds to a smaller $\varsigma$, we select three different values of $N \in \{1000, 2500, 4000\}$. It is worth noting that $N'$ is consistently maintained at least ten times larger than $N$, ensuring that the source dataset always comprises a significantly larger amount of samples than that of the target dataset. We then implement this using seven values of $\lambda \in \{0, 0.2, 0.4, 0.5, 0.6, 0.8, 1\}$. Our numerical results in Figure 4 and Table 4 demonstrate that the optimal weight within $\{0, 0.2, 0.4, 0.5, 0.6, 0.8, 1\}$ decreases (closer to 0) as $N$ increases ($\varsigma$ decreases). This validates Corollary 2 and the intuition that greater emphasis should be placed on the target dataset when it comprises a larger number of samples. In the extreme case where the target dataset contains infinite samples, it becomes optimal to rely exclusively on the target dataset.

Table 3: The optimal weight $\lambda^*$ within $\{0, 0.2, 0.4, 0.5, 0.6, 0.8, 1\}$ corresponding to different $\xi$. Left: CQL as the backbone algorithm; Right: IQL as the backbone algorithm.

| Game | $\xi_1([0, 99])$ | $\xi_2([25, 124])$ | $\xi_3([50, 149])$ | Game | $\xi_1([0, 99])$ | $\xi_2([25, 124])$ | $\xi_3([50, 149])$ |
|---|---|---|---|---|---|---|---|
| *Caveflyer* | 0.4 | 0.4 | 0.5 | *Caveflyer* | 0.4 | 0.4 | 0.5 |
| *Climber* | 0.6 | 0.8 | 1.0 | *Climber* | 0.6 | 0.6 | 1.0 |
| *Dodgeball* | 0.5 | 0.6 | 0.8 | *Dodgeball* | 0.5 | 0.6 | 0.8 |
| *Maze* | 0.4 | 0.6 | 0.8 | *Maze* | 0.4 | 0.6 | 0.8 |
| *Miner* | 0.4 | 0.6 | 0.8 | *Miner* | 0.2 | 0.6 | 0.8 |

## 4.2 Auxiliary *MuJoCo* Experiments

In addition to the *Procgen* benchmark, we conduct auxiliary experiments on two continuous-control tasks from the *MuJoCo* benchmark: *HalfCheetah* and *HumanoidStandup*. This study is intended to further substantiate our theoretical contributions in this work.

Table 4: The optimal weight $\lambda^*$ within $\{0, 0.2, 0.4, 0.5, 0.6, 0.8, 1\}$ corresponding to different $N$. Left: CQL as the backbone algorithm; Right: IQL as the backbone algorithm.

| Game | $N = 1000$ | $N = 2500$ | $N = 4000$ |
|---|---|---|---|
| *Caveflyer* | 0.5 | 0.5 | 0.2 |
| *Climber* | 1.0 | 0.8 | 0.4 |
| *Dodgeball* | 0.8 | 0.6 | 0.5 |
| *Maze* | 0.8 | 0.4 | 0.2 |
| *Miner* | 0.8 | 0.8 | 0.6 |

| Game | $N = 1000$ | $N = 2500$ | $N = 4000$ |
|---|---|---|---|
| *Caveflyer* | 0.5 | 0.5 | 0.2 |
| *Climber* | 1.0 | 0.8 | 0.8 |
| *Dodgeball* | 0.8 | 0.6 | 0.5 |
| *Maze* | 0.8 | 0.4 | 0.2 |
| *Miner* | 0.8 | 0.6 | 0.6 |

**Environments.** For each task, the target domain corresponds to the standard environment. We construct three source domains $(\xi_1, \xi_2, \xi_3)$ by scaling a subset of physics parameters: gravity and friction scales. With the target domain defined by $(\texttt{gravity\_scale}, \texttt{friction\_scale}) = (1.0, 1.0)$, the three source domains are specified by $(\texttt{gravity\_scale}, \texttt{friction\_scale}) \in \{(0.85, 0.85), (0.90, 0.90), (0.95, 0.95)\}$ for $\{\xi_1, \xi_2, \xi_3\}$, respectively. In this setting, $\xi_3$ exhibits the smallest dynamics gap relative to the target domain, whereas $\xi_1$ induces the largest.

**Experimental Setup.** We select IQL as the backbone algorithm and sweep $\lambda \in \{0, 0.2, 0.4, 0.5, 0.6, 0.8, 1.0\}$. We collect the limited target dataset with $N$ samples and the large source datasets with $N' \gg N$ in each of the source domains. In *HalfCheetah*, we choose $N' = 80000$ while $N$ can vary in $\{3000, 5000, 8000\}$. In *HumanoidStandup*, we select $N' = 100000$ while $N$ can vary in $\{6000, 8000, 10000\}$. Note that different *MuJoCo* tasks may require distinct amounts of source and target data, depending on the task complexity.

**Results.** Figure 5 demonstrates that the optimal weight $\lambda^\star$ in the *MuJoCo* experiments does not coincide with any of the three trivial choices $\{0, 1, 0.5\}$, further highlighting the importance of striking an appropriate trade-off between the two datasets. Figure 6 reveals that the optimal weight $\lambda^\star$ shifts towards larger values as the dynamics gap between the source and target domains decreases, and Figure 7 exhibits that increasing $N$ shifts the optimal weight $\lambda^\star$ toward smaller values. These observations are consistent with those in the *Procgen* experiments, continuing to validate our theoretical findings in this paper.

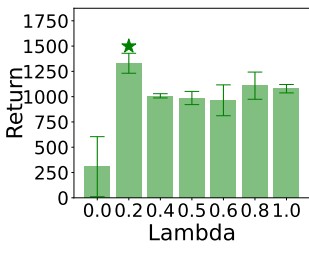

(a) *HalfCheetah*

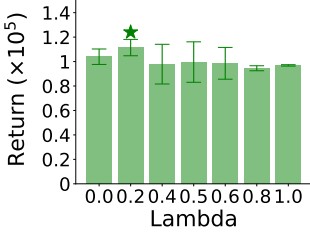

(b) *HumanoidStandup*

Figure 5: The performance of offline RL across two *MuJoCo* tasks. The source dataset contains $N'$ samples from the farthest source domain $\xi_1$, while the target dataset comprises $N$ samples from the target domain (*HalfCheetah*: $N = 3000$, $N' = 80000$; *HumanoidStandup*: $N = 6000$, $N' = 100000$). We consider seven weights, $\lambda \in \{0, 0.2, 0.4, 0.5, 0.6, 0.8, 1.0\}$, to trade off the source and target datasets with the star marking the optimal weight.

## 5 Conclusion

The performance of offline RL is highly dependent on the size of the target dataset. Even state-of-the-art offline RL algorithms often lack performance guarantees under a limited number of samples. To tackle offline RL with limited samples, domain adaptation can be employed, which considers related source datasets,

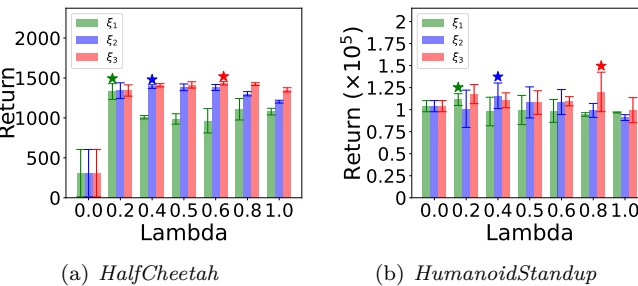

Figure 6: The performance of offline RL across two *MuJoCo* tasks. The target dataset comprises $N$ samples from the target domain, and three source domains are considered, each containing $N'$ samples with $(\texttt{gravity\_scale}, \texttt{friction\_scale}) \in \{(0.85, 0.85), (0.90, 0.90), (0.95, 0.95)\}$ (green–$\xi_1$; blue–$\xi_2$; red–$\xi_3$) (*HalfCheetah*: $N = 3000$, $N' = 80000$; *HumanoidStandup*: $N = 6000$, $N' = 100000$). Seven weights, $\lambda \in \{0, 0.2, 0.4, 0.5, 0.6, 0.8, 1.0\}$, are evaluated to trade off the source and target datasets with the star marking the optimal weight for each $\xi$.

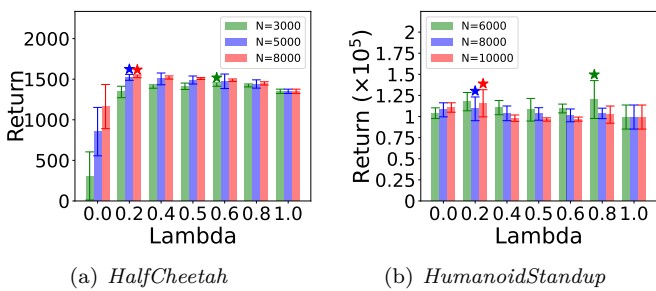

Figure 7: The performance of offline RL across two *MuJoCo* tasks. The source domain is fixed to the closest setting $\xi_3$, and target datasets from the target domain are considered with three different sample sizes: *HalfCheetah* $N \in \{3000, 5000, 8000\}$ (green, blue, red) with $N' = 80000$, and *HumanoidStandup* $N \in \{6000, 8000, 10000\}$ (green, blue, red) with $N' = 100000$. Seven weights, $\lambda \in \{0, 0.2, 0.4, 0.5, 0.6, 0.8, 1.0\}$, are evaluated to trade off source and target datasets with the star marking the optimal weight for each $N$.

e.g., simulators that typically offer unlimited (or a sufficiently large number of) samples. To the best of our knowledge, we propose in this work the first framework that theoretically explores the domain adaptation for offline RL with limited samples. Specifically, we establish the expected and worst-case performance bounds, as well as a convergence neighborhood under our framework. Moreover, this work provides the optimal weight for trading off the unlimited source dataset and the limited target dataset. It demonstrates that the optimal weight is not necessarily one of the trivial choices: using either dataset solely or combining the two datasets equally. Although this work centers on the theoretical analyses of our framework, we conduct a series of numerical experiments on the renowned *Procgen* and *MuJoCo* benchmarks, which substantiate our theoretical contributions. Last but not least, our established optimal weight is unlikely to be computed in practice, as it depends on quantities that are challenging to estimate. Hence, developing a practical and efficient algorithm to learn an approximate optimal weight remains a promising direction for future research.

## Acknowledgments

This work was sponsored by the Office of Naval Research (ONR), under contract number N00014-23-1-2377.

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

# A  Omitted Proofs

## A.1  Proof of Proposition 1

**Proposition 1.** Let Assumptions 2 and 3 hold. Recall the empirical Bellman operator $\hat{\mathcal{B}}$ in (4). Denote by $\mathcal{B}_{\mathcal{D}}$ ($\mathcal{B}_{\mathcal{D}'}$) the Bellman operator in (2) or (3) in which $s'$ follows the transition probability of the domain $\mathcal{D}$ ($\mathcal{D}'$). Note that $Q^{k+1}$ and $Q_{\lambda}^{k+1}$ represent the solutions to (6) and (10). Given any dataset $\hat{\mathcal{D}}_{\text{tr}}$ and its corresponding transition-excluded dataset $\hat{\mathcal{D}}$, denote by $N$ and $N(s,a)$ the total number of samples and the amount of $(s,a,\cdot)$ transition in $\hat{\mathcal{D}}_{\text{tr}}$. At each iteration ($k = 0, 1, 2, \cdots$), it holds that

$$Q^{k+1}(s,a) = \mathcal{B}_{\mathcal{D}}Q^k(s,a), \; \forall(s,a) \in \mathcal{S} \times \mathcal{A}, \tag{23}$$

$$Q_{\lambda}^{k+1}(s,a) = \frac{\frac{1-\lambda}{N}\sum_{j=1}^{N(s,a)}\hat{\mathcal{B}}_{\hat{s}'_j}Q^k(s,a) + \lambda P_{\mathcal{D}'}(s,a)\mathcal{B}_{\mathcal{D}'}Q^k(s,a)}{(1-\lambda)P_{\hat{\mathcal{D}}}(s,a) + \lambda P_{\mathcal{D}'}(s,a)}, \; \forall(s,a) \in \mathcal{S} \times \mathcal{A}. \tag{24}$$

*Proof.* Note that (12) is well-known in RL, however, we provide its proof here for completeness. For any $Q$, we note that

$$\mathcal{E}_{\mathcal{D}}(Q) \stackrel{(a)}{=} \mathbb{E}_{(s,a,s')\sim\mathcal{D}}\left[\left(Q(s,a) - \hat{\mathcal{B}}_{s'}Q^k(s,a)\right)^2\right] \tag{25}$$

$$\stackrel{(b)}{=} \sum_{s,a} P_D(s,a) \sum_{s'\sim\mathcal{D}} P_{\mathcal{D}}(s' \mid s,a)\left(Q(s,a) - \hat{\mathcal{B}}_{s'}Q^k(s,a)\right)^2, \tag{26}$$

where $(a)$ holds by definitions (4) and (5), $(b)$ follows from the definition of conditional expectation.

The derivative of $\mathcal{E}_{\mathcal{D}}(Q)$ w.r.t. $Q(s,a), \forall(s,a) \in \mathcal{S} \times \mathcal{A}$ is given by

$$\frac{\partial\mathcal{E}_{\mathcal{D}}(Q)}{\partial Q(s,a)} \stackrel{(a)}{=} 2P_{\mathcal{D}}(s,a)\sum_{s'} P_{\mathcal{D}}(s'|s,a)\left(Q(s,a) - \hat{\mathcal{B}}_{s'}Q^k(s,a)\right) \tag{27}$$

$$\stackrel{(b)}{=} 2P_{\mathcal{D}}(s,a)\left(Q(s,a) - \mathcal{B}_{\mathcal{D}}Q^k(s,a)\right), \tag{28}$$

where $(a)$ follows by taking the derivative, leveraging the fact that each $Q(s,a)$ is only present in exactly one term of the summation (28), $(b)$ follows from the facts that $Q(s,a)$ is independent of $s'$ and by definition $\mathcal{B}_{\mathcal{D}}Q^k(s,a) = \mathbb{E}_{s'\sim P_{\mathcal{D}}(s'|s,a)}\left[\hat{\mathcal{B}}_{s'}Q^k(s,a)\right]$ (see (2) or (3)). Since the objective (6) is strongly convex, its minimizer $Q^{k+1}$ is the unique point that satisfies

$$\frac{\partial\mathcal{E}_{\mathcal{D}}(Q^{k+1})}{\partial Q(s,a)} = 0, \; \forall(s,a) \in \mathcal{S} \times \mathcal{A}. \tag{29}$$

Combining the previous equation with (28) and the fact that $P_{\mathcal{D}}(s,a) > 0$ (by Assumption 3) yields

$$Q^{k+1}(s,a) = \mathcal{B}_{\mathcal{D}}Q^k(s,a), \; \forall(s,a) \in \mathcal{S} \times \mathcal{A}. \tag{30}$$

This completes the proof of (12) in Proposition 1.

We now turn our attention to proving (13). Note that

$$(1-\lambda)\mathcal{E}_{\hat{\mathcal{D}}}(Q) + \lambda\mathcal{E}_{\mathcal{D}'}(Q) \stackrel{(a)}{=} (1-\lambda)\frac{1}{N}\sum_{i=1}^{N}\left(Q(s_i,a_i) - \hat{\mathcal{B}}_{\hat{s}'}Q^k(s_i,a_i)\right)^2 + \lambda\mathcal{E}_{\mathcal{D}'}(Q) \tag{31}$$

$$\stackrel{(b)}{=} (1-\lambda)\frac{1}{N}\sum_{(s,a)\in\hat{\mathcal{D}}}\sum_{j=1}^{N(s,a)}\left(Q(s,a) - \hat{\mathcal{B}}_{\hat{s}'_j}Q^k(s,a)\right)^2 + \lambda\mathcal{E}_{\mathcal{D}'}(Q), \tag{32}$$

where $(a)$ follows from the definitions in (4) and (7), $(b)$ re-arranges (31) by summing first the $N(s,a)$ samples of the next state $\hat{s}'$ corresponding to a given $(s,a)$ pair in the dataset $\hat{\mathcal{D}}$.

The derivative of $(1-\lambda)\mathcal{E}_{\hat{\mathcal{D}}}(Q) + \lambda\mathcal{E}_{\mathcal{D}'}(Q)$ w.r.t. $Q(s,a)$ for any $(s,a) \in \mathcal{S} \times \mathcal{A}$ can be computed under two scenarios:

(i) For any $(s, a) \in \mathcal{S} \times \mathcal{A} \setminus \hat{\mathcal{D}}$, the derivative is given by

$$\frac{\partial \left((1 - \lambda)\mathcal{E}_{\hat{\mathcal{D}}}(Q) + \lambda\mathcal{E}_{\mathcal{D}'}(Q)\right)}{\partial Q(s, a)} = 2\lambda P_{\mathcal{D}'}(s, a)\left(Q(s, a) - \mathcal{B}_{\mathcal{D}'}Q^k(s, a)\right), \tag{33}$$

where the previous equation follows from an analog of (28) replacing $\mathcal{D}$ by $\mathcal{D}'$ and the fact that the first term in (32) depends solely on the state-action pairs in the dataset $\hat{\mathcal{D}}$.

(ii) For any $(s, a) \in \hat{\mathcal{D}}$, the derivative is given by

$$\frac{\partial \left((1 - \lambda)\mathcal{E}_{\hat{\mathcal{D}}}(Q) + \lambda\mathcal{E}_{\mathcal{D}'}(Q)\right)}{\partial Q(s, a)}$$

$$\overset{(a)}{=} (1 - \lambda)\frac{2}{N} \sum_{j=1}^{N(s,a)} \left(Q(s, a) - \hat{\mathcal{B}}_{\hat{s}'_j}Q^k(s, a)\right) + 2\lambda P_{\mathcal{D}'}(s, a)\left(Q(s, a) - \mathcal{B}_{\mathcal{D}'}Q^k(s, a)\right) \tag{34}$$

$$\overset{(b)}{=} (1 - \lambda)\frac{2}{N} \left(N(s, a)Q(s, a) - \sum_{j=1}^{N(s,a)} \hat{\mathcal{B}}_{\hat{s}'_j}Q^k(s, a)\right) + 2\lambda P_{\mathcal{D}'}(s, a)\left(Q(s, a) - \mathcal{B}_{\mathcal{D}'}Q^k(s, a)\right), \tag{35}$$

where the first term in $(a)$ is obtained by taking the derivative, leveraging the fact that each $Q(s, a)$ is only present in exactly one term of the summation (35), and the second term in $(a)$ is analogous to the derivation of $\partial\mathcal{E}_{\mathcal{D}}(Q)/\partial Q(s, a)$ (see (28)) replacing $\mathcal{D}$ by $\mathcal{D}'$, (b) is due to the fact that $\sum_{j=1}^{N(s,a)} Q(s, a) = N(s, a)Q(s, a)$.

It is worth highlighting that (35) reduces to (33) when $N(s, a) = 0$, i.e., when $(s, a) \in \mathcal{S} \times \mathcal{A} \setminus \hat{\mathcal{D}}$ (see Remark 1). Consequently, by combining the two scenarios of $(s, a) \in \hat{\mathcal{D}}$ and $(s, a) \in \mathcal{S} \times \mathcal{A} \setminus \hat{\mathcal{D}}$, it holds for any $(s, a) \in \mathcal{S} \times \mathcal{A}$ that

$$\frac{\partial \left((1 - \lambda)\mathcal{E}_{\hat{\mathcal{D}}}(Q) + \lambda\mathcal{E}_{\mathcal{D}'}(Q)\right)}{\partial Q(s, a)}$$

$$= (1 - \lambda)\frac{2}{N} \left(N(s, a)Q(s, a) - \sum_{j=1}^{N(s,a)} \hat{\mathcal{B}}_{\hat{s}'_j}Q^k(s, a)\right) + 2\lambda P_{\mathcal{D}'}(s, a)\left(Q(s, a) - \mathcal{B}_{\mathcal{D}'}Q^k(s, a)\right). \tag{36}$$

Similar to (29), since $Q_\lambda^{k+1}$ is the unique solution to the convex problem (10), the gradient of the objective is zero. Equating the right hand side of the previous equation to zero follows that

$$Q_\lambda^{k+1}(s, a)\left((1 - \lambda)P_{\hat{\mathcal{D}}}(s, a) + \lambda P_{\mathcal{D}'}(s, a)\right) = \frac{1 - \lambda}{N} \sum_{j=1}^{N(s,a)} \hat{\mathcal{B}}_{\hat{s}'_j}Q^k(s, a) + \lambda P_{\mathcal{D}'}(s, a)\mathcal{B}_{\mathcal{D}'}Q^k(s, a), \tag{37}$$

where in the above expression we use $P_{\hat{\mathcal{D}}}(s, a) = N(s, a)/N$. It holds by Remark 2 that $(1 - \lambda)P_{\hat{\mathcal{D}}}(s, a) + \lambda P_{\mathcal{D}'}(s, a) > 0$. Then we further obtain

$$Q_\lambda^{k+1}(s, a) = \frac{\frac{1-\lambda}{N} \sum_{j=1}^{N(s,a)} \hat{\mathcal{B}}_{\hat{s}'_j}Q^k(s, a) + \lambda P_{\mathcal{D}'}(s, a)\mathcal{B}_{\mathcal{D}'}Q^k(s, a)}{(1 - \lambda)P_{\hat{\mathcal{D}}}(s, a) + \lambda P_{\mathcal{D}'}(s, a)}, \quad \forall (s, a) \in \mathcal{S} \times \mathcal{A}. \tag{38}$$

This completes the proof of Proposition 1. □

## A.2 Proof of Theorem 1

**Theorem 1** (Expected Performance Bound). Recall $\xi$ in (14), $\varsigma$ in (15) and define $C := \min_{(s,a) \in \mathcal{S} \times \mathcal{A}} P_{\mathcal{D}}(s, a)$. Let the conditions of Proposition 1 hold. Given any dataset $\hat{\mathcal{D}}_{\text{tr}}$, it holds at each iteration $(k = 0, 1, 2, \cdots)$ that

$$\mathbb{E}_{\hat{s}' \sim P_{\mathcal{D}}(\hat{s}'|s,a)} \left[\mathcal{E}_{\mathcal{D}}(Q_\lambda^{k+1})\right] - \mathcal{E}_{\mathcal{D}}(Q^{k+1}) \leq \left(\frac{1 - \lambda}{1 - \lambda + \lambda/\beta_u}\right)^2 \varsigma + \left(\left(\frac{\lambda}{(1 - \lambda)\beta_l + \lambda}\right)^2 + e^{-NC}\right)\xi. \tag{39}$$

*Proof.* We start by writing $\mathcal{E}_\mathcal{D}(Q_\lambda^{k+1}) - \mathcal{E}_D(Q^{k+1})$ using the definitions in (4) and (5).

$$\mathcal{E}_\mathcal{D}(Q_\lambda^{k+1}) - \mathcal{E}_D(Q^{k+1}) = \mathop{\mathbb{E}}_{(s,a,s')\sim\mathcal{D}}\left[(Q_\lambda^{k+1}(s,a) - \hat{\mathcal{B}}_{s'}Q^k(s,a))^2 - (Q^{k+1}(s,a) - \hat{\mathcal{B}}_{s'}Q^k(s,a))^2\right]. \quad (40)$$

Expanding the squares in the above expression follows that

$$\begin{aligned}&\mathcal{E}_\mathcal{D}(Q_\lambda^{k+1}) - \mathcal{E}_D(Q^{k+1})\\ &= \mathop{\mathbb{E}}_{(s,a,s')\sim\mathcal{D}}\left[(Q_\lambda^{k+1}(s,a))^2 - (Q^{k+1}(s,a))^2 + 2\hat{\mathcal{B}}_{s'}Q^k(s,a)\left(Q^{k+1}(s,a) - Q_\lambda^{k+1}(s,a)\right)\right].\end{aligned} \quad (41)$$

Conditioning on $(s,a)$, using the fact that only $\hat{B}_{s'}Q^k(s,a)$ depends on $s'$, and that by definition $\mathbb{E}_{s'\sim P_\mathcal{D}(s'|s,a)}\left[\hat{\mathcal{B}}_{s'}Q^k(s,a)\right] = \mathcal{B}_\mathcal{D}Q^k(s,a)$, the above equation reduces to

$$\begin{aligned}&\mathcal{E}_\mathcal{D}(Q_\lambda^{k+1}) - \mathcal{E}_D(Q^{k+1})\\ &= \mathop{\mathbb{E}}_{(s,a)\sim\mathcal{D}}\left[(Q_\lambda^{k+1}(s,a))^2 - (Q^{k+1}(s,a))^2 + 2\mathcal{B}_\mathcal{D}Q^k(s,a)\left(Q^{k+1}(s,a) - Q_\lambda^{k+1}(s,a)\right)\right].\end{aligned} \quad (42)$$

Replacing $Q^{k+1}(s,a)$ with $\mathcal{B}_\mathcal{D}Q^k(s,a)$ (by (12) in Proposition 1) in the previous equation yields

$$\begin{aligned}&\mathcal{E}_\mathcal{D}(Q_\lambda^{k+1}) - \mathcal{E}_D(Q^{k+1})\\ &= \mathop{\mathbb{E}}_{(s,a)\sim\mathcal{D}}\left[(Q_\lambda^{k+1}(s,a))^2 - \left(\mathcal{B}_\mathcal{D}Q^k(s,a)\right)^2 + 2\mathcal{B}_\mathcal{D}Q^k(s,a)\left(\mathcal{B}_\mathcal{D}Q^k(s,a) - Q_\lambda^{k+1}(s,a)\right)\right] \quad (43)\\ &= \mathop{\mathbb{E}}_{(s,a)\sim\mathcal{D}}\left[(Q_\lambda^{k+1}(s,a))^2 + \left(\mathcal{B}_\mathcal{D}Q^k(s,a)\right)^2 - 2\mathcal{B}_\mathcal{D}Q^k(s,a)Q_\lambda^{k+1}(s,a)\right]. \quad (44)\end{aligned}$$

We next take the expectation on (44) with respect to $\hat{s}'$ (from the dataset $\hat{\mathcal{D}}_{\mathrm{tr}}$), using the fact that only $Q_\lambda^{k+1}(s,a)$ depends on $\hat{s}'$ and the conditional expectation it follows that

$$\begin{aligned}&\mathop{\mathbb{E}}_{\hat{s}'\sim P_\mathcal{D}(\hat{s}'|s,a)}\left[\mathcal{E}_\mathcal{D}(Q_\lambda^{k+1})\right] - \mathcal{E}_D(Q^{k+1})\\ &= \mathop{\mathbb{E}}_{(s,a)\sim\mathcal{D}}\left[\mathop{\mathbb{E}}_{\hat{s}'\sim P_\mathcal{D}(\hat{s}'|s,a)}\left[\left(Q_\lambda^{k+1}(s,a)\right)^2\right] + \left(\mathcal{B}_\mathcal{D}Q^k(s,a)\right)^2 - 2\mathcal{B}_\mathcal{D}Q^k(s,a)\left(\mathop{\mathbb{E}}_{\hat{s}'\sim P_\mathcal{D}(\hat{s}'|s,a)}\left[Q_\lambda^{k+1}(s,a)\right]\right)\right].\end{aligned} \quad (45)$$

Using the definition of the variance of a random variable, we rewrite the first term on the right hand side of the above expression

$$\mathop{\mathbb{E}}_{\hat{s}'\sim P_\mathcal{D}(\hat{s}'|s,a)}\left[\left(Q_\lambda^{k+1}(s,a)\right)^2\right] = \sigma_{\hat{s}'}^2\left(Q_\lambda^{k+1}(s,a)\right) + \left(\mathop{\mathbb{E}}_{\hat{s}'\sim P_\mathcal{D}(\hat{s}'|s,a)}\left[Q_\lambda^{k+1}(s,a)\right]\right)^2. \quad (46)$$

Substituting the previous expression in (45) yields

$$\begin{aligned}&\mathop{\mathbb{E}}_{\hat{s}'\sim P_\mathcal{D}(\hat{s}'|s,a)}\left[\mathcal{E}_\mathcal{D}(Q_\lambda^{k+1})\right] - \mathcal{E}_D(Q^{k+1})\\ &= \mathop{\mathbb{E}}_{(s,a)\sim\mathcal{D}}\left[\sigma_{\hat{s}'}^2\left(Q_\lambda^{k+1}(s,a)\right) + \left(\mathop{\mathbb{E}}_{\hat{s}'\sim P_\mathcal{D}(\hat{s}'|s,a)}\left[Q_\lambda^{k+1}(s,a)\right]\right)^2 + \left(\mathcal{B}_\mathcal{D}Q^k(s,a)\right)^2\right.\\ &\qquad\qquad\left. - 2\mathcal{B}_\mathcal{D}Q^k(s,a)\left(\mathop{\mathbb{E}}_{\hat{s}'\sim P_\mathcal{D}(\hat{s}'|s,a)}\left[Q_\lambda^{k+1}(s,a)\right]\right)\right].\end{aligned} \quad (47)$$

Note that the last three terms on the right hand side of the above expression are the square of a difference. Hence, the previous equation reduces to

$$\mathbb{E}_{\hat{s}'\sim P_{\mathcal{D}}(\hat{s}'|s,a)}\left[\mathcal{E}_{\mathcal{D}}(Q_\lambda^{k+1})\right] - \mathcal{E}_D(Q^{k+1})$$

$$= \mathbb{E}_{(s,a)\sim\mathcal{D}}\left[\sigma_{\hat{s}'}^2\left(Q_\lambda^{k+1}(s,a)\right) + \left(\underbrace{\mathbb{E}_{\hat{s}'\sim P_{\mathcal{D}}(\hat{s}'|s,a)}\left[Q_\lambda^{k+1}(s,a)\right] - \mathcal{B}_{\mathcal{D}}Q^k(s,a)}_{U_1}\right)^2\right]. \tag{48}$$

We next work on $\sigma_{\hat{s}'}^2\left(Q_\lambda^{k+1}(s,a)\right)$ and $U_1$ separately by focusing on $\sigma_{\hat{s}'}^2\left(Q_\lambda^{k+1}(s,a)\right)$ first.

(i) For any $(s,a) \in \mathcal{S}\times\mathcal{A}\setminus\hat{\mathcal{D}}$, $Q_\lambda^{k+1}(s,a)$ in (13) reduces to $\mathcal{B}_{\mathcal{D}'}Q^k(s,a)$, which is independent of $\hat{s}'$. Then we have

$$\sigma_{\hat{s}'}^2\left(Q_\lambda^{k+1}(s,a)\right) = \sigma_{\hat{s}'}^2\left(\mathcal{B}_{\mathcal{D}'}Q^k(s,a)\right) = 0. \tag{49}$$

(ii) For any $(s,a) \in \hat{\mathcal{D}}$, i.e., $N(s,a) > 0$, we note that

$$\sigma_{\hat{s}'}^2\left(Q_\lambda^{k+1}(s,a)\right) \overset{(a)}{=} \sigma_{\hat{s}'}^2\left(\frac{\frac{1-\lambda}{N}\sum_{j=1}^{N(s,a)}\hat{\mathcal{B}}_{\hat{s}_j'}Q^k(s,a) + \lambda P_{\mathcal{D}'}(s,a)\mathcal{B}_{\mathcal{D}'}Q^k(s,a)}{(1-\lambda)P_{\hat{\mathcal{D}}}(s,a) + \lambda P_{\mathcal{D}'}(s,a)}\right) \tag{50}$$

$$\overset{(b)}{=} \sigma_{\hat{s}'}^2\left(\frac{\frac{1-\lambda}{N}\sum_{j=1}^{N(s,a)}\hat{\mathcal{B}}_{\hat{s}_j'}Q^k(s,a)}{(1-\lambda)P_{\hat{\mathcal{D}}}(s,a) + \lambda P_{\mathcal{D}'}(s,a)}\right) \tag{51}$$

$$= \sigma_{\hat{s}'}^2\left(\frac{(1-\lambda)P_{\hat{\mathcal{D}}}(s,a)\frac{1}{N(s,a)}\sum_{j=1}^{N(s,a)}\hat{\mathcal{B}}_{\hat{s}_j'}Q^k(s,a)}{(1-\lambda)P_{\hat{\mathcal{D}}}(s,a) + \lambda P_{\mathcal{D}'}(s,a)}\right) \tag{52}$$

$$\overset{(c)}{=} \left(\frac{(1-\lambda)P_{\hat{\mathcal{D}}}(s,a)}{(1-\lambda)P_{\hat{\mathcal{D}}}(s,a) + \lambda P_{\mathcal{D}'}(s,a)}\right)^2 \frac{\sigma_{\hat{s}'}^2\left(\hat{\mathcal{B}}_{\hat{s}'}Q^k(s,a)\right)}{N(s,a)}, \tag{53}$$

where $(a)$ follows from (13) in Proposition 1, $(b)$ is obtained by the theorem of variance of a shifted random variable, since $\lambda P_{\mathcal{D}'}(s,a)\mathcal{B}_{\mathcal{D}'}Q^k(s,a)$ does not depend on $\hat{s}'$, $(c)$ is due to $\sigma^2(cX) = c^2\sigma^2(X)$ where $X$ denotes any random variable and $c$ is a constant.

Therefore, we have

$$\sigma_{\hat{s}'}^2\left(Q_\lambda^{k+1}(s,a)\right) = \begin{cases} 0, & \forall(s,a) \in \mathcal{S}\times\mathcal{A}\setminus\hat{\mathcal{D}}, \\ \left(\dfrac{(1-\lambda)P_{\hat{\mathcal{D}}}(s,a)}{(1-\lambda)P_{\hat{\mathcal{D}}}(s,a) + \lambda P_{\mathcal{D}'}(s,a)}\right)^2 \dfrac{\sigma_{\hat{s}'}^2\left(\hat{\mathcal{B}}_{\hat{s}'}Q^k(s,a)\right)}{N(s,a)}, & \forall(s,a) \in \hat{\mathcal{D}}. \end{cases} \tag{54}$$

Notice that (48) can be rewritten as

$$\mathbb{E}_{\hat{s}'\sim P_{\mathcal{D}}(\hat{s}'|s,a)}\left[\mathcal{E}_{\mathcal{D}}(Q_\lambda^{k+1})\right] - \mathcal{E}_D(Q^{k+1})$$

$$= \mathbb{E}_{(s,a)\sim\mathcal{D}}\left[\sigma_{\hat{s}'}^2\left(Q_\lambda^{k+1}(s,a)\right)\right] + \mathbb{E}_{(s,a)\sim\mathcal{D}}\left[U_1^2\right] \tag{55}$$

$$= \sum_{(s,a)\in\mathcal{S}\times\mathcal{A}} P_{\mathcal{D}}(s,a)\sigma_{\hat{s}'}^2\left(Q_\lambda^{k+1}(s,a)\right) + \mathbb{E}_{(s,a)\sim\mathcal{D}}\left[U_1^2\right] \tag{56}$$

$$= \sum_{(s,a)\in\hat{\mathcal{D}}} P_{\mathcal{D}}(s,a)\sigma_{\hat{s}'}^2\left(Q_\lambda^{k+1}(s,a)\right) + \sum_{(s,a)\in\mathcal{S}\times\mathcal{A}\setminus\hat{\mathcal{D}}} P_{\mathcal{D}}(s,a)\sigma_{\hat{s}'}^2\left(Q_\lambda^{k+1}(s,a)\right) + \mathbb{E}_{(s,a)\sim\mathcal{D}}\left[U_1^2\right] \tag{57}$$

Substituting (54) into the previous equation yields

$$\underset{\hat{s}'\sim P_{\mathcal{D}}(\hat{s}'|s,a)}{\mathbb{E}}\left[\mathcal{E}_{\mathcal{D}}(Q_\lambda^{k+1})\right] - \mathcal{E}_D(Q^{k+1})$$

$$= \sum_{(s,a)\in\hat{\mathcal{D}}} P_{\mathcal{D}}(s,a)\left(\frac{(1-\lambda)P_{\hat{\mathcal{D}}}(s,a)}{(1-\lambda)P_{\hat{\mathcal{D}}}(s,a)+\lambda P_{\mathcal{D}'}(s,a)}\right)^2 \frac{\sigma_{\hat{s}'}^2\left(\hat{\mathcal{B}}_{\hat{s}'}Q^k(s,a)\right)}{N(s,a)} + \underset{(s,a)\sim\mathcal{D}}{\mathbb{E}}\left[U_1^2\right] \qquad (58)$$

It then follows from the law of total probability that

$$\underset{\hat{s}'\sim P_{\mathcal{D}}(\hat{s}'|s,a)}{\mathbb{E}}\left[\mathcal{E}_{\mathcal{D}}(Q_\lambda^{k+1})\right] - \mathcal{E}_D(Q^{k+1})$$

$$= \sum_{(s,a)\in\hat{\mathcal{D}}}\left(P_{\mathcal{D}}(s,a\mid(s,a)\in\hat{\mathcal{D}})\cdot P((s,a)\in\hat{\mathcal{D}}) + P_{\mathcal{D}}(s,a\mid(s,a)\in\mathcal{S}\times\mathcal{A}\setminus\hat{\mathcal{D}})\cdot P((s,a)\in\mathcal{S}\times\mathcal{A}\setminus\hat{\mathcal{D}})\right)$$

$$\cdot\left(\frac{(1-\lambda)P_{\hat{\mathcal{D}}}(s,a)}{(1-\lambda)P_{\hat{\mathcal{D}}}(s,a)+\lambda P_{\mathcal{D}'}(s,a)}\right)^2 \frac{\sigma_{\hat{s}'}^2\left(\hat{\mathcal{B}}_{\hat{s}'}Q^k(s,a)\right)}{N(s,a)} + \underset{(s,a)\sim\mathcal{D}}{\mathbb{E}}\left[U_1^2\right] \qquad (59)$$

$$= \sum_{(s,a)\in\hat{\mathcal{D}}} P_{\mathcal{D}}(s,a\mid(s,a)\in\hat{\mathcal{D}})\cdot P((s,a)\in\hat{\mathcal{D}})\cdot\left(\frac{(1-\lambda)P_{\hat{\mathcal{D}}}(s,a)}{(1-\lambda)P_{\hat{\mathcal{D}}}(s,a)+\lambda P_{\mathcal{D}'}(s,a)}\right)^2 \frac{\sigma_{\hat{s}'}^2\left(\hat{\mathcal{B}}_{\hat{s}'}Q^k(s,a)\right)}{N(s,a)}$$

$$+ \underset{(s,a)\sim\mathcal{D}}{\mathbb{E}}\left[U_1^2\right] \qquad (60)$$

where the last equation holds by the fact that $P_{\mathcal{D}}(s,a\mid(s,a)\in\mathcal{S}\times\mathcal{A}\setminus\hat{\mathcal{D}})=0$ as all $(s,a)\in\hat{\mathcal{D}}$ now. It follows by the definition of conditional expectation and $P((s,a)\in\hat{\mathcal{D}})\leq 1$ that

$$\underset{\hat{s}'\sim P_{\mathcal{D}}(\hat{s}'|s,a)}{\mathbb{E}}\left[\mathcal{E}_{\mathcal{D}}(Q_\lambda^{k+1})\right] - \mathcal{E}_D(Q^{k+1})$$

$$\leq \underset{(s,a)\sim\mathcal{D}|\hat{\mathcal{D}}}{\mathbb{E}}\left[\left(\frac{(1-\lambda)P_{\hat{\mathcal{D}}}(s,a)}{(1-\lambda)P_{\hat{\mathcal{D}}}(s,a)+\lambda P_{\mathcal{D}'}(s,a)}\right)^2 \frac{\sigma_{\hat{s}'}^2\left(\hat{\mathcal{B}}_{\hat{s}'}Q^k(s,a)\right)}{N(s,a)}\right] + \underset{(s,a)\sim\mathcal{D}}{\mathbb{E}}\left[U_1^2\right]. \qquad (61)$$

We now turn our attention to $U_1$. For any $(s,a)\in\mathcal{S}\times\mathcal{A}$, we obtain

$$U_1 \overset{(a)}{=} \underset{\hat{s}'\sim P_{\mathcal{D}}(\hat{s}'|s,a)}{\mathbb{E}}\left[\frac{\frac{1-\lambda}{N}\sum_{j=1}^{N(s,a)}\hat{\mathcal{B}}_{\hat{s}'_j}Q^k(s,a)+\lambda P_{\mathcal{D}'}(s,a)\mathcal{B}_{\mathcal{D}'}Q^k(s,a)}{(1-\lambda)P_{\hat{\mathcal{D}}}(s,a)+\lambda P_{\mathcal{D}'}(s,a)}\right] - \mathcal{B}_{\mathcal{D}}Q^k(s,a) \qquad (62)$$

$$\overset{(b)}{=} \frac{\frac{1-\lambda}{N}N(s,a)\mathcal{B}_{\mathcal{D}}Q^k(s,a)+\lambda P_{\mathcal{D}'}(s,a)\mathcal{B}_{\mathcal{D}'}Q^k(s,a)}{(1-\lambda)P_{\hat{\mathcal{D}}}(s,a)+\lambda P_{\mathcal{D}'}(s,a)} - \mathcal{B}_{\mathcal{D}}Q^k(s,a) \qquad (63)$$

$$\overset{(c)}{=} \frac{(1-\lambda)P_{\hat{\mathcal{D}}}(s,a)\mathcal{B}_{\mathcal{D}}Q^k(s,a)+\lambda P_{\mathcal{D}'}(s,a)\mathcal{B}_{\mathcal{D}'}Q^k(s,a)}{(1-\lambda)P_{\hat{\mathcal{D}}}(s,a)+\lambda P_{\mathcal{D}'}(s,a)} - \mathcal{B}_{\mathcal{D}}Q^k(s,a) \qquad (64)$$

$$\overset{(d)}{=} \frac{\lambda P_{\mathcal{D}'}(s,a)\left(\mathcal{B}_{\mathcal{D}'}Q^k(s,a)-\mathcal{B}_{\mathcal{D}}Q^k(s,a)\right)}{(1-\lambda)P_{\hat{\mathcal{D}}}(s,a)+\lambda P_{\mathcal{D}'}(s,a)}, \qquad (65)$$

where $(a)$ follows from (13) in Proposition 1, $(b)$ follows by the definition of $\mathcal{B}_{\mathcal{D}}Q^k(s,a)$ (see (2) or (3)), $(c)$ follows by the definition $P_{\hat{\mathcal{D}}}(s,a)=N(s,a)/N$, and $(d)$ follows from combining and canceling terms.

Substituting the expression of $U_1$ in (65) into (61) results in

$$\underset{\hat{s}'\sim P_{\mathcal{D}}(\hat{s}'|s,a)}{\mathbb{E}}\left[\mathcal{E}_{\mathcal{D}}(Q_\lambda^{k+1})\right] - \mathcal{E}_D(Q^{k+1}) \leq \underset{(s,a)\sim\mathcal{D}|\hat{\mathcal{D}}}{\mathbb{E}}\left[\left(\frac{(1-\lambda)P_{\hat{\mathcal{D}}}(s,a)}{(1-\lambda)P_{\hat{\mathcal{D}}}(s,a)+\lambda P_{\mathcal{D}'}(s,a)}\right)^2 \frac{\sigma_{\hat{s}'}^2\left(\hat{\mathcal{B}}_{\hat{s}'}Q^k(s,a)\right)}{N(s,a)}\right] +$$

$$\underset{(s,a)\sim\mathcal{D}}{\mathbb{E}}\left[\left(\frac{\lambda P_{\mathcal{D}'}(s,a)}{(1-\lambda)P_{\hat{\mathcal{D}}}(s,a)+\lambda P_{\mathcal{D}'}(s,a)}\right)^2 \left(\mathcal{B}_{\mathcal{D}}Q^k(s,a)-\mathcal{B}_{\mathcal{D}'}Q^k(s,a)\right)^2\right]. \qquad (66)$$

The previous inequality can be simplified as

$$
\mathbb{E}_{\hat{s}' \sim P_{\mathcal{D}}(\hat{s}'|s,a)} \left[ \mathcal{E}_{\mathcal{D}}(Q_\lambda^{k+1}) \right] - \mathcal{E}_{\mathcal{D}}(Q^{k+1})
$$

$$
\leq \mathbb{E}_{(s,a) \sim \mathcal{D}|\hat{\mathcal{D}}} \left[ \left( \frac{(1-\lambda)P_{\hat{\mathcal{D}}}(s,a)}{(1-\lambda)P_{\hat{\mathcal{D}}}(s,a) + \lambda P_{\mathcal{D}'}(s,a)} \right)^2 \frac{\sigma_{\hat{s}'}^2 \left( \hat{\mathcal{B}}_{\hat{s}'} Q^k(s,a) \right)}{N(s,a)} \right]
$$

$$
+ \sum_{(s,a) \in \mathcal{S} \times \mathcal{A}} P_{\mathcal{D}}(s,a) \left( \frac{\lambda P_{\mathcal{D}'}(s,a)}{(1-\lambda)P_{\hat{\mathcal{D}}}(s,a) + \lambda P_{\mathcal{D}'}(s,a)} \right)^2 \left( \mathcal{B}_{\mathcal{D}} Q^k(s,a) - \mathcal{B}_{\mathcal{D}'} Q^k(s,a) \right)^2 \tag{67}
$$

$$
= \mathbb{E}_{(s,a) \sim \mathcal{D}|\hat{\mathcal{D}}} \left[ \left( \frac{(1-\lambda)P_{\hat{\mathcal{D}}}(s,a)}{(1-\lambda)P_{\hat{\mathcal{D}}}(s,a) + \lambda P_{\mathcal{D}'}(s,a)} \right)^2 \frac{\sigma_{\hat{s}'}^2 \left( \hat{\mathcal{B}}_{\hat{s}'} Q^k(s,a) \right)}{N(s,a)} \right]
$$

$$
+ \sum_{(s,a) \in \hat{\mathcal{D}}} P_{\mathcal{D}}(s,a) \left( \frac{\lambda P_{\mathcal{D}'}(s,a)}{(1-\lambda)P_{\hat{\mathcal{D}}}(s,a) + \lambda P_{\mathcal{D}'}(s,a)} \right)^2 \left( \mathcal{B}_{\mathcal{D}} Q^k(s,a) - \mathcal{B}_{\mathcal{D}'} Q^k(s,a) \right)^2
$$

$$
+ \sum_{(s,a) \in \mathcal{S} \times \mathcal{A} \setminus \hat{\mathcal{D}}} P_{\mathcal{D}}(s,a) \left( \frac{\lambda P_{\mathcal{D}'}(s,a)}{(1-\lambda)P_{\hat{\mathcal{D}}}(s,a) + \lambda P_{\mathcal{D}'}(s,a)} \right)^2 \left( \mathcal{B}_{\mathcal{D}} Q^k(s,a) - \mathcal{B}_{\mathcal{D}'} Q^k(s,a) \right)^2 \tag{68}
$$

It follows from the law of total probability that

$$
\mathbb{E}_{\hat{s}' \sim P_{\mathcal{D}}(\hat{s}'|s,a)} \left[ \mathcal{E}_{\mathcal{D}}(Q_\lambda^{k+1}) \right] - \mathcal{E}_{\mathcal{D}}(Q^{k+1})
$$

$$
\leq \mathbb{E}_{(s,a) \sim \mathcal{D}|\hat{\mathcal{D}}} \left[ \left( \frac{(1-\lambda)P_{\hat{\mathcal{D}}}(s,a)}{(1-\lambda)P_{\hat{\mathcal{D}}}(s,a) + \lambda P_{\mathcal{D}'}(s,a)} \right)^2 \frac{\sigma_{\hat{s}'}^2 \left( \hat{\mathcal{B}}_{\hat{s}'} Q^k(s,a) \right)}{N(s,a)} \right]
$$

$$
+ \sum_{(s,a) \in \hat{\mathcal{D}}} \left( P_{\mathcal{D}}(s,a \mid (s,a) \in \hat{\mathcal{D}}) \cdot P((s,a) \in \hat{\mathcal{D}}) + P_{\mathcal{D}}(s,a \mid (s,a) \in \mathcal{S} \times \mathcal{A} \setminus \hat{\mathcal{D}}) \cdot P((s,a) \in \mathcal{S} \times \mathcal{A} \setminus \hat{\mathcal{D}}) \right)
$$

$$
\left( \frac{\lambda P_{\mathcal{D}'}(s,a)}{(1-\lambda)P_{\hat{\mathcal{D}}}(s,a) + \lambda P_{\mathcal{D}'}(s,a)} \right)^2 \left( \mathcal{B}_{\mathcal{D}} Q^k(s,a) - \mathcal{B}_{\mathcal{D}'} Q^k(s,a) \right)^2 \tag{69}
$$

$$
+ \sum_{(s,a) \in \mathcal{S} \times \mathcal{A} \setminus \hat{\mathcal{D}}} \left( P_{\mathcal{D}}(s,a \mid (s,a) \in \hat{\mathcal{D}}) \cdot P((s,a) \in \hat{\mathcal{D}}) + P_{\mathcal{D}}(s,a \mid (s,a) \in \mathcal{S} \times \mathcal{A} \setminus \hat{\mathcal{D}}) \cdot P((s,a) \in \mathcal{S} \times \mathcal{A} \setminus \hat{\mathcal{D}}) \right)
$$

$$
\left( \frac{\lambda P_{\mathcal{D}'}(s,a)}{(1-\lambda)P_{\hat{\mathcal{D}}}(s,a) + \lambda P_{\mathcal{D}'}(s,a)} \right)^2 \left( \mathcal{B}_{\mathcal{D}} Q^k(s,a) - \mathcal{B}_{\mathcal{D}'} Q^k(s,a) \right)^2 \tag{70}
$$

Notice that $P_{\mathcal{D}}(s,a \mid (s,a) \in \mathcal{S} \times \mathcal{A} \setminus \hat{\mathcal{D}}) = 0$ for all $(s,a) \in \hat{\mathcal{D}}$ in (69) and $P_{\mathcal{D}}(s,a \mid (s,a) \in \hat{\mathcal{D}}) = 0$ for all $(s,a) \in \mathcal{S} \times \mathcal{A} \setminus \hat{\mathcal{D}}$ in (70). Subsequently, the previous inequality reduces to

$$
\mathbb{E}_{\hat{s}' \sim P_{\mathcal{D}}(\hat{s}'|s,a)} \left[ \mathcal{E}_{\mathcal{D}}(Q_\lambda^{k+1}) \right] - \mathcal{E}_{\mathcal{D}}(Q^{k+1})
$$

$$
\leq \mathbb{E}_{(s,a) \sim \mathcal{D}|\hat{\mathcal{D}}} \left[ \left( \frac{(1-\lambda)P_{\hat{\mathcal{D}}}(s,a)}{(1-\lambda)P_{\hat{\mathcal{D}}}(s,a) + \lambda P_{\mathcal{D}'}(s,a)} \right)^2 \frac{\sigma_{\hat{s}'}^2 \left( \hat{\mathcal{B}}_{\hat{s}'} Q^k(s,a) \right)}{N(s,a)} \right]
$$

$$
+ \sum_{(s,a) \in \hat{\mathcal{D}}} \left( P_{\mathcal{D}}(s,a \mid (s,a) \in \hat{\mathcal{D}}) \cdot P((s,a) \in \hat{\mathcal{D}}) \right) \left( \frac{\lambda P_{\mathcal{D}'}(s,a)}{(1-\lambda)P_{\hat{\mathcal{D}}}(s,a) + \lambda P_{\mathcal{D}'}(s,a)} \right)^2 \left( \mathcal{B}_{\mathcal{D}} Q^k(s,a) - \mathcal{B}_{\mathcal{D}'} Q^k(s,a) \right)^2
$$

$$
+ \sum_{(s,a) \in \mathcal{S} \times \mathcal{A} \setminus \hat{\mathcal{D}}} \left( P_{\mathcal{D}}(s,a \mid (s,a) \in \mathcal{S} \times \mathcal{A} \setminus \hat{\mathcal{D}}) \cdot P((s,a) \in \mathcal{S} \times \mathcal{A} \setminus \hat{\mathcal{D}}) \right) \left( \frac{\lambda P_{\mathcal{D}'}(s,a)}{(1-\lambda)P_{\hat{\mathcal{D}}}(s,a) + \lambda P_{\mathcal{D}'}(s,a)} \right)^2
$$

$$
\cdot \left( \mathcal{B}_{\mathcal{D}} Q^k(s,a) - \mathcal{B}_{\mathcal{D}'} Q^k(s,a) \right)^2. \tag{71}
$$

Moreover, it holds that

$$P((s,a) \in \hat{\mathcal{D}}) \leq 1, \tag{72}$$

$$P((s,a) \in \mathcal{S} \times \mathcal{A} \setminus \hat{\mathcal{D}}) = (1 - P_{\mathcal{D}}(s,a))^N \overset{(a)}{\leq} e^{-N P_{\mathcal{D}}(s,a)} \overset{(b)}{\leq} e^{-NC}, \tag{73}$$

where $N$ denotes the size of the target dataset, $(a)$ follows from $\log(1-\alpha) \leq -\alpha$, $\forall \alpha \in (0,1)$, and $(b)$ holds by the monotonicity of $e^{-x}$ and $C := \min_{(s,a) \in \mathcal{S} \times \mathcal{A}} P_{\mathcal{D}}(s,a)$. Substituting (72) and (73) into (71) yields

$$\mathbb{E}_{\hat{s}' \sim P_{\mathcal{D}}(\hat{s}'|s,a)} \left[ \mathcal{E}_{\mathcal{D}}(Q_\lambda^{k+1}) \right] - \mathcal{E}_{\mathcal{D}}(Q^{k+1})$$

$$\leq \mathbb{E}_{(s,a) \sim \mathcal{D}|\hat{\mathcal{D}}} \left[ \left( \frac{(1-\lambda)P_{\hat{\mathcal{D}}}(s,a)}{(1-\lambda)P_{\hat{\mathcal{D}}}(s,a) + \lambda P_{\mathcal{D}'}(s,a)} \right)^2 \frac{\sigma_{\hat{s}'}^2 \left( \hat{\mathcal{B}}_{\hat{s}'} Q^k(s,a) \right)}{N(s,a)} \right]$$

$$+ \sum_{(s,a) \in \hat{\mathcal{D}}} \left( P_{\mathcal{D}}(s,a \mid (s,a) \in \hat{\mathcal{D}}) \right) \left( \frac{\lambda P_{\mathcal{D}'}(s,a)}{(1-\lambda)P_{\hat{\mathcal{D}}}(s,a) + \lambda P_{\mathcal{D}'}(s,a)} \right)^2 \left( \mathcal{B}_{\mathcal{D}} Q^k(s,a) - \mathcal{B}_{\mathcal{D}'} Q^k(s,a) \right)^2$$

$$+ \sum_{(s,a) \in \mathcal{S} \times \mathcal{A} \setminus \hat{\mathcal{D}}} P_{\mathcal{D}}(s,a \mid (s,a) \in \mathcal{S} \times \mathcal{A} \setminus \hat{\mathcal{D}}) \cdot e^{-NC} \cdot \left( \frac{\lambda P_{\mathcal{D}'}(s,a)}{(1-\lambda)P_{\hat{\mathcal{D}}}(s,a) + \lambda P_{\mathcal{D}'}(s,a)} \right)^2$$

$$\cdot \left( \mathcal{B}_{\mathcal{D}} Q^k(s,a) - \mathcal{B}_{\mathcal{D}'} Q^k(s,a) \right)^2. \tag{74}$$

It holds by the definition of conditional expectation and $P_{\hat{\mathcal{D}}}(s,a) = 0$ for any $(s,a) \in \mathcal{S} \times \mathcal{A} \setminus \hat{\mathcal{D}}$ that

$$\mathbb{E}_{\hat{s}' \sim P_{\mathcal{D}}(\hat{s}'|s,a)} \left[ \mathcal{E}_{\mathcal{D}}(Q_\lambda^{k+1}) \right] - \mathcal{E}_{\mathcal{D}}(Q^{k+1})$$

$$\leq \mathbb{E}_{(s,a) \sim \mathcal{D}|\hat{\mathcal{D}}} \left[ \underbrace{\left( \frac{(1-\lambda)P_{\hat{\mathcal{D}}}(s,a)}{(1-\lambda)P_{\hat{\mathcal{D}}}(s,a) + \lambda P_{\mathcal{D}'}(s,a)} \right)^2}_{U_2} \frac{\sigma_{\hat{s}'}^2 \left( \hat{\mathcal{B}}_{\hat{s}'} Q^k(s,a) \right)}{N(s,a)} \right]$$

$$+ \mathbb{E}_{(s,a) \sim \mathcal{D}|\hat{\mathcal{D}}} \left[ \underbrace{\left( \frac{\lambda P_{\mathcal{D}'}(s,a)}{(1-\lambda)P_{\hat{\mathcal{D}}}(s,a) + \lambda P_{\mathcal{D}'}(s,a)} \right)^2}_{U_3} \left( \mathcal{B}_{\mathcal{D}} Q^k(s,a) - \mathcal{B}_{\mathcal{D}'} Q^k(s,a) \right)^2 \right]$$

$$+ \mathbb{E}_{(s,a) \sim \mathcal{D}|\mathcal{S} \times \mathcal{A} \setminus \hat{\mathcal{D}}} \left[ e^{-NC} \cdot \left( \mathcal{B}_{\mathcal{D}} Q^k(s,a) - \mathcal{B}_{\mathcal{D}'} Q^k(s,a) \right)^2 \right]. \tag{75}$$

Dividing both the numerator and denominator by $P_{\hat{\mathcal{D}}}(s,a)$ in $U_2$ and by $P_{\mathcal{D}'}(s,a)$ in $U_3$ yields

$$\mathbb{E}_{\hat{s}' \sim P_{\mathcal{D}}(\hat{s}'|s,a)} \left[ \mathcal{E}_{\mathcal{D}}(Q_\lambda^{k+1}) \right] - \mathcal{E}_{\mathcal{D}}(Q^{k+1})$$

$$\leq \mathbb{E}_{(s,a) \sim \mathcal{D}|\hat{\mathcal{D}}} \left[ \underbrace{\left( \frac{1-\lambda}{1 - \lambda + \lambda P_{\mathcal{D}'}(s,a)/P_{\hat{\mathcal{D}}}(s,a)} \right)^2}_{U_2} \frac{\sigma_{\hat{s}'}^2 \left( \hat{\mathcal{B}}_{\hat{s}'} Q^k(s,a) \right)}{N(s,a)} \right]$$

$$+ \mathbb{E}_{(s,a) \sim \mathcal{D}|\hat{\mathcal{D}}} \left[ \underbrace{\left( \frac{\lambda}{(1-\lambda)P_{\hat{\mathcal{D}}}(s,a)/P_{\mathcal{D}'}(s,a) + \lambda} \right)^2}_{U_3} \left( \mathcal{B}_{\mathcal{D}} Q^k(s,a) - \mathcal{B}_{\mathcal{D}'} Q^k(s,a) \right)^2 \right]$$

$$+ \mathbb{E}_{(s,a) \sim \mathcal{D}|\mathcal{S} \times \mathcal{A} \setminus \hat{\mathcal{D}}} \left[ e^{-NC} \cdot \left( \mathcal{B}_{\mathcal{D}} Q^k(s,a) - \mathcal{B}_{\mathcal{D}'} Q^k(s,a) \right)^2 \right]. \tag{76}$$

Assumption 3 implies that both $P_{\mathcal{D}'}(s,a)/P_{\hat{\mathcal{D}}}(s,a)$ and $P_{\hat{\mathcal{D}}}(s,a)/P_{\mathcal{D}'}(s,a)$ are bounded, i.e.,

$$\frac{1}{\beta_u} \leq \frac{P_{\mathcal{D}'}(s,a)}{P_{\hat{\mathcal{D}}}(s,a)} \leq \frac{1}{\beta_l}, \ \beta_l \leq \frac{P_{\hat{\mathcal{D}}}(s,a)}{P_{\mathcal{D}'}(s,a)} \leq \beta_u, \ \forall (s,a) \in \hat{\mathcal{D}}. \tag{77}$$

Substituting the previous expressions into (76) yields

$$
\mathop{\mathbb{E}}_{\hat{s}' \sim P_{\mathcal{D}}(\hat{s}'|s,a)} \left[ \mathcal{E}_{\mathcal{D}}(Q_\lambda^{k+1}) \right] - \mathcal{E}_{\mathcal{D}}(Q^{k+1}) \le \mathop{\mathbb{E}}_{(s,a) \sim \mathcal{D}|\hat{\mathcal{D}}} \left[ \left( \frac{1-\lambda}{1-\lambda+\lambda/\beta_u} \right)^2 \frac{\sigma_{\hat{s}'}^2 \left( \hat{\mathcal{B}}_{\hat{s}'} Q^k(s,a) \right)}{N(s,a)} \right]
$$

$$
+ \mathop{\mathbb{E}}_{(s,a) \sim \mathcal{D}|\hat{\mathcal{D}}} \left[ \left( \frac{\lambda}{(1-\lambda)\beta_l + \lambda} \right)^2 \left( \mathcal{B}_{\mathcal{D}} Q^k(s,a) - \mathcal{B}_{\mathcal{D}'} Q^k(s,a) \right)^2 \right]
$$

$$
+ \mathop{\mathbb{E}}_{(s,a) \sim \mathcal{D}|\mathcal{S} \times \mathcal{A} \setminus \hat{\mathcal{D}}} \left[ e^{-NC} \cdot \left( \mathcal{B}_{\mathcal{D}} Q^k(s,a) - \mathcal{B}_{\mathcal{D}'} Q^k(s,a) \right)^2 \right] \tag{78}
$$

$$
= \left( \frac{1-\lambda}{1-\lambda+\lambda/\beta_u} \right)^2 \cdot \mathop{\mathbb{E}}_{(s,a) \sim \mathcal{D}|\hat{\mathcal{D}}} \left[ \frac{\sigma_{\hat{s}'}^2 \left( \hat{\mathcal{B}}_{\hat{s}'} Q^k(s,a) \right)}{N(s,a)} \right]
$$

$$
+ \left( \frac{\lambda}{(1-\lambda)\beta_l + \lambda} \right)^2 \cdot \mathop{\mathbb{E}}_{(s,a) \sim \mathcal{D}|\hat{\mathcal{D}}} \left[ \left( \mathcal{B}_{\mathcal{D}} Q^k(s,a) - \mathcal{B}_{\mathcal{D}'} Q^k(s,a) \right)^2 \right]
$$

$$
+ e^{-NC} \cdot \mathop{\mathbb{E}}_{(s,a) \sim \mathcal{D}|\mathcal{S} \times \mathcal{A} \setminus \hat{\mathcal{D}}} \left[ \cdot \left( \mathcal{B}_{\mathcal{D}} Q^k(s,a) - \mathcal{B}_{\mathcal{D}'} Q^k(s,a) \right)^2 \right]. \tag{79}
$$

Given any dataset $\hat{\mathcal{D}}_{\mathrm{tr}}$ and its corresponding transition-excluded dataset $\hat{\mathcal{D}}$, it holds that

$$
\mathop{\mathbb{E}}_{(s,a) \sim \mathcal{D}|\hat{\mathcal{D}}} \left[ \frac{\sigma_{\hat{s}'}^2 \left( \hat{\mathcal{B}}_{\hat{s}'} Q^k(s,a) \right)}{N(s,a)} \right] \le \max_{(s,a) \in \hat{\mathcal{D}}} \left[ \frac{\sigma_{\hat{s}'}^2 \left( \hat{\mathcal{B}}_{\hat{s}'} Q^k(s,a) \right)}{N(s,a)} \right], \tag{80}
$$

$$
\mathop{\mathbb{E}}_{(s,a) \sim \mathcal{D}|\hat{\mathcal{D}}} \left[ \left( \mathcal{B}_{\mathcal{D}} Q^k(s,a) - \mathcal{B}_{\mathcal{D}'} Q^k(s,a) \right)^2 \right] \le \max_{(s,a) \in \hat{\mathcal{D}}} \left[ \left( \mathcal{B}_{\mathcal{D}} Q^k(s,a) - \mathcal{B}_{\mathcal{D}'} Q^k(s,a) \right)^2 \right] \tag{81}
$$

$$
\le \max_{(s,a) \in \mathcal{S} \times \mathcal{A}} \left[ \left( \mathcal{B}_{\mathcal{D}} Q^k(s,a) - \mathcal{B}_{\mathcal{D}'} Q^k(s,a) \right)^2 \right], \tag{82}
$$

$$
\mathop{\mathbb{E}}_{(s,a) \sim \mathcal{D}|\mathcal{S} \times \mathcal{A} \setminus \hat{\mathcal{D}}} \left[ \left( \mathcal{B}_{\mathcal{D}} Q^k(s,a) - \mathcal{B}_{\mathcal{D}'} Q^k(s,a) \right)^2 \right] \le \max_{(s,a) \in \mathcal{S} \times \mathcal{A} \setminus \hat{\mathcal{D}}} \left[ \left( \mathcal{B}_{\mathcal{D}} Q^k(s,a) - \mathcal{B}_{\mathcal{D}'} Q^k(s,a) \right)^2 \right] \tag{83}
$$

$$
\le \max_{(s,a) \in \mathcal{S} \times \mathcal{A}} \left[ \left( \mathcal{B}_{\mathcal{D}} Q^k(s,a) - \mathcal{B}_{\mathcal{D}'} Q^k(s,a) \right)^2 \right]. \tag{84}
$$

Then by the definitions of $\xi$ in (14) and $\varsigma$ in (15), (79) is equivalent to

$$
\mathop{\mathbb{E}}_{\hat{s}' \sim P_{\mathcal{D}}(\hat{s}'|s,a)} \left[ \mathcal{E}_{\mathcal{D}}(Q_\lambda^{k+1}) \right] - \mathcal{E}_{\mathcal{D}}(Q^{k+1}) \le \left( \frac{1-\lambda}{1-\lambda+\lambda/\beta_u} \right)^2 \varsigma + \left( \frac{\lambda}{(1-\lambda)\beta_l + \lambda} \right)^2 \xi + e^{-NC} \xi, \tag{85}
$$

$$
\le \left( \frac{1-\lambda}{1-\lambda+\lambda/\beta_u} \right)^2 \varsigma + \left( \left( \frac{\lambda}{(1-\lambda)\beta_l + \lambda} \right)^2 + e^{-NC} \right) \xi. \tag{86}
$$

This completes the proof of Theorem 1. $\qquad \square$

## A.3 Proof of Theorem 2

**Theorem 2** (Tighter Expected Performance Bound). Let the conditions of Theorem 1 hold. Suppose that for any $(s,a) \in \mathcal{S} \times \mathcal{A}$, $P_{\mathcal{D}}(s,a)$, $P_{\mathcal{D}'}(s,a)$ and $P_{\hat{\mathcal{D}}}(s,a)$ are positive and there exist constants $\beta_u \ge \beta_l > 0$ such that $P_{\hat{\mathcal{D}}}(s,a)/P_{\mathcal{D}'}(s,a) \in [\beta_l, \beta_u]$, $\forall (s,a) \in \mathcal{S} \times \mathcal{A}$. Given any dataset $\hat{\mathcal{D}}_{\mathrm{tr}}$, it holds at each iteration $(k=0,1,2,\cdots)$ that

$$
\mathop{\mathbb{E}}_{\hat{s}' \sim P_{\mathcal{D}}(\hat{s}'|s,a)} \left[ \mathcal{E}_{\mathcal{D}}(Q_\lambda^{k+1}) \right] - \mathcal{E}_{\mathcal{D}}(Q^{k+1}) \le \left( \frac{1-\lambda}{1-\lambda+\lambda/\beta_u} \right)^2 \varsigma + \left( \frac{\lambda}{(1-\lambda)\beta_l + \lambda} \right)^2 \xi. \tag{87}
$$

*Proof.* Analogous to the proof of Proposition 1, it holds with the stricter version of Assumption 3 that

$$Q_\lambda^{k+1}(s,a) = \frac{\frac{1-\lambda}{N}\sum_{j=1}^{N(s,a)}\hat{\mathcal{B}}_{\hat{s}'_j}Q^k(s,a) + \lambda P_{\mathcal{D}'}(s,a)\mathcal{B}_{\mathcal{D}'}Q^k(s,a)}{(1-\lambda)P_{\hat{\mathcal{D}}}(s,a) + \lambda P_{\mathcal{D}'}(s,a)}, \; \forall (s,a) \in \mathcal{S}\times\mathcal{A}. \tag{88}$$

It follows from (48) in the proof of Theorem 1 that

$$\mathop{\mathbb{E}}_{\hat{s}'\sim P_{\mathcal{D}}(\hat{s}'|s,a)}\left[\mathcal{E}_{\mathcal{D}}(Q_\lambda^{k+1})\right] - \mathcal{E}_D(Q^{k+1})$$

$$= \mathop{\mathbb{E}}_{(s,a)\sim\mathcal{D}}\left[\sigma_{\hat{s}'}^2\left(Q_\lambda^{k+1}(s,a)\right) + \left(\underbrace{\mathop{\mathbb{E}}_{\hat{s}'\sim P_{\mathcal{D}}(\hat{s}'|s,a)}\left[Q_\lambda^{k+1}(s,a)\right] - \mathcal{B}_{\mathcal{D}}Q^k(s,a)}_{U_1}\right)^2\right], \tag{89}$$

where

$$\sigma_{\hat{s}'}^2\left(Q_\lambda^{k+1}(s,a)\right) = \left(\frac{(1-\lambda)P_{\hat{\mathcal{D}}}(s,a)}{(1-\lambda)P_{\hat{\mathcal{D}}}(s,a)+\lambda P_{\mathcal{D}'}(s,a)}\right)^2 \frac{\sigma_{\hat{s}'}^2\left(\hat{\mathcal{B}}_{\hat{s}'}Q^k(s,a)\right)}{N(s,a)}, \; \forall (s,a)\in\mathcal{S}\times\mathcal{A}, \tag{90}$$

$$U_1 = \frac{\lambda P_{\mathcal{D}'}(s,a)\left(\mathcal{B}_{\mathcal{D}'}Q^k(s,a) - \mathcal{B}_{\mathcal{D}}Q^k(s,a)\right)}{(1-\lambda)P_{\hat{\mathcal{D}}}(s,a)+\lambda P_{\mathcal{D}'}(s,a)}, \; \forall (s,a)\in\mathcal{S}\times\mathcal{A}. \tag{91}$$

It then holds for any $(s,a)\in\mathcal{S}\times\mathcal{A}$ that

$$\mathop{\mathbb{E}}_{\hat{s}'\sim P_{\mathcal{D}}(\hat{s}'|s,a)}\left[\mathcal{E}_{\mathcal{D}}(Q_\lambda^{k+1})\right] - \mathcal{E}_{\mathcal{D}}(Q^{k+1}) = \mathop{\mathbb{E}}_{(s,a)\sim\mathcal{D}}\left[\left(\frac{(1-\lambda)P_{\hat{\mathcal{D}}}(s,a)}{(1-\lambda)P_{\hat{\mathcal{D}}}(s,a)+\lambda P_{\mathcal{D}'}(s,a)}\right)^2 \frac{\sigma_{\hat{s}'}^2\left(\hat{\mathcal{B}}_{\hat{s}'}Q^k(s,a)\right)}{N(s,a)}\right] +$$

$$\mathop{\mathbb{E}}_{(s,a)\sim\mathcal{D}}\left[\left(\frac{\lambda P_{\mathcal{D}'}(s,a)}{(1-\lambda)P_{\hat{\mathcal{D}}}(s,a)+\lambda P_{\mathcal{D}'}(s,a)}\right)^2\left(\mathcal{B}_{\mathcal{D}}Q^k(s,a) - \mathcal{B}_{\mathcal{D}'}Q^k(s,a)\right)^2\right]. \tag{92}$$

Since $P_{\hat{\mathcal{D}}}(s,a) > 0$ and $P_{\mathcal{D}'}(s,a) > 0$ for any $(s,a)\in\mathcal{S}\times\mathcal{A}$, dividing both the numerator and denominator by $P_{\hat{\mathcal{D}}}(s,a)$ in the first term above and by $P_{\mathcal{D}'}(s,a)$ in the second term above yields

$$\mathop{\mathbb{E}}_{\hat{s}'\sim P_{\mathcal{D}}(\hat{s}'|s,a)}\left[\mathcal{E}_{\mathcal{D}}(Q_\lambda^{k+1})\right] - \mathcal{E}_{\mathcal{D}}(Q^{k+1})$$

$$= \mathop{\mathbb{E}}_{(s,a)\sim\mathcal{D}}\left[\left(\frac{1-\lambda}{1-\lambda+\lambda P_{\mathcal{D}'}(s,a)/P_{\hat{\mathcal{D}}}(s,a)}\right)^2 \frac{\sigma_{\hat{s}'}^2\left(\hat{\mathcal{B}}_{\hat{s}'}Q^k(s,a)\right)}{N(s,a)}\right]$$

$$+ \mathop{\mathbb{E}}_{(s,a)\sim\mathcal{D}}\left[\left(\frac{\lambda}{(1-\lambda)P_{\hat{\mathcal{D}}}(s,a)/P_{\mathcal{D}'}(s,a)+\lambda}\right)^2\left(\mathcal{B}_{\mathcal{D}}Q^k(s,a) - \mathcal{B}_{\mathcal{D}'}Q^k(s,a)\right)^2\right]. \tag{93}$$

Theorem 2 assumes that both $P_{\mathcal{D}'}(s,a)/P_{\hat{\mathcal{D}}}(s,a)$ and $P_{\hat{\mathcal{D}}}(s,a)/P_{\mathcal{D}'}(s,a)$ are bounded, i.e.,

$$\frac{1}{\beta_u} \le \frac{P_{\mathcal{D}'}(s,a)}{P_{\hat{\mathcal{D}}}(s,a)} \le \frac{1}{\beta_l}, \; \beta_l \le \frac{P_{\hat{\mathcal{D}}}(s,a)}{P_{\mathcal{D}'}(s,a)} \le \beta_u, \; \forall (s,a)\in\mathcal{S}\times\mathcal{A}. \tag{94}$$

It then holds that

$$
\mathop{\mathbb{E}}_{\hat{s}' \sim P_{\mathcal{D}}(\hat{s}'|s,a)} \left[ \mathcal{E}_{\mathcal{D}}(Q_\lambda^{k+1}) \right] - \mathcal{E}_{\mathcal{D}}(Q^{k+1})
$$

$$
\leq \mathop{\mathbb{E}}_{(s,a)\sim\mathcal{D}} \left[ \left( \frac{1-\lambda}{1-\lambda+\lambda/\beta_u} \right)^2 \frac{\sigma_{\hat{s}'}^2 \left( \hat{\mathcal{B}}_{\hat{s}'} Q^k(s,a) \right)}{N(s,a)} \right] + \mathop{\mathbb{E}}_{(s,a)\sim\mathcal{D}} \left[ \left( \frac{\lambda}{(1-\lambda)\beta_l + \lambda} \right)^2 \left( \mathcal{B}_{\mathcal{D}} Q^k(s,a) - \mathcal{B}_{\mathcal{D}'} Q^k(s,a) \right)^2 \right]
$$

$$
= \left( \frac{1-\lambda}{1-\lambda+\lambda/\beta_u} \right)^2 \cdot \mathop{\mathbb{E}}_{(s,a)\sim\mathcal{D}} \left[ \frac{\sigma_{\hat{s}'}^2 \left( \hat{\mathcal{B}}_{\hat{s}'} Q^k(s,a) \right)}{N(s,a)} \right] + \left( \frac{\lambda}{(1-\lambda)\beta_l + \lambda} \right)^2 \cdot \mathop{\mathbb{E}}_{(s,a)\sim\mathcal{D}} \left[ \left( \mathcal{B}_{\mathcal{D}} Q^k(s,a) - \mathcal{B}_{\mathcal{D}'} Q^k(s,a) \right)^2 \right].
$$

$$(95)$$

Given any dataset $\hat{\mathcal{D}}_{\mathrm{tr}}$ and its corresponding transition-excluded dataset $\hat{\mathcal{D}}$, it holds that

$$
\mathop{\mathbb{E}}_{(s,a)\sim\mathcal{D}} \left[ \frac{\sigma_{\hat{s}'}^2 \left( \hat{\mathcal{B}}_{\hat{s}'} Q^k(s,a) \right)}{N(s,a)} \right] \leq \max_{(s,a)\in\hat{\mathcal{D}}} \left[ \frac{\sigma_{\hat{s}'}^2 \left( \hat{\mathcal{B}}_{\hat{s}'} Q^k(s,a) \right)}{N(s,a)} \right], \tag{96}
$$

$$
\mathop{\mathbb{E}}_{(s,a)\sim\mathcal{D}} \left[ \left( \mathcal{B}_{\mathcal{D}} Q^k(s,a) - \mathcal{B}_{\mathcal{D}'} Q^k(s,a) \right)^2 \right] \leq \max_{(s,a)\in\mathcal{S}\times\mathcal{A}} \left[ \left( \mathcal{B}_{\mathcal{D}} Q^k(s,a) - \mathcal{B}_{\mathcal{D}'} Q^k(s,a) \right)^2 \right]. \tag{97}
$$

Then by the definitions of $\xi$ in (14) and $\varsigma$ in (15), (95) is equivalent to

$$
\mathop{\mathbb{E}}_{\hat{s}' \sim P_{\mathcal{D}}(\hat{s}'|s,a)} \left[ \mathcal{E}_{\mathcal{D}}(Q_\lambda^{k+1}) \right] - \mathcal{E}_{\mathcal{D}}(Q^{k+1}) \leq \left( \frac{1-\lambda}{1-\lambda+\lambda/\beta_u} \right)^2 \varsigma + \left( \frac{\lambda}{(1-\lambda)\beta_l + \lambda} \right)^2 \xi. \tag{98}
$$

This completes the proof of Theorem 2.

$\square$

## A.4 Proof of Corollary 1

**Corollary 1**. Under the assumptions of Theorem 1 or Theorem 2, the optimal weight $\lambda^*$ that minimizes the bounds in (16) or (17) respectively is $\lambda^* = 0$ when $\varsigma = 0$ and $\lambda^* = 1$ when $\xi = 0$.

*Proof.* Notice that Theorems 1 and 2 yield the same $\lambda^*$, as the extra term $e^{-NC}\xi$ in (16) is independent of $\lambda$. Hence, we, herein, present the proof for (16), noting that the proof with respect to (17) follows identically.

We start the proof by considering $\varsigma = 0$. In this case, the bound in (16) reduces to

$$
\left( \frac{\lambda}{(1-\lambda)\beta_l + \lambda} \right)^2 \xi + e^{-NC}\xi, \tag{99}
$$

where $e^{-NC}\xi$ is a constant. Since the minimum of the squared term is 0, achieved by $\lambda = 0$, it holds that letting $\lambda = 0$ minimizes the bound $\left( \frac{\lambda}{(1-\lambda)\beta_l + \lambda} \right)^2 \xi$, thus minimizing the bound in (16).

We now turn our attention to the scenario of $\xi = 0$. Likewise, the bound in (16) reduces to

$$
\left( \frac{1-\lambda}{1-\lambda+\lambda/\beta_u} \right)^2 \varsigma. \tag{100}
$$

Therefore, setting $\lambda = 1$ minimizes the bound above as it makes the squared term equal to 0, thus minimizing the bound in (16). These complete the proof of Corollary 1.

$\square$

## A.5 Proof of Corollary 2

**Corollary 2.** Under the assumptions of Theorem 1 or Theorem 2, if $\beta_l = \beta_u = \beta > 0$, the optimal weight $\lambda^*$ that minimizes the bound in (16) or (17) respectively takes the form

$$\lambda^* = \frac{\beta\varsigma}{\beta\varsigma + \xi}. \tag{101}$$

*Proof.* Notice that Theorems 1 and 2 yield the same $\lambda^*$, as the extra term $e^{-NC}\xi$ in (16) is independent of $\lambda$. Hence, we, herein, present the proof for (16), noting that the proof with respect to (17) follows identically.

Note when $\beta_l = \beta_u = \beta$ that the right hand side of (16) reduces to

$$\left(\frac{1-\lambda}{1-\lambda+\lambda/\beta}\right)^2 \varsigma + \left(\frac{\lambda}{(1-\lambda)\beta+\lambda}\right)^2 \xi + e^{-NC}\xi, \tag{102}$$

i.e.,

$$\left(\frac{(1-\lambda)\beta}{(1-\lambda)\beta+\lambda}\right)^2 \varsigma + \left(\frac{\lambda}{(1-\lambda)\beta+\lambda}\right)^2 \xi + e^{-NC}\xi. \tag{103}$$

Taking the derivative of the previous bound with respect to $\lambda$ yields

$$\text{Derivative} = 2\varsigma \left(\frac{(1-\lambda)\beta}{(1-\lambda)\beta+\lambda}\right) \frac{-\beta\left((1-\lambda)\beta+\lambda\right)-(1-\lambda)\beta(1-\beta)}{\left((1-\lambda)\beta+\lambda\right)^2}$$
$$+ 2\xi \left(\frac{\lambda}{(1-\lambda)\beta+\lambda}\right) \frac{(1-\lambda)\beta+\lambda-\lambda(1-\beta)}{\left((1-\lambda)\beta+\lambda\right)^2}. \tag{104}$$

By combining and canceling terms the previous derivative reduces to

$$\text{Derivative} = 2\varsigma \frac{(\lambda-1)\beta^2}{\left((1-\lambda)\beta+\lambda\right)^3} + 2\xi \frac{\lambda\beta}{\left((1-\lambda)\beta+\lambda\right)^3}. \tag{105}$$

Notice that in the previous equation $(1-\lambda)\beta+\lambda > 0$ due to that $\beta > 0$ and $\lambda \in [0,1]$. We further solve the optimal weight $\lambda^*$ by letting the above derivative be zero, i.e.,

$$\lambda^* = \frac{\beta\varsigma}{\beta\varsigma + \xi}. \tag{106}$$

This completes the proof of Corollary 2.

$\square$

## A.6 Proof of Theorem 3

**Theorem 3** (Worst-Case Performance Bound). Denote by $\beta'_u$ the upper bound of $P_{\mathcal{D}}(s,a)/P_{\mathcal{D}'}(s,a)$ for any $(s,a) \in \mathcal{S} \times \mathcal{A}$. Let the conditions of Theorem 1 and Assumption 1 hold. Given any dataset $\hat{\mathcal{D}}_{\text{tr}}$, the following bound holds at each iteration $(k = 0,1,2,\cdots)$ with probability at least $1 - \delta$

$$\mathcal{E}_{\mathcal{D}}(Q_\lambda^{k+1}) - \mathcal{E}_{\mathcal{D}}(Q^{k+1})$$
$$\leq \left(\frac{1-\lambda}{1-\lambda+\lambda/\beta_u}\right)^2 \varsigma + \left(\frac{\lambda}{(1-\lambda)\beta_l+\lambda}\right)^2 \xi + e^{-NC}\xi$$
$$+ \sqrt{\frac{1}{2}\log\left(\frac{1}{\delta}\right)} \frac{|\mathcal{S}||\mathcal{A}|}{\sqrt{N}} \left(\frac{\beta'_u}{(1-\lambda)\beta_l+\lambda} \frac{2(1-\lambda)\gamma B}{1-\gamma}\right) \cdot \left(\frac{(1-\lambda)\beta_u(4B/(1-\gamma))+2\lambda\sqrt{\xi}}{(1-\lambda)\beta_l+\lambda}\right). \tag{107}$$

*Proof.* Note that

$$
\begin{aligned}
&\mathcal{E}_{\mathcal{D}}(Q_\lambda^{k+1}) - \mathcal{E}_{\mathcal{D}}(Q^{k+1}) \\
&= \mathcal{E}_{\mathcal{D}}(Q_\lambda^{k+1}) - \underset{\hat{s}' \sim P_{\mathcal{D}}(\hat{s}'|s,a)}{\mathbb{E}}\left[\mathcal{E}_{\mathcal{D}}(Q_\lambda^{k+1})\right] + \underset{\hat{s}' \sim P_{\mathcal{D}}(\hat{s}'|s,a)}{\mathbb{E}}\left[\mathcal{E}_{\mathcal{D}}(Q_\lambda^{k+1})\right] - \mathcal{E}_{\mathcal{D}}(Q^{k+1})
\end{aligned}
\tag{108}
$$

$$
\leq \mathcal{E}_{\mathcal{D}}(Q_\lambda^{k+1}) - \underset{\hat{s}' \sim P_{\mathcal{D}}(\hat{s}'|s,a)}{\mathbb{E}}\left[\mathcal{E}_{\mathcal{D}}(Q_\lambda^{k+1})\right] + \left(\frac{1-\lambda}{1-\lambda+\lambda/\beta_u}\right)^2 \varsigma + \left(\frac{\lambda}{(1-\lambda)\beta_l + \lambda}\right)^2 \xi + e^{-NC}\xi,
\tag{109}
$$

where the previous inequality follows from Theorem 1. Thus, we are left to bound $\mathcal{E}_{\mathcal{D}}(Q_\lambda^{k+1}) - \underset{\hat{s}' \sim P_{\mathcal{D}}(\hat{s}'|s,a)}{\mathbb{E}}\left[\mathcal{E}_{\mathcal{D}}(Q_\lambda^{k+1})\right]$. To proceed, we rely on the following technical lemma.

**Lemma 1** (*McDiarmid Inequality*)**.** *Let $\tau_1, \cdots, \tau_n$ be independent random variables taking on values in a set $H$ and let $c_1, \cdots, c_n$ be positive real constants. If $\varphi : H^n \to \mathbb{R}$ satisfies*

$$
\sup_{\tau_1, \cdots, \tau_n, \tau_i' \in H} |\varphi(\tau_1, \cdots, \tau_i, \cdots, \tau_n) - \varphi(\tau_1, \cdots, \tau_i', \cdots, \tau_n)| \leq c_i,
\tag{110}
$$

*for $1 \leq i \leq n$, then it holds that*

$$
P\left(\varphi(\tau_1, \cdots, \tau_n) - \mathbb{E}\left[\varphi(\tau_1, \cdots, \tau_n)\right] \geq \epsilon\right) \leq \exp\left(\frac{-2\epsilon^2}{\sum_{i=1}^{n} c_i^2}\right).
\tag{111}
$$

To obtain the similar generalization bound akin to the above, we aim to compute the bound of $\left|\mathcal{E}_{\mathcal{D}}(Q_\lambda^{k+1}) - \mathcal{E}_{\mathcal{D}}(\hat{Q}_\lambda^{k+1})\right|$, where $Q_\lambda^{k+1}$ and $\hat{Q}_\lambda^{k+1}$ (both see (13)) differ in a single sample of $\hat{s}'$ only. More specifically, they take the sequences of random samples $(\hat{s}_1', \cdots, \hat{s}_i', \cdots, \hat{s}_N')$ and $(\hat{s}_1'', \cdots, \hat{s}_i'', \cdots, \hat{s}_N'')$ respectively, where $\hat{s}_j' = \hat{s}_j''$ for any $j \in \{1, 2, \cdots, N\}$ and $j \neq i$. Note that

$$
\begin{aligned}
&\left|\mathcal{E}_{\mathcal{D}}(Q_\lambda^{k+1}) - \mathcal{E}_{\mathcal{D}}(\hat{Q}_\lambda^{k+1})\right| \\
&\overset{(a)}{=} \left|\underset{(s,a,s') \sim \mathcal{D}}{\mathbb{E}}\left[\left(Q_\lambda^{k+1}(s,a) - \hat{\mathcal{B}}_{s'}Q^k(s,a)\right)^2 - \left(\hat{Q}_\lambda^{k+1}(s,a) - \hat{\mathcal{B}}_{s'}Q^k(s,a)\right)^2\right]\right|
\end{aligned}
\tag{112}
$$

$$
\overset{(b)}{=} \left|\underset{(s,a,s') \sim \mathcal{D}}{\mathbb{E}}\left[(Q_\lambda^{k+1}(s,a))^2 - (\hat{Q}_\lambda^{k+1}(s,a))^2 + 2\hat{\mathcal{B}}_{s'}Q^k(s,a)\left(\hat{Q}_\lambda^{k+1}(s,a) - Q_\lambda^{k+1}(s,a)\right)\right]\right|,
\tag{113}
$$

where $(a)$ follows from the definition in (4) and (5), $(b)$ follows from expanding the square and canceling the terms.

Notice that $\hat{\mathcal{B}}_{s'}Q^k(s,a)$ is the only term in the previous expression that depends on $s'$. To proceed, we condition on $(s,a)$ in the previous equation, i.e.,

$$
\begin{aligned}
&\left|\mathcal{E}_{\mathcal{D}}(Q_\lambda^{k+1}) - \mathcal{E}_{\mathcal{D}}(\hat{Q}_\lambda^{k+1})\right| \\
&\overset{(a)}{=} \left|\underset{(s,a) \sim \mathcal{D}}{\mathbb{E}}\left[(Q_\lambda^{k+1}(s,a))^2 - (\hat{Q}_\lambda^{k+1}(s,a))^2 + 2\mathcal{B}_{\mathcal{D}}Q^k(s,a)\left(\hat{Q}_\lambda^{k+1}(s,a) - Q_\lambda^{k+1}(s,a)\right)\right]\right|
\end{aligned}
\tag{114}
$$

$$
\overset{(b)}{=} \left|\underset{(s,a) \sim \mathcal{D}}{\mathbb{E}}\left[\left(Q_\lambda^{k+1}(s,a) - \hat{Q}_\lambda^{k+1}(s,a)\right)\left(Q_\lambda^{k+1}(s,a) + \hat{Q}_\lambda^{k+1}(s,a) - 2\mathcal{B}_{\mathcal{D}}Q^k(s,a)\right)\right]\right|,
\tag{115}
$$

$$
\overset{(c)}{\leq} \underset{(s,a) \sim \mathcal{D}}{\mathbb{E}}\left[\underbrace{\left|\left(Q_\lambda^{k+1}(s,a) - \hat{Q}_\lambda^{k+1}(s,a)\right)\right|}_{G_1} \cdot \underbrace{\left|\left(Q_\lambda^{k+1}(s,a) + \hat{Q}_\lambda^{k+1}(s,a) - 2\mathcal{B}_{\mathcal{D}}Q^k(s,a)\right)\right|}_{G_2}\right],
\tag{116}
$$

where $(a)$ follows by definition $\mathcal{B}_{\mathcal{D}}Q^k(s,a) = \mathbb{E}_{s' \sim P_{\mathcal{D}}(s'|s,a)}\left[\hat{\mathcal{B}}_{s'}Q^k(s,a)\right]$ (see (2) or (3)), $(b)$ follows by combining the terms, and $(c)$ follows by Jensen's inequality.

We next work on $G_1$ and $G_2$ individually. Substituting the expressions of $Q_\lambda^{k+1}(s,a)$ and $\hat{Q}_\lambda^{k+1}(s,a)$ from (13) into $G_1$ and $G_2$ yields

$$G_1 = \left| \frac{(1-\lambda)/N\left(\hat{\mathcal{B}}_{\hat{s}_i'}Q^k(s,a) - \hat{\mathcal{B}}_{\hat{s}_i''}Q^k(s,a)\right)}{(1-\lambda)P_{\hat{\mathcal{D}}}(s,a) + \lambda P_{\mathcal{D}'}(s,a)} \right|. \tag{117}$$

$$G_2 = \left| \frac{\frac{1-\lambda}{N}\sum_{j=1}^{N(s,a)}\left(\hat{\mathcal{B}}_{\hat{s}_j'}Q^k(s,a) + \hat{\mathcal{B}}_{\hat{s}_j''}Q^k(s,a)\right) + 2\lambda P_{\mathcal{D}'}(s,a)\mathcal{B}_{\mathcal{D}'}Q^k(s,a)}{(1-\lambda)P_{\hat{\mathcal{D}}}(s,a) + \lambda P_{\mathcal{D}'}(s,a)} - 2\mathcal{B}_{\mathcal{D}}Q^k(s,a) \right|. \tag{118}$$

By combining and canceling the terms $G_2$ can be further simplified as

$$G_2 = \left| \frac{\frac{1-\lambda}{N}\sum_{j=1}^{N(s,a)}\left(\hat{\mathcal{B}}_{\hat{s}_j'}Q^k(s,a) + \hat{\mathcal{B}}_{\hat{s}_j''}Q^k(s,a)\right) - 2(1-\lambda)P_{\hat{\mathcal{D}}}(s,a)\mathcal{B}_{\mathcal{D}}Q^k(s,a)}{(1-\lambda)P_{\hat{\mathcal{D}}}(s,a) + \lambda P_{\mathcal{D}'}(s,a)} \right. $$
$$\left. + \frac{2\lambda P_{\mathcal{D}'}(s,a)\left(\mathcal{B}_{\mathcal{D}'}Q^k(s,a) - \mathcal{B}_{\mathcal{D}}Q^k(s,a)\right)}{(1-\lambda)P_{\hat{\mathcal{D}}}(s,a) + \lambda P_{\mathcal{D}'}(s,a)} \right|. \tag{119}$$

Note that Assumption 1 and (11) combining with (2) or (3) implies

$$\left| \hat{\mathcal{B}}_{\hat{s}_i'}Q^k(s,a) - \hat{\mathcal{B}}_{\hat{s}_i''}Q^k(s,a) \right| \le \gamma\left(\frac{B}{1-\gamma} + \frac{B}{1-\gamma}\right) = \frac{2\gamma B}{1-\gamma}, \forall i \tag{120}$$

$$\left| \hat{\mathcal{B}}_{\hat{s}_j'}Q^k(s,a) \right| \le B + \gamma\frac{B}{1-\gamma} = \frac{B}{1-\gamma}, \forall j \tag{121}$$

$$\left| \sum_{j=1}^{N(s,a)} \hat{\mathcal{B}}_{\hat{s}_j'}Q^k(s,a) \right| \le \sum_{j=1}^{N(s,a)} \left| \hat{\mathcal{B}}_{\hat{s}_j'}Q^k(s,a) \right| \le N(s,a)\frac{B}{1-\gamma}. \tag{122}$$

Applying the above bounds to $G_1$ and $G_2$ yields

$$G_1 \le \left| \frac{(1-\lambda)/N}{(1-\lambda)P_{\hat{\mathcal{D}}}(s,a) + \lambda P_{\mathcal{D}'}(s,a)}\frac{2\gamma B}{1-\gamma} \right|. \tag{123}$$

$$G_2 \le \left| \frac{2\frac{1-\lambda}{N}N(s,a)\frac{B}{1-\gamma} - 2(1-\lambda)P_{\hat{\mathcal{D}}}(s,a)\mathcal{B}_{\mathcal{D}}Q^k(s,a)}{(1-\lambda)P_{\hat{\mathcal{D}}}(s,a) + \lambda P_{\mathcal{D}'}(s,a)} \right. $$
$$\left. + \frac{2\lambda P_{\mathcal{D}'}(s,a)\left(\mathcal{B}_{\mathcal{D}'}Q^k(s,a) - \mathcal{B}_{\mathcal{D}}Q^k(s,a)\right)}{(1-\lambda)P_{\hat{\mathcal{D}}}(s,a) + \lambda P_{\mathcal{D}'}(s,a)} \right| \tag{124}$$

$$\le \left| \frac{2(1-\lambda)P_{\hat{\mathcal{D}}}(s,a)(\frac{B}{1-\gamma} + \frac{B}{1-\gamma})}{(1-\lambda)P_{\hat{\mathcal{D}}}(s,a) + \lambda P_{\mathcal{D}'}(s,a)} + \frac{2\lambda P_{\mathcal{D}'}(s,a)\left(\mathcal{B}_{\mathcal{D}'}Q^k(s,a) - \mathcal{B}_{\mathcal{D}}Q^k(s,a)\right)}{(1-\lambda)P_{\hat{\mathcal{D}}}(s,a) + \lambda P_{\mathcal{D}'}(s,a)} \right| \tag{125}$$

$$= \left| \frac{(1-\lambda)P_{\hat{\mathcal{D}}}(s,a)(4B/(1-\gamma)) + 2\lambda P_{\mathcal{D}'}(s,a)\left(\mathcal{B}_{\mathcal{D}'}Q^k(s,a) - \mathcal{B}_{\mathcal{D}}Q^k(s,a)\right)}{(1-\lambda)P_{\hat{\mathcal{D}}}(s,a) + \lambda P_{\mathcal{D}'}(s,a)} \right|. \tag{126}$$

Substituting $G_1$ and $G_2$ back into (116) yields

$$\left| \mathcal{E}_{\mathcal{D}}(Q_\lambda^{k+1}) - \mathcal{E}_{\mathcal{D}}(\hat{Q}_\lambda^{k+1}) \right|$$
$$\le \mathbb{E}_{(s,a)\sim\mathcal{D}}\left[ \left| \frac{(1-\lambda)/N}{(1-\lambda)P_{\hat{\mathcal{D}}}(s,a) + \lambda P_{\mathcal{D}'}(s,a)}\frac{2\gamma B}{1-\gamma} \right| \right.$$
$$\left. \cdot \left| \frac{(1-\lambda)P_{\hat{\mathcal{D}}}(s,a)(4B/(1-\gamma)) + 2\lambda P_{\mathcal{D}'}(s,a)\left| \mathcal{B}_{\mathcal{D}'}Q^k(s,a) - \mathcal{B}_{\mathcal{D}}Q^k(s,a) \right|}{(1-\lambda)P_{\hat{\mathcal{D}}}(s,a) + \lambda P_{\mathcal{D}'}(s,a)} \right| \right]. \tag{127}$$

We rewrite the previous inequality using the definition of the expectation

$$
\left| \mathcal{E}_{\mathcal{D}}(Q_\lambda^{k+1}) - \mathcal{E}_{\mathcal{D}}(\hat{Q}_\lambda^{k+1}) \right|
$$
$$
\leq \sum_{(s,a)\in\mathcal{S}\times\mathcal{A}} P_{\mathcal{D}}(s,a) \left( \frac{(1-\lambda)/N}{(1-\lambda)P_{\hat{\mathcal{D}}}(s,a) + \lambda P_{\mathcal{D}'}(s,a)} \frac{2\gamma B}{1-\gamma} \right)
$$
$$
\cdot \left| \left( \frac{(1-\lambda)P_{\hat{\mathcal{D}}}(s,a)(4B/(1-\gamma)) + 2\lambda P_{\mathcal{D}'}(s,a) \left| \mathcal{B}_{\mathcal{D}'}Q^k(s,a) - \mathcal{B}_{\mathcal{D}}Q^k(s,a)) \right|}{(1-\lambda)P_{\hat{\mathcal{D}}}(s,a) + \lambda P_{\mathcal{D}'}(s,a)} \right) \right| \tag{128}
$$
$$
= \frac{1}{N} \sum_{(s,a)\in\mathcal{S}\times\mathcal{A}} \left( \frac{P_{\mathcal{D}}(s,a)}{(1-\lambda)P_{\hat{\mathcal{D}}}(s,a) + \lambda P_{\mathcal{D}'}(s,a)} \frac{2(1-\lambda)\gamma B}{1-\gamma} \right)
$$
$$
\cdot \left| \left( \frac{(1-\lambda)P_{\hat{\mathcal{D}}}(s,a)(4B/(1-\gamma)) + 2\lambda P_{\mathcal{D}'}(s,a) \left| \mathcal{B}_{\mathcal{D}'}Q^k(s,a) - \mathcal{B}_{\mathcal{D}}Q^k(s,a)) \right|}{(1-\lambda)P_{\hat{\mathcal{D}}}(s,a) + \lambda P_{\mathcal{D}'}(s,a)} \right) \right|. \tag{129}
$$

Dividing both the numerator and denominator by $P_{\mathcal{D}'}(s,a)$ in the right hand side of the previous expression, and re-ordering terms yields

$$
\left| \mathcal{E}_{\mathcal{D}}(Q_\lambda^{k+1}) - \mathcal{E}_{\mathcal{D}}(\hat{Q}_\lambda^{k+1}) \right|
$$
$$
\leq \frac{1}{N} \sum_{(s,a)\in\mathcal{S}\times\mathcal{A}} \left( \frac{P_{\mathcal{D}}(s,a)/P_{\mathcal{D}'}(s,a)}{(1-\lambda)P_{\hat{\mathcal{D}}}(s,a)/P_{\mathcal{D}'}(s,a) + \lambda} \frac{2(1-\lambda)\gamma B}{1-\gamma} \right)
$$
$$
\cdot \left| \left( \frac{(1-\lambda)P_{\hat{\mathcal{D}}}(s,a)/P_{\mathcal{D}'}(s,a)(4B/(1-\gamma)) + 2\lambda \left| \mathcal{B}_{\mathcal{D}'}Q^k(s,a) - \mathcal{B}_{\mathcal{D}}Q^k(s,a)) \right|}{(1-\lambda)P_{\hat{\mathcal{D}}}(s,a)/P_{\mathcal{D}'}(s,a) + \lambda} \right) \right|. \tag{130}
$$

By using Assumption 3 and the definition of $\beta_u'$ the previous inequality reduces to

$$
\left| \mathcal{E}_{\mathcal{D}}(Q_\lambda^{k+1}) - \mathcal{E}_{\mathcal{D}}(\hat{Q}_\lambda^{k+1}) \right|
$$
$$
\leq \frac{1}{N} \sum_{(s,a)\in\mathcal{S}\times\mathcal{A}} \left( \frac{\beta_u'}{(1-\lambda)\beta_l + \lambda} \frac{2(1-\lambda)\gamma B}{1-\gamma} \right)
$$
$$
\cdot \left| \left( \frac{(1-\lambda)\beta_u(4B/(1-\gamma)) + 2\lambda \left| \mathcal{B}_{\mathcal{D}'}Q^k(s,a) - \mathcal{B}_{\mathcal{D}}Q^k(s,a)) \right|}{(1-\lambda)\beta_l + \lambda} \right) \right| \tag{131}
$$
$$
\leq \frac{1}{N} \sum_{(s,a)\in\mathcal{S}\times\mathcal{A}} \left( \frac{\beta_u'}{(1-\lambda)\beta_l + \lambda} \frac{2(1-\lambda)\gamma B}{1-\gamma} \right)
$$
$$
\cdot \left| \left( \frac{(1-\lambda)\beta_u(4B/(1-\gamma)) + 2\lambda \max\limits_{(s,a)\in\mathcal{S}\times\mathcal{A}} \left| \mathcal{B}_{\mathcal{D}'}Q^k(s,a) - \mathcal{B}_{\mathcal{D}}Q^k(s,a)) \right|}{(1-\lambda)\beta_l + \lambda} \right) \right| \tag{132}
$$

By the definition of $\xi$ as in (14), the above expression is equivalent to

$$
\left| \mathcal{E}_{\mathcal{D}}(Q_\lambda^{k+1}) - \mathcal{E}_{\mathcal{D}}(\hat{Q}_\lambda^{k+1}) \right|
$$
$$
\leq \frac{1}{N} \sum_{(s,a)\in\mathcal{S}\times\mathcal{A}} \left( \frac{\beta_u'}{(1-\lambda)\beta_l + \lambda} \frac{2(1-\lambda)\gamma B}{1-\gamma} \right) \cdot \left| \left( \frac{(1-\lambda)\beta_u(4B/(1-\gamma)) + 2\lambda\sqrt{\xi}}{(1-\lambda)\beta_l + \lambda} \right) \right| \tag{133}
$$
$$
= \frac{|\mathcal{S}||\mathcal{A}|}{N} \left( \frac{\beta_u'}{(1-\lambda)\beta_l + \lambda} \frac{2(1-\lambda)\gamma B}{1-\gamma} \right) \cdot \left( \frac{(1-\lambda)\beta_u(4B/(1-\gamma)) + 2\lambda\sqrt{\xi}}{(1-\lambda)\beta_l + \lambda} \right) \tag{134}
$$
$$
= c, \tag{135}
$$

where $c$ is the bound as shown in Lemma 1. Lemma 1 implies that

$$P\left(\mathcal{E}_{\mathcal{D}}(Q_\lambda^{k+1}) - \mathop{\mathbb{E}}_{\hat{s}'\sim P_{\mathcal{D}}(\hat{s}'|s,a)}\left[\mathcal{E}_{\mathcal{D}}(Q_\lambda^{k+1})\right] \geq \epsilon\right) \leq \exp\frac{-2\epsilon^2}{\sum_{i=1}^N c^2} \tag{136}$$

i.e.,

$$P\left(\mathcal{E}_{\mathcal{D}}(Q_\lambda^{k+1}) - \mathop{\mathbb{E}}_{\hat{s}'\sim P_{\mathcal{D}}(\hat{s}'|s,a)}\left[\mathcal{E}_{\mathcal{D}}(Q_\lambda^{k+1})\right] < \epsilon\right) \geq 1 - \exp\frac{-2\epsilon^2}{Nc^2}. \tag{137}$$

Let the right hand side of the previous expression to be $1 - \delta$. Then,

$$\epsilon = \sqrt{\frac{1}{2}\log(\frac{1}{\delta})Nc} \tag{138}$$

$$= \sqrt{\frac{1}{2}\log(\frac{1}{\delta})}\frac{|\mathcal{S}||\mathcal{A}|}{\sqrt{N}}\left(\frac{\beta_u'}{(1-\lambda)\,\beta_l+\lambda}\frac{2(1-\lambda)\gamma B}{1-\gamma}\right)\cdot\left(\frac{(1-\lambda)\beta_u(4B/(1-\gamma))+2\lambda\sqrt{\xi}}{(1-\lambda)\,\beta_l+\lambda}\right). \tag{139}$$

Therefore, the following bound holds with probability at least $1 - \delta$

$$\mathcal{E}_{\mathcal{D}}(Q_\lambda^{k+1}) - \mathcal{E}_{\mathcal{D}}(Q^{k+1})$$
$$\leq \left(\frac{1-\lambda}{1-\lambda+\lambda/\beta_u}\right)^2\varsigma + \left(\frac{\lambda}{(1-\lambda)\,\beta_l+\lambda}\right)^2\xi + e^{-NC}\xi$$
$$+ \sqrt{\frac{1}{2}\log(\frac{1}{\delta})}\frac{|\mathcal{S}||\mathcal{A}|}{\sqrt{N}}\left(\frac{\beta_u'}{(1-\lambda)\,\beta_l+\lambda}\frac{2(1-\lambda)\gamma B}{1-\gamma}\right)\cdot\left(\frac{(1-\lambda)\beta_u(4B/(1-\gamma))+2\lambda\sqrt{\xi}}{(1-\lambda)\,\beta_l+\lambda}\right), \tag{140}$$

which completes the proof of Theorem 3.

$\square$

## A.7   Proof of Theorem 4

To proceed, we rely on the following technical lemma.

**Lemma 2.** *Given any dataset $\hat{\mathcal{D}}_{tr}$ and its corresponding transition-excluded dataset $\hat{\mathcal{D}}$, let us define $\xi_{max} = \sup_{k\in\mathbb{N}} \xi(Q^k)$ and $\varsigma_{max} = \sup_{k\in\mathbb{N}} \varsigma(Q^k)$. Then, it holds that*

$$\mathop{\mathbb{E}}_{\hat{s}'\sim P_{\mathcal{D}}(\hat{s}'|s,a)}\left[\mathop{\mathbb{E}}_{(s,a)\sim\mathcal{D}}\left[||Q_\lambda^{k+1}(s,a) - \mathcal{B}_{\mathcal{D}}Q_\lambda^k(s,a)||_\infty\right]\right] \leq \frac{1-\lambda}{1-\lambda+\lambda/\beta_u}\sqrt{\varsigma_{max}} + \frac{\lambda}{(1-\lambda)\,\beta_l+\lambda}\sqrt{\xi_{max}}$$
$$+ e^{-NC}\sqrt{\xi_{max}}, \ \forall k\in\mathbb{N}. \tag{141}$$

*Proof.* Given any dataset $\hat{\mathcal{D}}_{\mathrm{tr}}$ and its corresponding transition-excluded dataset $\hat{\mathcal{D}}$, it holds that

$$\mathop{\mathbb{E}}_{\hat{s}'\sim P_{\mathcal{D}}(\hat{s}'|s,a)}\left[\mathop{\mathbb{E}}_{(s,a)\sim\mathcal{D}}\left[||Q_\lambda^{k+1}(s,a) - \mathcal{B}_{\mathcal{D}}Q_\lambda^k(s,a)||_\infty\right]\right]$$

$$\overset{(a)}{\leq} \mathop{\mathbb{E}}_{(s,a)\sim\mathcal{D}}\left[\mathop{\mathbb{E}}_{\hat{s}'\sim P_{\mathcal{D}}(\hat{s}'|s,a)}\left[||Q_\lambda^{k+1}(s,a) - \mathcal{B}_{\mathcal{D}}Q_\lambda^k(s,a)||\right]\right] \tag{142}$$

$$\overset{(b)}{=} \mathop{\mathbb{E}}_{(s,a)\sim\mathcal{D}}\left[\mathop{\mathbb{E}}_{\hat{s}'\sim P_{\mathcal{D}}(\hat{s}'|s,a)}\left[||\frac{\frac{1-\lambda}{N}\sum_{j=1}^{N(s,a)}\hat{\mathcal{B}}_{\hat{s}_j'}Q_\lambda^k(s,a)+\lambda P_{\mathcal{D}'}(s,a)\mathcal{B}_{\mathcal{D}'}Q_\lambda^k(s,a)}{(1-\lambda)P_{\hat{\mathcal{D}}}(s,a)+\lambda P_{\mathcal{D}'}(s,a)} - \mathcal{B}_{\mathcal{D}}Q_\lambda^k(s,a)||\right]\right] \tag{143}$$

$$\overset{(c)}{=} \mathop{\mathbb{E}}_{(s,a)\sim\mathcal{D}}\left[\mathop{\mathbb{E}}_{\hat{s}'\sim P_{\mathcal{D}}(\hat{s}'|s,a)}\left[||\frac{\frac{1-\lambda}{N}\sum_{j=1}^{N(s,a)}\hat{\mathcal{B}}_{\hat{s}_j'}Q_\lambda^k(s,a) - (1-\lambda)P_{\hat{\mathcal{D}}}(s,a)\mathcal{B}_{\mathcal{D}}Q_\lambda^k(s,a))}{(1-\lambda)P_{\hat{\mathcal{D}}}(s,a)+\lambda P_{\mathcal{D}'}(s,a)}\right.\right.$$
$$\left.\left.+ \frac{\lambda P_{\mathcal{D}'}(s,a)(\mathcal{B}_{\mathcal{D}'}Q_\lambda^k(s,a) - \mathcal{B}_{\mathcal{D}}Q_\lambda^k(s,a))}{(1-\lambda)P_{\hat{\mathcal{D}}}(s,a)+\lambda P_{\mathcal{D}'}(s,a)}||\right]\right] \tag{144}$$

where $(a)$ follows from swapping the two expectations and the fact that $||\cdot||_\infty \leq ||\cdot||$, $(b)$ follows by definition of $Q_\lambda^{k+1}(s,a)$ (see (13)) that operates over $Q_\lambda^k(s,a)$, $(c)$ follows by combining the terms.

By using the triangle inequality the previous expression reduces to

$$
\mathbb{E}_{\hat{s}'\sim P_{\mathcal{D}}(\hat{s}'|s,a)}\left[\mathbb{E}_{(s,a)\sim\mathcal{D}}\left[||Q_\lambda^{k+1}(s,a)-\mathcal{B}_{\mathcal{D}}Q_\lambda^k(s,a)||_\infty\right]\right]
$$
$$
\leq \mathbb{E}_{(s,a)\sim\mathcal{D}}\left[\mathbb{E}_{\hat{s}'\sim P_{\mathcal{D}}(\hat{s}'|s,a)}\left[||\frac{\frac{1-\lambda}{N}\sum_{j=1}^{N(s,a)}\hat{\mathcal{B}}_{\hat{s}_j'}Q_\lambda^k(s,a)-(1-\lambda)P_{\hat{\mathcal{D}}}(s,a)\mathcal{B}_{\mathcal{D}}Q_\lambda^k(s,a))}{(1-\lambda)P_{\hat{\mathcal{D}}}(s,a)+\lambda P_{\mathcal{D}'}(s,a)}||\right]\right]
$$
$$
+\mathbb{E}_{(s,a)\sim\mathcal{D}}\left[\mathbb{E}_{\hat{s}'\sim P_{\mathcal{D}}(\hat{s}'|s,a)}\left[||\frac{\lambda P_{\mathcal{D}'}(s,a)(\mathcal{B}_{\mathcal{D}'}Q_\lambda^k(s,a)-\mathcal{B}_{\mathcal{D}}Q_\lambda^k(s,a))}{(1-\lambda)P_{\hat{\mathcal{D}}}(s,a)+\lambda P_{\mathcal{D}'}(s,a)}||\right]\right]. \tag{145}
$$

By the definition $P_{\hat{\mathcal{D}}}(s,a)=N(s,a)/N$, the previous inequality is equivalent to

$$
\mathbb{E}_{\hat{s}'\sim P_{\mathcal{D}}(\hat{s}'|s,a)}\left[\mathbb{E}_{(s,a)\sim\mathcal{D}}\left[||Q_\lambda^{k+1}(s,a)-\mathcal{B}_{\mathcal{D}}Q_\lambda^k(s,a)||_\infty\right]\right]
$$
$$
\leq \mathbb{E}_{(s,a)\sim\mathcal{D}}\left[\mathbb{E}_{\hat{s}'\sim P_{\mathcal{D}}(\hat{s}'|s,a)}\left[||\frac{(1-\lambda)\frac{P_{\hat{\mathcal{D}}}(s,a)}{N(s,a)}\sum_{j=1}^{N(s,a)}\hat{\mathcal{B}}_{\hat{s}_j'}Q_\lambda^k(s,a)-(1-\lambda)P_{\hat{\mathcal{D}}}(s,a)\mathcal{B}_{\mathcal{D}}Q_\lambda^k(s,a))}{(1-\lambda)P_{\hat{\mathcal{D}}}(s,a)+\lambda P_{\mathcal{D}'}(s,a)}||\right]\right]
$$
$$
+\mathbb{E}_{(s,a)\sim\mathcal{D}}\left[\mathbb{E}_{\hat{s}'\sim P_{\mathcal{D}}(\hat{s}'|s,a)}\left[||\frac{\lambda P_{\mathcal{D}'}(s,a)(\mathcal{B}_{\mathcal{D}'}Q_\lambda^k(s,a)-\mathcal{B}_{\mathcal{D}}Q_\lambda^k(s,a))}{(1-\lambda)P_{\hat{\mathcal{D}}}(s,a)+\lambda P_{\mathcal{D}'}(s,a)}||\right]\right] \tag{146}
$$
$$
\leq \mathbb{E}_{(s,a)\sim\mathcal{D}|\hat{\mathcal{D}}}\left[\mathbb{E}_{\hat{s}'\sim P_{\mathcal{D}}(\hat{s}'|s,a)}\left[||\frac{(1-\lambda)\frac{P_{\hat{\mathcal{D}}}(s,a)}{N(s,a)}\sum_{j=1}^{N(s,a)}\hat{\mathcal{B}}_{\hat{s}_j'}Q_\lambda^k(s,a)-(1-\lambda)P_{\hat{\mathcal{D}}}(s,a)\mathcal{B}_{\mathcal{D}}Q_\lambda^k(s,a))}{(1-\lambda)P_{\hat{\mathcal{D}}}(s,a)+\lambda P_{\mathcal{D}'}(s,a)}||\right]\right]
$$
$$
+\mathbb{E}_{(s,a)\sim\mathcal{D}|\hat{\mathcal{D}}}\left[\mathbb{E}_{\hat{s}'\sim P_{\mathcal{D}}(\hat{s}'|s,a)}\left[||\frac{\lambda P_{\mathcal{D}'}(s,a)(\mathcal{B}_{\mathcal{D}'}Q_\lambda^k(s,a)-\mathcal{B}_{\mathcal{D}}Q_\lambda^k(s,a))}{(1-\lambda)P_{\hat{\mathcal{D}}}(s,a)+\lambda P_{\mathcal{D}'}(s,a)}||\right]\right]
$$
$$
+\mathbb{E}_{(s,a)\sim\mathcal{D}|\mathcal{S}\times\mathcal{A}\setminus\hat{\mathcal{D}}}\left[\mathbb{E}_{\hat{s}'\sim P_{\mathcal{D}}(\hat{s}'|s,a)}\left[||e^{-NC}\cdot(\mathcal{B}_{\mathcal{D}'}Q_\lambda^k(s,a)-\mathcal{B}_{\mathcal{D}}Q_\lambda^k(s,a))||\right]\right], \tag{147}
$$

where the previous inequality follows from the proof of Theorem 1.

Dividing both the numerator and denominator by $P_{\hat{\mathcal{D}}}(s,a)$ in the first term and by $P_{\mathcal{D}'}(s,a)$ in the second term of the right hand side of the previous inequality, and re-ordering terms yields

$$
\mathbb{E}_{\hat{s}'\sim P_{\mathcal{D}}(\hat{s}'|s,a)}\left[\mathbb{E}_{(s,a)\sim\mathcal{D}}\left[||Q_\lambda^{k+1}(s,a)-\mathcal{B}_{\mathcal{D}}Q_\lambda^k(s,a)||_\infty\right]\right]
$$
$$
\leq \mathbb{E}_{(s,a)\sim\mathcal{D}|\hat{\mathcal{D}}}\left[\mathbb{E}_{\hat{s}'\sim P_{\mathcal{D}}(\hat{s}'|s,a)}\left[||\frac{(1-\lambda)\left(\frac{1}{N(s,a)}\sum_{j=1}^{N(s,a)}\hat{\mathcal{B}}_{\hat{s}_j'}Q_\lambda^k(s,a)-\mathcal{B}_{\mathcal{D}}Q_\lambda^k(s,a)\right)}{1-\lambda+\lambda P_{\mathcal{D}'}(s,a)/P_{\hat{\mathcal{D}}}(s,a)}||\right]\right]
$$
$$
+\mathbb{E}_{(s,a)\sim\mathcal{D}|\hat{\mathcal{D}}}\left[\mathbb{E}_{\hat{s}'\sim P_{\mathcal{D}}(\hat{s}'|s,a)}\left[||\frac{\lambda(\mathcal{B}_{\mathcal{D}'}Q_\lambda^k(s,a)-\mathcal{B}_{\mathcal{D}}Q_\lambda^k(s,a))}{(1-\lambda)P_{\hat{\mathcal{D}}}(s,a)/P_{\mathcal{D}'}(s,a)+\lambda}||\right]\right]
$$
$$
+\mathbb{E}_{(s,a)\sim\mathcal{D}|\mathcal{S}\times\mathcal{A}\setminus\hat{\mathcal{D}}}\left[\mathbb{E}_{\hat{s}'\sim P_{\mathcal{D}}(\hat{s}'|s,a)}\left[||e^{-NC}\cdot(\mathcal{B}_{\mathcal{D}'}Q_\lambda^k(s,a)-\mathcal{B}_{\mathcal{D}}Q_\lambda^k(s,a))||\right]\right]. \tag{148}
$$

Assumption 3 implies that both $P_{\mathcal{D}'}(s,a)/P_{\hat{\mathcal{D}}}(s,a)$ and $P_{\hat{\mathcal{D}}}(s,a)/P_{\mathcal{D}'}(s,a)$ are bounded, i.e.,

$$
\frac{1}{\beta_u}\leq\frac{P_{\mathcal{D}'}(s,a)}{P_{\hat{\mathcal{D}}}(s,a)}\leq\frac{1}{\beta_l},\ \beta_l\leq\frac{P_{\hat{\mathcal{D}}}(s,a)}{P_{\mathcal{D}'}(s,a)}\leq\beta_u,\ \forall(s,a)\in\hat{\mathcal{D}}. \tag{149}
$$

Applying the previous bounds to (148) yields

$$
\mathbb{E}_{\hat{s}'\sim P_{\mathcal{D}}(\hat{s}'|s,a)} \left[ \mathbb{E}_{(s,a)\sim\mathcal{D}} \left[ \|Q_\lambda^{k+1}(s,a) - \mathcal{B}_{\mathcal{D}}Q_\lambda^k(s,a)\|_\infty \right] \right]
$$

$$
\leq \mathbb{E}_{(s,a)\sim\mathcal{D}|\hat{\mathcal{D}}} \left[ \mathbb{E}_{\hat{s}'\sim P_{\mathcal{D}}(\hat{s}'|s,a)} \left[ \| \frac{(1-\lambda)\left(\frac{1}{N(s,a)}\sum_{j=1}^{N(s,a)}\hat{\mathcal{B}}_{\hat{s}'_j}Q_\lambda^k(s,a) - \mathcal{B}_{\mathcal{D}}Q_\lambda^k(s,a)\right)}{1-\lambda+\lambda/\beta_u} \| \right] \right]
$$

$$
+ \mathbb{E}_{(s,a)\sim\mathcal{D}|\hat{\mathcal{D}}} \left[ \mathbb{E}_{\hat{s}'\sim P_{\mathcal{D}}(\hat{s}'|s,a)} \left[ \| \frac{\lambda(\mathcal{B}_{\mathcal{D}'}Q_\lambda^k(s,a) - \mathcal{B}_{\mathcal{D}}Q_\lambda^k(s,a))}{(1-\lambda)\beta_l+\lambda} \| \right] \right]
$$

$$
+ \mathbb{E}_{(s,a)\sim\mathcal{D}|\mathcal{S}\times\mathcal{A}\backslash\hat{\mathcal{D}}} \left[ \mathbb{E}_{\hat{s}'\sim P_{\mathcal{D}}(\hat{s}'|s,a)} \left[ \| e^{-NC}\cdot(\mathcal{B}_{\mathcal{D}'}Q_\lambda^k(s,a) - \mathcal{B}_{\mathcal{D}}Q_\lambda^k(s,a)) \| \right] \right]. \tag{150}
$$

Extracting the constant terms in the above expression outside expectations and using the fact that $\mathcal{B}_{\mathcal{D}}Q_\lambda^k(s,a)$ and $\mathcal{B}_{\mathcal{D}'}Q_\lambda^k(s,a)$ do not depend on $\hat{s}'$ (at the step $k+1$) yields

$$
\mathbb{E}_{\hat{s}'\sim P_{\mathcal{D}}(\hat{s}'|s,a)} \left[ \mathbb{E}_{(s,a)\sim\mathcal{D}} \left[ \|Q_\lambda^{k+1}(s,a) - \mathcal{B}_{\mathcal{D}}Q_\lambda^k(s,a)\|_\infty \right] \right]
$$

$$
\leq \frac{1-\lambda}{1-\lambda+\lambda/\beta_u} \mathbb{E}_{(s,a)\sim\mathcal{D}|\hat{\mathcal{D}}} \left[ \mathbb{E}_{\hat{s}'\sim P_{\mathcal{D}}(\hat{s}'|s,a)} \left[ \| \left( \frac{1}{N(s,a)} \sum_{j=1}^{N(s,a)} \hat{\mathcal{B}}_{\hat{s}'_j}Q_\lambda^k(s,a) - \mathcal{B}_{\mathcal{D}}Q_\lambda^k(s,a) \right) \| \right] \right]
$$

$$
+ \frac{\lambda}{(1-\lambda)\beta_l+\lambda} \mathbb{E}_{(s,a)\sim\mathcal{D}|\hat{\mathcal{D}}} \left[ \|(\mathcal{B}_{\mathcal{D}}Q_\lambda^k(s,a) - \mathcal{B}_{\mathcal{D}'}Q_\lambda^k(s,a))\| \right]
$$

$$
+ e^{-NC}\cdot \mathbb{E}_{(s,a)\sim\mathcal{D}|\mathcal{S}\times\mathcal{A}\backslash\hat{\mathcal{D}}} \left[ \|(\mathcal{B}_{\mathcal{D}}Q_\lambda^k(s,a) - \mathcal{B}_{\mathcal{D}'}Q_\lambda^k(s,a))\| \right] \tag{151}
$$

$$
\leq \frac{1-\lambda}{1-\lambda+\lambda/\beta_u} \max_{(s,a)\in\hat{\mathcal{D}}} \left[ \mathbb{E}_{\hat{s}'\sim P_{\mathcal{D}}(\hat{s}'|s,a)} \left[ \| \left( \frac{1}{N(s,a)} \sum_{j=1}^{N(s,a)} \hat{\mathcal{B}}_{\hat{s}'_j}Q_\lambda^k(s,a) - \mathcal{B}_{\mathcal{D}}Q_\lambda^k(s,a) \right) \| \right] \right]
$$

$$
+ \left( \frac{\lambda}{(1-\lambda)\beta_l+\lambda} + e^{-NC} \right) \max_{(s,a)\in\mathcal{S}\times\mathcal{A}} \left[ \|(\mathcal{B}_{\mathcal{D}}Q_\lambda^k(s,a) - \mathcal{B}_{\mathcal{D}'}Q_\lambda^k(s,a))\| \right]. \tag{152}
$$

By definition of $\varsigma_{\max}$ and $\xi_{\max}$ we obtain

$$
\mathbb{E}_{\hat{s}'\sim P_{\mathcal{D}}(\hat{s}'|s,a)} \left[ \mathbb{E}_{(s,a)\sim\mathcal{D}} \left[ \|Q_\lambda^{k+1}(s,a) - \mathcal{B}_{\mathcal{D}}Q_\lambda^k(s,a)\|_\infty \right] \right] \leq \frac{1-\lambda}{1-\lambda+\lambda/\beta_u}\sqrt{\varsigma_{\max}} + \frac{\lambda}{(1-\lambda)\beta_l+\lambda}\sqrt{\xi_{\max}}
$$

$$
+ e^{-NC}\sqrt{\xi_{\max}}, \ \forall k\in\mathbb{N}. \tag{153}
$$

This completes the proof of Lemma 2. $\qquad\square$

Having introduced Lemma 2, we are in the stage of proving Theorem 4.

**Theorem 4** (Convergence). Let the conditions of Theorem 1 hold. Given any dataset $\hat{\mathcal{D}}_{\text{tr}}$, it holds at each iteration $(k=0,1,2,\cdots)$ that

$$
\mathbb{E}_{\hat{s}'\sim P_{\mathcal{D}}(\hat{s}'|s,a)} \left[ \mathbb{E}_{(s,a)\sim\mathcal{D}} \left[ \|Q_\lambda^{k+1}(s,a) - Q^*(s,a)\|_\infty \right] \right] \leq
$$

$$
\gamma^{k+1} \mathbb{E}_{(s,a)\sim\mathcal{D}} \left[ \|Q^0(s,a) - Q^*(s,a)\|_\infty \right] + \frac{1-\gamma^{k+1}}{1-\gamma} \left( \frac{1-\lambda}{1-\lambda+\lambda/\beta_u}\sqrt{\varsigma_{\max}} + \left( \frac{\lambda}{(1-\lambda)\beta_l+\lambda} + e^{-NC} \right)\sqrt{\xi_{\max}} \right). \tag{154}
$$

*Proof.* From the triangle inequality we obtain

$$
\mathbb{E}_{\hat{s}'\sim P_{\mathcal{D}}(\hat{s}'|s,a)} \left[ \mathbb{E}_{(s,a)\sim\mathcal{D}} \left[ ||Q_\lambda^{k+1}(s,a) - Q^*(s,a)||_\infty \right] \right] \tag{155}
$$

$$
\leq \mathbb{E}_{\hat{s}'\sim P_{\mathcal{D}}(\hat{s}'|s,a)} \left[ \mathbb{E}_{(s,a)\sim\mathcal{D}} \left[ ||Q_\lambda^{k+1}(s,a) - \mathcal{B}_{\mathcal{D}}Q_\lambda^k(s,a)||_\infty + ||\mathcal{B}_{\mathcal{D}}Q_\lambda^k(s,a) - Q^*(s,a)||_\infty \right] \right]. \tag{156}
$$

Given that $Q^*$ is the fixed point of the Bellman optimality operator (2) (i.e., $\mathcal{B}_{\mathcal{D}}Q^*(s,a) = Q^*(s,a)$), we obtain

$$
\mathbb{E}_{\hat{s}'\sim P_{\mathcal{D}}(\hat{s}'|s,a)} \left[ \mathbb{E}_{(s,a)\sim\mathcal{D}} \left[ ||Q_\lambda^{k+1}(s,a) - Q^*(s,a)||_\infty \right] \right]
$$

$$
\leq \mathbb{E}_{\hat{s}'\sim P_{\mathcal{D}}(\hat{s}'|s,a)} \left[ \mathbb{E}_{(s,a)\sim\mathcal{D}} \left[ ||Q_\lambda^{k+1}(s,a) - \mathcal{B}_{\mathcal{D}}Q_\lambda^k(s,a)||_\infty + ||\mathcal{B}_{\mathcal{D}}Q_\lambda^k(s,a) - \mathcal{B}_{\mathcal{D}}Q^*(s,a)||_\infty \right] \right]. \tag{157}
$$

Applying the contraction property (Sutton & Barto, 2018) of the Bellman optimality operator to the previous inequality and re-arranging the terms yields

$$
\mathbb{E}_{\hat{s}'\sim P_{\mathcal{D}}(\hat{s}'|s,a)} \left[ \mathbb{E}_{(s,a)\sim\mathcal{D}} \left[ ||Q_\lambda^{k+1}(s,a) - Q^*(s,a)||_\infty \right] \right]
$$

$$
\leq \mathbb{E}_{\hat{s}'\sim P_{\mathcal{D}}(\hat{s}'|s,a)} \left[ \mathbb{E}_{(s,a)\sim\mathcal{D}} \left[ ||Q_\lambda^{k+1}(s,a) - \mathcal{B}_{\mathcal{D}}Q_\lambda^k(s,a)||_\infty + \gamma||Q_\lambda^k(s,a) - Q^*(s,a)||_\infty \right] \right] \tag{158}
$$

$$
= \mathbb{E}_{\hat{s}'\sim P_{\mathcal{D}}(\hat{s}'|s,a)} \left[ \mathbb{E}_{(s,a)\sim\mathcal{D}} \left[ ||Q_\lambda^{k+1}(s,a) - \mathcal{B}_{\mathcal{D}}Q_\lambda^k(s,a)||_\infty \right] \right]
$$

$$
+ \gamma \mathbb{E}_{\hat{s}'\sim P_{\mathcal{D}}(\hat{s}'|s,a)} \left[ \mathbb{E}_{(s,a)\sim\mathcal{D}} \left[ ||Q_\lambda^k(s,a) - Q^*(s,a)||_\infty \right] \right]. \tag{159}
$$

Unrolling the previous inequality until $Q_\lambda^0$ yields

$$
\mathbb{E}_{\hat{s}'\sim P_{\mathcal{D}}(\hat{s}'|s,a)} \left[ \mathbb{E}_{(s,a)\sim\mathcal{D}} \left[ ||Q_\lambda^{k+1}(s,a) - Q^*(s,a)||_\infty \right] \right]
$$

$$
\leq \mathbb{E}_{\hat{s}'\sim P_{\mathcal{D}}(\hat{s}'|s,a)} \left[ \mathbb{E}_{(s,a)\sim\mathcal{D}} \left[ ||Q_\lambda^{k+1}(s,a) - \mathcal{B}_{\mathcal{D}}Q_\lambda^k(s,a)||_\infty \right] \right]
$$

$$
+ \gamma \mathbb{E}_{\hat{s}'\sim P_{\mathcal{D}}(\hat{s}'|s,a)} \left[ \mathbb{E}_{(s,a)\sim\mathcal{D}} \left[ ||Q_\lambda^k(s,a) - \mathcal{B}_{\mathcal{D}}Q_\lambda^{k-1}(s,a)||_\infty \right] \right] + \cdots
$$

$$
+ \gamma^k \mathbb{E}_{\hat{s}'\sim P_{\mathcal{D}}(\hat{s}'|s,a)} \left[ \mathbb{E}_{(s,a)\sim\mathcal{D}} \left[ ||Q_\lambda^1(s,a) - \mathcal{B}_{\mathcal{D}}Q_\lambda^0(s,a)||_\infty \right] \right]
$$

$$
+ \gamma^{k+1} \mathbb{E}_{\hat{s}'\sim P_{\mathcal{D}}(\hat{s}'|s,a)} \left[ \mathbb{E}_{(s,a)\sim\mathcal{D}} \left[ ||Q_\lambda^0(s,a) - Q^*(s,a)||_\infty \right] \right]. \tag{160}
$$

Since $Q^0$ is the initial Q-function, it holds that

$$
Q^0(s,a) = Q_\lambda^0(s,a), \ \forall(s,a) \in \mathcal{S} \times \mathcal{A}. \tag{161}
$$

Employing Lemma 2 further implies that

$$\mathbb{E}_{\hat{s}' \sim P_{\mathcal{D}}(\hat{s}'|s,a)} \left[ \mathbb{E}_{(s,a) \sim \mathcal{D}} \left[ ||Q_{\lambda}^{k+1}(s,a) - Q^*(s,a)||_{\infty} \right] \right] \leq$$
$$\left( \frac{1-\lambda}{1-\lambda+\lambda/\beta_u} \sqrt{\varsigma_{\max}} + \left( \frac{\lambda}{(1-\lambda)\beta_l+\lambda} + e^{-NC} \right) \sqrt{\xi_{\max}} \right) + \cdots +$$
$$\gamma^k \left( \frac{1-\lambda}{1-\lambda+\lambda/\beta_u} \sqrt{\varsigma_{\max}} + \left( \frac{\lambda}{(1-\lambda)\beta_l+\lambda} + e^{-NC} \right) \sqrt{\xi_{\max}} \right) + \gamma^{k+1} \mathbb{E}_{(s,a) \sim \mathcal{D}} \left[ ||Q^0(s,a) - Q^*(s,a)||_{\infty} \right]$$

(162)

$$= \gamma^{k+1} \mathbb{E}_{(s,a) \sim \mathcal{D}} \left[ ||Q^0(s,a) - Q^*(s,a)||_{\infty} \right]$$
$$+ \frac{1-\gamma^{k+1}}{1-\gamma} \left( \frac{1-\lambda}{1-\lambda+\lambda/\beta_u} \sqrt{\varsigma_{\max}} + \left( \frac{\lambda}{(1-\lambda)\beta_l+\lambda} + e^{-NC} \right) \sqrt{\xi_{\max}} \right).$$

(163)

This completes the proof of Theorem 4. □

## A.8 *Procgen* Environments

We select five *Procgen* games (Cobbe et al., 2020) to substantiate our theoretical contributions in this work, whose details are provided below.

**Description of *Caveflyer* (Cobbe et al., 2020).** "The player needs to traverse a complex network of caves to reach the exit. Player movement is reminiscent of the classic Atari game "Asteroids" where the ship can rotate and propel forward or backward along its current axis. The primary reward is granted upon successfully reaching the end of the level, though additional reward can be earned by destroying target objects with the ship's lasers along the way. The level is fraught with both stationary and moving lethal obstacles, demanding precise navigation and quick reflexes."

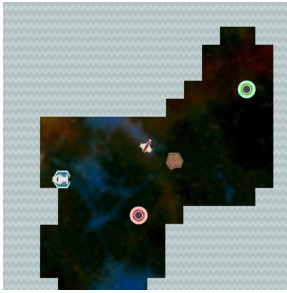

Figure 8: The screenshot of *Caveflyer* (Cobbe et al., 2020).

**Description of *Climber* (Cobbe et al., 2020).** "The player needs to climb a series of platforms, collecting stars scattered along the path. A small reward is granted for each star collected, with a substantial reward provided for gathering all stars within a level. If every star is collected, the episode terminates. The level is also populated with lethal flying monsters, adding extra challenges to the player's journey."

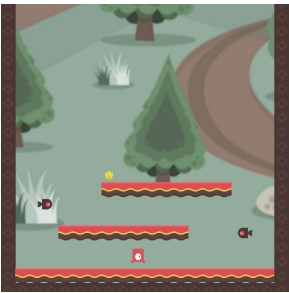

Figure 9: The screenshot of *Climber* (Cobbe et al., 2020).

**Description of *Dodgeball* (Cobbe et al., 2020).** "Inspired by the Atari game "Berzerk", the player spawns in a room with a randomly generated configuration of walls and enemies. Contact with a wall results in an immediate game over, terminating the episode. The player moves slowly, allowing for careful navigation throughout the room. Enemies, moving slowly too, throw balls at the player. The player can retaliate by throwing balls as well, but only in the direction they are facing. Once all enemies are eliminated, the player can advance to the unlocked platform, earning a substantial level completion bonus."

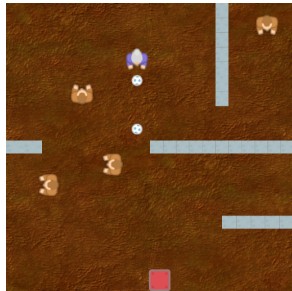

Figure 10: The screenshot of *Dodgeball* (Cobbe et al., 2020).

**Description of *Maze* (Cobbe et al., 2020).** "The player, embodying a mouse, needs to navigate a maze to locate the sole piece of cheese and obtain a reward. The mazes, generated using Kruskal's algorithm, vary in size from $3 \times 3$ to $25 \times 25$, with dimensions uniformly sampled across this range. To navigate the maze, the player can move up, down, left, or right."

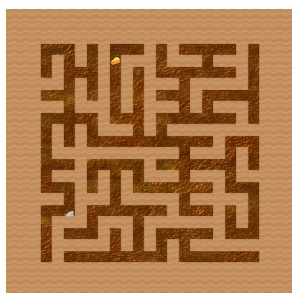

Figure 11: The screenshot of *Maze* (Cobbe et al., 2020).

**Description of *Miner* (Cobbe et al., 2020).** "Inspired by the game "BoulderDash", the player (robot) can dig through dirt to navigate the world. The game world is governed by gravity, where dirt supports both boulders and diamonds. Boulders and diamonds fall through free spaces and roll off each other. If either a boulder or a diamond falls on the player, the game terminates immediately. The objective is to collect

all the diamonds in the level and then reach the exit. The player earns a small reward for each diamond collected and a huge reward for successful completion in the level."

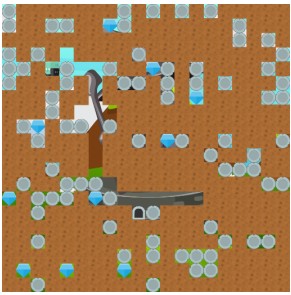

Figure 12: The screenshot of *Miner* (Cobbe et al., 2020).

***Procgen* levels.** An example (*Maze*) with different *Procgen* levels is provided in Figure 13.

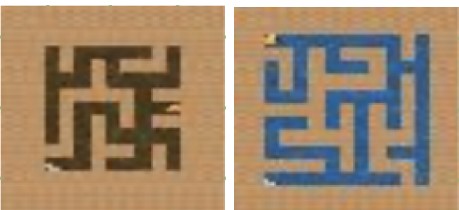

Figure 13: *Maze* with different levels.

## A.9  *Procgen* Experimental Hyperparameters

Key hyperparameters for the datasets and algorithms are summarized in Table 5.

Table 5: Experimental hyperparameters.

| Hyperparameters | Value |
|---|---|
| Target domain levels | $[100, 199]$ |
| Source domain levels | $[0, 99], [25, 124], [50, 149]$ |
| Number of target samples ($N$) | $1000, 2500, 4000$ |
| Number of source samples ($N'$) | $40000$ |
| Weight ($\lambda$) | $\{0, 0.2, 0.4, 0.5, 0.6, 0.8, 1\}$ |
| Number of episodes for evaluation | $500$ |
| Learning rate | $0.0005$ |
| Batch size | $256$ |
| Neural network hidden size | $256$ |
| Discount factor ($\gamma$) | $0.99$ |
| CQL conservativeness constant ($\alpha$) | $4$ |
| Gradient norm clip | $0.1$ |
| IQL expectile ($\tau_{\text{exp}}$) | $0.8$ |
| IQL temperature ($\beta$) | $0.1$ |

