# OpenReview forum: "Provable Domain Adaptation for Offline Reinforcement Learning with Limited Samples"
_TMLR — Accepted by TMLR_

### Review · Reviewer_YFb3 · 2025-11-19

**Summary Of Contributions:**

This paper studies offline reinforcement learning where you have a small, target dataset and an unlimited but biased source dataset, e.g., a simulator, and asks how to optimally combine them. The authors propose an algorithm-agnostic framework which updates Q by minimizing a $\lambda$-weighted TD loss over target and source transitions and develop a theory that decomposes error into variance from limited target samples and bias from the dynamics gap between source and target MDPs. Within a tabular setting and under standard coverage and boundedness assumptions, they derive expected and worst-case performance bounds, show the algorithm converges to a neighborhood of the optimal $Q^{\star}$, and, under simplifying assumptions on sampling ratios, obtain a closed-form optimal weight $\lambda^{\star}$ which balances variance and bias and is typically not one of the trivial choices using only source, only target, or equal weighting. They have also conducted experiments on the offline Procgen benchmark with CQL and IQL backbones showing that performance as a function of $\lambda$ qualitatively matches the theoretical predictions and that naive weighting choices are often suboptimal, though the theoretically optimal $\lambda$ depends on quantities (eg. dynamics gap) which are hard to estimate in practice.

**Audience:**

Yes

**Audience Explanation:**

I think the paper offers a useful, theoretically grounded perspective on a problem of broad interest in the community. It is well written and, in my opinion, would add value to the current body of literature.

**Broader Impact Concerns:**

There are no broader impact concerns.

**Claims And Evidence:**

Yes

**Claims Explanation:**

This paper presents a neat and unifying formulation, and is overall well written. The derivations in the tabular setting under exact minimization and coverage clearly support the basic theoretical claims: 1) that $\\lambda$-weighted TD updates induce a bias–variance tradeoff between source and target data, 2) that one can obtain expected and worst-case bounds on error, and 3) that there is, under simplifying assumptions, a closed-form optimal $\\lambda^*$ balancing dynamics gap and variance. The main results appear technically sound at a high level. However, all the provided guarantees depend on a set of relatively strong assumptions: a finite tabular MDP, exact minimization of the TD objective, full coverage of the state–action space, and effectively unlimited source data such that expectations under $D'$ can be treated as known. In my opinion, another challenge is thereby introduced in establishing the practicality of the proposed algorithm.

**Requested Changes:**

The worst case performance bound, Theorem~3, has a dependence on $|\mathcal{S}||\mathcal{A}|$, a number of $\\beta$-type constants, and the usual $1/\sqrt{N}$ and thus is almost certainly vacuous in non-tiny settings. The analysis itself appears to be standard in spirit (Hoeffding-type plus a union bound) though the paper does not illustrate numerically how large these bounds are even for modest $|\mathcal{S}|$ and $|\mathcal{A}|$. Furthermore, the bound is used mainly to argue that ``we have a generalization guarantee,'' rather than to inform algorithmic design or the choice of parameters. Given the TMLR focus, such work is strengthened either by (1) providing a small-scale example where the bound is numerically non-trivial or (2) explicitly stating that the worst case bound is primarily of conceptual interest.

---

> ### Author Response · Authors · 2025-12-09
> **Response to Reviewer YFb3**
>
> We sincerely thank the reviewer for providing insightful and constructive reviews. We are glad to hear that in the reviewer’s opinion our work is well-written, technically sound, theoretically grounded, presents a neat and unifying formulation, and is useful for the problem of broad interest in the community.
>
> The concern and suggestions raised by the reviewer can significantly improve the quality of our paper. Detailed responses to the specific questions raised by the reviewer follow.
>
>
>
> 1. ''*The main results appear technically sound at a high level. However, all the provided guarantees depend on a set of relatively strong assumptions: a finite tabular MDP, exact minimization of the TD objective, full coverage of the state–action space, and effectively unlimited source data such that expectations under $D'$ can be treated as known. In my opinion, another challenge is thereby introduced in establishing the practicality of the proposed algorithm.*''
> - **Response:** We sincerely appreciate the reviewer’s comments. **This is a very good point, as we also highlight in the Conclusion section**. The following modifications to the manuscript will be made to **make the focus and limitations of our work explicit throughout the manuscript and not only in the Conclusion**.
>
> **(i)** We will **make it more explicit in our revised manuscript that this work focuses on the tabular MDP**, where the exact optimization is not a critical challenge. Regarding the finiteness of the MDP, it becomes **standard in the literature unless we incorporate strong assumptions on the function approximation abilities of the parameterization and on the optimization**, which falls beyond the scope of this work. With respect to the full coverage of the state–action space, it is **NOT necessary for the target domain in our work as we explicitly discuss in Theorems 1 and 2**. The same applies to the source domain, but we have **simplified the presentation of our work** by assuming that the source domain contains a sufficiently large amount of samples. **This is formalized in Assumption 2 and its subsequent discussion**. We will provide a **more explicit discussion of these assumptions in our new version of the paper to address any potential concerns**.
>
> **(ii)** Our work primarily focuses on the theoretical analysis of our proposed framework that combines a limited target dataset and a large-but-biased source dataset, **rather than on developing an algorithm**. The analysis necessarily involves certain quantities that may be difficult to estimate in practice, such as the state-space size $|S|$ for pixel-based inputs (e.g., the offline *Procgen* environments in our paper), the action-space size $|A|$ in continuous-action tasks, and the dynamics gap $\xi$ between the source and target domains over the entire state-action space. **While we also think** that developing a practical algorithm to learn an approximate optimal weight **remains a promising direction for future research (refer to the Conclusion), it falls beyond the scope of this work**. In addition to highlighting it in the Conclusion, **we will make this discussion more explicit in the revised version of our paper when interpreting the theoretical results**. We believe this clarification will help address the reviewer’s subsequent concern as well.
>
>
>
> 2. ''*The bound is used mainly to argue that "we have a generalization guarantee", rather than to inform algorithmic design or the choice of parameters. Given the TMLR focus, such work is strengthened either by (1) providing a small-scale example where the bound is numerically non-trivial or (2) explicitly stating that the worst case bound is primarily of conceptual interest.*''
> - **Response:**  We thank the reviewer for pointing this out. As noted in our earlier response, this work focuses on the theoretical analysis of our proposed framework rather than on developing a practical algorithm. The worst-case performance bound might indeed be difficult to estimate and overly conservative in practice, **as we mention in Section 4.3**. Moreover, the worst-case performance bound offers less intuition than that of the expected performance bound for choosing the optimal weight $\lambda^\star$. Consequently, **the reviewer is correct that the worst-case performance bound in our paper is primarily of conceptual interest. We will make this clarification more explicit in our revised manuscript.**
>
>
>
> We are grateful for the reviewers' thoughtful feedback, which will guide us in strengthening the manuscript. The revisions will significantly enhance the rigor and clarity of our paper.

---

### Review · Reviewer_99q4 · 2025-11-29

**Summary Of Contributions:**

This paper studies hybrid offline reinforcement learning, where the agent optimizes its policy with a small set of offline data from a target domain alongside a substantially larger dataset from a source domain. Since the target data is scarce, it is insufficient to learn a meaningful policy on its own. However, leveraging the data from the potentially mismatched source domain can introduce generalization gaps in the final policy. Therefore, a straightforward choice is to coordinate the usage of both datasets with a mixing hyperparameter, denoted as $\lambda$ in this paper.

Motivated by this, this paper theoretically studies the relationship between $\lambda$, the size of the target dataset, and the dynamic bias. The major results include the expected and worst-case bounds on the differences between the empirical TD error and the expected TD error in the target domain. Besides, under simplifying conditions, this bound also implies the existence of an optimal weight $\lambda$. Finally, the experimental section subsequently validates the theoretical corollaries, particularly the trend of the optimal $\lambda$ as a function of the target dataset size and the dynamics gap.

**Additional Comments:**

Minor issue: It would be clearer to separate the notation used for the domains (the underlying dynamics) and the datasets (the empirical collections of transitions). Currently, they all share the notation of $\mathcal{D}$ (with or without hats), and this can lead to confusion when distinguishing between empirical operators and expected operators.

**Audience:**

Yes

**Audience Explanation:**

Yes. The problem studied in this paper is hybrid RL, where a probably mismatched simulator and a handful of offline interactions from the target domain are available for reinforcement learning. This combination effectively mitigates the critical challenges associated with using either data source independently: direct learning in the target domain is often prohibitively sample-costly, while merely learning within the simulator suffers from significant sim-to-real gaps. The final theoretical finding, although conceptually straightforward, establishes a connection between the mixing weight and the dataset/simulator conditions, which can potentially guide the hyperparameter choice prior to the actual learning process.

**Broader Impact Concerns:**

N/A.

**Claims And Evidence:**

No

**Claims Explanation:**

Part of the claims presented in this paper are supported by convincing theoretical analysis and empirical results, while others require clarification or correction. Here is a detailed comment.

1. Existing hybrid RL methods, including H2O and ORIS, are **mispresented** and therefore compromise the significance of the optimization objective studies in this paper. This paper studies a scheme where the critic loss is constructed by mixing the loss from the target domain dataset and the loss from the large source domain dataset using a constant mixing weight $\lambda$.  It then claims that existing hybrid RL methods, such as H2O and ORIS, can be recognized as instantiations of this approach with specific choices of $\lambda$. However, this claim is not correct. To be specific, H2O utilizes an importance-sampling-like weight to account for the dynamics mismatch between the source domain ($D'$) and the target domain ($D$). Specifically, it learns a discriminator to capture the transition ratio and uses this ratio to weight the samples from the source domain (as shown in Equation 8 of [1]):
$$\mathbb{E}\_{s'\sim P_{D'}(s'|s, a)}[\frac{P_D(s'|s, a)}{P_{D'}(s'|s, a)}\mathcal{E}(Q, s, a, s')]$$
However, this paper claims that H2O is using the unweighted expected error:
$$\mathbb{E}\_{s'\sim P_{D'}(s'|s, a)}[\mathcal{E}(Q, s, a, s')]$$
Similarly, ORIS uses a state-specific weight that quantifies the divergence between the dynamics at state $s$ to adjust the weight for each sample strategically. The constant weighting scheme employed and analyzed in this paper cannot encompass H2O and ORIS and may not be suitable for practical applications.

2. The final theoretical results implied the scaling trend between the optimal weight $\lambda$ and the dataset size and dynamics gap, respectively. This is supported and validated by the experiment section.


[1] When to Trust Your Simulator: Dynamics-Aware Hybrid Offline-and-Online Reinforcement Learning.

**Requested Changes:**

1. (Major concern) Please address my concerns about the misrepresentation of established hybrid RL methods if such misrepresentation does exist. Besides, to demonstrate the necessity and efficacy of the simpler weighting scheme analyzed in this paper, please explicitly compare and contrast it with the more complex, sample-specific weighting used by H2O and ORIS.

2. The current interpretation of the dynamics gap across different Procgen task levels is unclear. For the reader to properly interpret your theoretical claims and empirical results, please articulate how the dynamics vary between levels.

3. Experiments on locomotion or manipulation tasks (e.g., those in Gym-MuJoCo or DeepMind Control Suite) are strongly recommended. These environments are established testbeds for hybrid and domain-adaptation RL, as prior works (including H2O and ORIS) have focused on them. It is easy to create a series of tasks that differ in transition dynamics by tweaking certain parameters of the simulator, such as the gravity and friction coefficient.

---

> ### Author Response · Authors · 2025-12-18
> **Response to Reviewer 99q4 (Part 1)**
>
> We sincerely appreciate the reviewer’s thoughtful suggestions and feedback. We are pleased to hear that the reviewer finds our theoretical analysis and experimental results convincing, recognizes the significance of our work in mitigating the critical challenges associated with using either limited target dataset or large-but-biased source dataset independently, and acknowledges the potential of our work in guiding the hyperparameter choice prior to the actual learning process.
>
>
> Furthermore, we believe that the concerns and requested changes outlined by the reviewer will significantly strengthen our paper. Below, we provide detailed responses to each comment and requested change from the reviewer.
>
>
>
> 1. ''*(Major concern) Please address my concerns about the misrepresentation of established hybrid RL methods if such misrepresentation does exist. Besides, to demonstrate the necessity and efficacy of the simpler weighting scheme analyzed in this paper, please explicitly compare and contrast it with the more complex, sample-specific weighting used by H2O and ORIS.*''
> - **Response:** We sincerely thank the reviewer for bringing this to our attention.
>
> **i)** We would like to clarify that **our work focuses on the theoretical analysis** of our proposed framework that combines a limited target dataset and a large-but-biased source dataset, **rather than on developing an algorithm that outperforms H2O or ORIS to achieve a state-of-the-art performance**. We center on the **phase prior to the actual learning process, as correctly noted by the reviewer**. In this context, we are interested in the proper trade-off between the **"two datasets" (fixed and pre-collected) instead of specific samples during the actual learning process**, where employing a constant weight $\lambda$ is a straightforward scheme. **While we also think that developing a practical algorithm** to learn an approximate optimal weight or adaptive weight **remains a promising direction for future research (as we mention in the Conclusion), it falls beyond the scope of this work**. We will make the focus and limitations of our work **more explicit** in the revised manuscript.
>
> **ii)** **We agree with the reviewer** that our presentation of hybrid RL literature such as H2O and ORIS should be corrected. While both our framework and the existing literature highlight the necessity of weighted loss functions for the target and source datasets, **the focus differs**: prior work emphasizes **algorithm designs** that use dynamic weights (e.g., the transition ratio $P_M / P_{\hat{M}}$ in H2O) that vary per sample during the learning process, whereas our scheme centers on the trade-off between the two datasets (across **all samples**) **prior to** the learning process. **More importantly, our work establishes the first theoretical guarantees** on the performance bounds that explicitly depend on the variance of the target dataset and the dynamics gap between the source and target domains, and provides a **closed-form solution for the optimal weight** under simplifying condiditions. **In the new version of our paper, we will make this discussion explicit and revise the literature presentation accordingly.**
>
>
>
> 2. ''*The current interpretation of the dynamics gap across different Procgen task levels is unclear. For the reader to properly interpret your theoretical claims and empirical results, please articulate how the dynamics vary between levels.*''
> - **Response:** We sincerely thank the reviewer for pointing this out. In *Procgen*, **different level can have distinct layouts** (such as the **amount and position of various items and moving entities**). As a result, the **same action** taken in the **same state** can lead to **different successor states depending on the level** (e.g., being blocked by a moving entity in one level but not in another), yielding **level-dependent transition dynamics**. An introduction to *Procgen levels* can also be found in Section E of [1]. **In our revised paper, we will make this discussion more clear and will include a visual comparison of different levels**.
>
> [1] Mediratta et al. The Generalization Gap in Offline Reinforcement Learning. ICLR 2024.

---

> ### Author Response · Authors · 2025-12-18
> **Response to Reviewer 99q4 (Part 2)**
>
> 3. ''*Experiments on locomotion or manipulation tasks (e.g., those in Gym-MuJoCo or DeepMind Control Suite) are strongly recommended. These environments are established testbeds for hybrid and domain-adaptation RL, as prior works (including H2O and ORIS) have focused on them. It is easy to create a series of tasks that differ in transition dynamics by tweaking certain parameters of the simulator, such as the gravity and friction coefficient.*''
> - **Response:** We sincerely appreciate the reviewer’s suggestions. As emphasized in the paper, **our work primarily focuses on theoretical analysis. Nevertheless, we agree with the reviewer that additional experiments on standard benchmarks could be beneficial**. To this end, we consider **standard MuJoCo tasks**, where the target dataset contains a limited number of samples ($N$) and the source dataset exhibits a dynamics gap ($\xi$) relative to the target. **Following the reviewer’s suggestion**, the **dynamics gap** between the source and target datasets is induced by **differences in gravity and friction coefficients**. As with the *Procgen* experiments, the **new MuJoCo experiments continue to validate our theoretical analysis**, which will be included **in the revised version of our paper**.
>
>
>
> 4. ''*Minor issue: It would be clearer to separate the notation used for the domains (the underlying dynamics) and the datasets (the empirical collections of transitions). Currently, they all share the notation of $D$ (with or without hats), and this can lead to confusion when distinguishing between empirical operators and expected operators.*''
> - **Response:** Thank you for raising this point. Our intention was to use **the same notation** to emphasize that **the empirical collections of transitions** (*dataset $\hat{D}$*) are drawn from **"the corresponding" (NOT any other) underlying dynamics** (*domain* $D$). **With or without hats** is also used to distinguish the sample from the *dataset $\hat{D}$* (w.r.t. empirical Bellman operator) and *domain* $D$ (w.r.t. expected Bellman operator). We would be **glad to further clarify and address any remaining questions or potential confusion**.

---

> ### Author Response · Authors · 2025-12-18
>
> We are grateful for the reviewers' thoughtful feedback, which will guide us in strengthening the manuscript. The revisions will significantly enhance the rigor and clarity of our paper.

---

### Review · Reviewer_cXkD · 2025-12-08

**Summary Of Contributions:**

This paper proposes the first framework that theoretically explores the impact of the weights assigned to each dataset on the performance
of offline RL.  Experiments on the Procgen benchmark prove the point.

**Audience:**

Yes

**Audience Explanation:**

A good solution for offline RL applications that don't have more data.

**Broader Impact Concerns:**

No more Broader Impact Concerns.

**Claims And Evidence:**

Yes

**Claims Explanation:**

In the 1.1 related work, Offline reinforcement learning with a domain adaptation method ignores the weight's importance. The paper discusses the impact of the weights assigned to the two datasets on the performance of offline RL.

**Requested Changes:**

Several problems need to be clarified,
1. The theoretical results characterize the optimal $\lambda^\*$ in terms of $\varsigma$, $\xi$, and $\beta$, but these quantities are not directly observable in practice. Do you have a concrete, implementable procedure to choose \lambda without access to the true MDP dynamics?
2. Have the authors attempted any preliminary experiments outside Procgen, and if so, what were the outcomes?
3. In experiment setups, are there concrete examples where $\lambda = 0$ fails primarily because of coverage gaps rather than sample size?

---

> ### Author Response · Authors · 2025-12-18
> **Response to Reviewer cXkD**
>
> We sincerely thank the reviewer for providing a thoughtful review. We are glad to learn that the reviewer recognizes the importance and contributions of our work and regards it as a promising solution for offline RL applications with limited samples.
>
> In addition, we believe that the questions and suggestions raised by the reviewer will help improve the clarity and quality of our paper. Below, we provide detailed responses to each of the reviewer’s requested changes.
>
>
>
> 1. ''*The theoretical results characterize the optimal $\lambda^\star$ in terms of $\varsigma$, $\xi$, and $\beta$, but these quantities are not directly observable in practice. Do you have a concrete, implementable procedure to choose $\lambda$ without access to the true MDP dynamics?*''
> - **Response:** We sincerely appreciate the reviewer’s comments. Notice that our work **primarily focuses on the theoretical analysis of our proposed framework** that combines a limited target dataset and a large-but-biased source dataset, **rather than on developing an algorithm**. Our theoretical results involve quantities that may be challenging to estimate in practice such as the dynamics gap $\xi$ between the source and target domains, **especially when the true MDP dynamics are not accessible**, since $\xi$ depends explicitly on the transition dynamics of the underlying MDPs (see Eq. (14) and Eq. (2)). **While we also believe that developing a practical algorithm** to learn an approximate optimal weight **remains a promising direction for future research (as we mention in the Conclusion), it falls beyond the scope of our work**. We will make this discussion more explicit in the revised version of our manuscript.
>
>
>
> 2. ''*Have the authors attempted any preliminary experiments outside Procgen, and if so, what were the outcomes?*''
> - **Response:** We sincerely thank the reviewer for bringing this to our attention. Although **our work primarily focuses on theoretical analysis**, we believe that incorporating **additional experiments on other benchmarks would further strengthen the empirical validation of our work**. To this end, we now consider **standard MuJoCo tasks**, in which the target dataset contains a limited number of samples ($N$) and the source dataset exhibits a dynamics gap ($\xi$) relative to the target. The **dynamics gap** between the source and target datasets is induced by **differences in gravity and friction coefficients**. Consistent with the *Procgen* experiments, the **new MuJoCo experiments further support our theoretical results**, and will be included **in the revised version of our paper**.
>
>
>
> 3. ''*In experiment setups, are there concrete examples where $\lambda = 0$ fails primarily because of coverage gaps rather than sample size?*''
> - **Response:** Thank you for raising a good point. Let us focus on the scenario of $\lambda =0$, which corresponds to using the limited target dataset solely. Since our data is randomly drawn from the domain (i.e., **uniform coverage**), the **expected performance** of offline RL **"generally" improves with the sample size**, i.e., a larger $N$ typically leads to better performance at $\lambda =0$. **This trend can be observed in Figure 4**. Nonetheless, **we agree with the reviewer** that there might be cases in which **$\lambda =0$ fails due to coverage gaps rather than sample size**. For instance, **a large amount of data from a small portion** of the target domain may lead to poor performance, because of the **limited generalization capability** of RL.

---

> ### Author Response · Authors · 2025-12-18
>
> We are grateful for the reviewers' thoughtful feedback, which will guide us in strengthening the manuscript. The revisions will significantly enhance the rigor and clarity of our paper.

---

### Decision · Action_Editor_xBKE · 2026-01-25

**Recommendation:** Accept as is

**Audience:**

Yes

**Audience Explanation:**

The topic of domain adaptation is of broad interest in ML and remains an outstanding theoretical and empirical challenge. Offline RL is particularly challenging. Therefore, this paper should be of interest to audiences in TMLR in the RL and out-of-distribution ML communities.

**Claims And Evidence:**

Yes

**Claims Explanation:**

The authors adequately clarified the scope of their work, which is to theoretically and empirically study the impact of a simple mixing scheme between the source and target datasets (without any importance weighting) for offline RL. While one reviewer raised issues with novelty (especially in light of not studying more sophisticated importance weighting schemes), I do not believe this rises to the level of issues in accuracy/convincing/clear evidence.